# Patterns of suspended particulate matter across the continental margin in the Canadian Beaufort Sea during summer

Jens K. Ehn[1], Rick A. Reynolds[2], Dariusz Stramski[2], David Doxaran[3], Bruno Lansard[4], and Marcel Babin[5]

[1]Centre for Earth Observation Science, University of Manitoba, Winnipeg, Manitoba, Canada.
[2]Marine Physical Laboratory, Scripps Institution of Oceanography, University of California San Diego, La Jolla, California, U.S.A.
[3]Sorbonne Université, CNRS, Laboratoire d'Océanographie de Villefanche, Villefranche-sur-mer, France.
[4]Laboratoire des Sciences du Climat et de l'Environnement, LSCE/IPSL, CEA-CNRS-UVSQ-Université Paris Saclay, Gif-sur-Yvette, France
[5]Joint International ULaval-CNRS Laboratory Takuvik, Québec-Océan, Département de Biologie, Université Laval, Québec, Québec, Canada.

**Correspondence:** Jens K. Ehn (jens.ehn@umanitoba.ca)

**Abstract.** The particulate beam attenuation coefficient at 660 nm, $c_\mathrm{p}(660)$, was measured in conjunction with properties of suspended particle assemblages in August 2009 within the Canadian Beaufort Sea continental margin, a region heavily influenced by freshwater and sediment discharge from the Mackenzie River, but also by sea ice melt. The mass concentration of suspended particulate matter (SPM) ranged from 0.04 to 140 g m$^{-3}$, its composition varied from mineral to organic-dominated, and the median particle diameter determined over the range 0.7–120 $\mu$m varied from 0.78 to 9.45 $\mu$m, with the fraction of particles < 1 $\mu$m in surface waters reflecting the degree influenced by river water. Despite this range in particle characteristics, a strong relationship between SPM and $c_\mathrm{p}(660)$ was found, and used to determine SPM distributions across the shelf based on measurements of $c_\mathrm{p}(660)$ taken during summer seasons of 2004, 2008, and 2009. SPM spatial patterns on the stratified shelf reflected the vertically sheared two-layer estuarine circulation and SPM sources (i.e., fluvial inputs, bottom resuspension, and biological productivity). Along-shelf winds generated lateral Ekman flows, isopycnal movements, and upwelling or downwelling at the shelfbreak. Cross-shelf transects measured during three summers illustrate how sea ice meltwater affects river plume extent, while the presence of meltwater on the shelf was associated with enhanced near-bottom SPM during return flow of upwelled Pacific-origin water. SPM decreased sharply past the shelfbreak with further transport of particulate matter occurring near the bottom and in interleaving nepheloid layers. These findings expand our knowledge of particle distributions in the Beaufort Sea controlled by river discharge, sea ice, and wind, each of which is sensitive to weather and climate variations.

## 1 Introduction

The Mackenzie Shelf in the southeastern Beaufort Sea (Arctic Ocean) is subject to great seasonal and interannual variability in its sea ice coverage (Galley et al., 2008; Yang, 2009; Stroeve et al., 2014), freshwater input (McClelland et al., 2012), and

atmospheric forcing (Yang, 2009; Asplin et al., 2012; Moore, 2012; Kirillov et al., 2016), all of which strongly influence the water circulation and particle dynamics. The shelf is about 120 km wide, 500 km long, < 80 m deep, and is estimated to receive on average about 330 km$^3$ per year of freshwater from the Mackenzie River with a sediment load of 130 Tg per year (Macdonald et al., 1998; O'Brien et al., 2006). The large freshwater load, both from river runoff and sea ice melt, results in the Mackenzie Shelf displaying typical stratified estuarine circulation characteristics (Carmack and Macdonald, 2002). The Mackenzie Shelf is bordered to the east by Amundsen Gulf, to the west by the Mackenzie Trough, and is intersected at ∼134° W by Kugmallit Valley. These are all shown to be locations of intensified shelf-basin exchange driven by winds and modified by sea ice interactions (e.g., Dmitrenko et al., 2016; Forest et al., 2015; O'Brien et al., 2011; Williams and Carmack, 2008). Easterly along-shelf winds generate offshore Ekman transport of surface waters and upwelling of nutrient-rich Pacific-origin water onto the shelf, whereas westerly winds create downwelling flow and enhance offshore transport of sediment in the bottom boundary layer. Much of the sediment transport occurs during winter and is associated with storms, eddy transport, and sea ice brine convection (Forest et al., 2015; O'Brien et al., 2011).

The significance of sediment discharge to the region is underscored by the fact that this sediment load from the Mackenzie River surpasses the combined load of all other major rivers discharging into the Arctic Ocean. Additional sediment sources of minerogenic sediment to the shelf include coastal and bottom erosion, and other rivers, which have been estimated to provide ∼9 Tg per year (Macdonald et al., 1998). This makes the Mackenzie Shelf the most turbid shelf sea in the Arctic Ocean. Biological production, by both marine phytoplankton and sea ice associated algae towards the end of the ice-covered season, is a major authochthonous source of biogenous sediments in the Beaufort Sea during summer (Forest et al., 2007, 2010; Tremblay et al., 2008), although the ice and turbid seawater are thought to greatly limit primary production on the Mackenzie Shelf (Carmack and Macdonald, 2002). The particulate sinking flux therefore comprises highly variable fractions of allochthonous and autochthonous origins (Sallon et al., 2011), making particle characterization in the area a complex task. The vertical export of autochtonous organic material to the deep waters of Canada Basin is found to be surprisingly small, however (Honjo et al., 2010). As the organic material reaching the deep ocean layers is thousands of years old it must be transported there laterally from the shelf or slope reservoirs of highly refractory material (Honjo et al., 2010). This highlights the importance of understanding the distribution and lateral transport of particulate material from the shelf.

The mechanisms and pathways of cross-shelf and slope particle transport in the Beaufort Sea continental margin remain poorly understood (O'Brien et al., 2011). This is largely because of a lack of data of sufficient resolution; biogeochemically important constituents in such a large and dynamic system are difficult to characterize with traditional methods that rely on discrete water sampling. To infer particle transport pathways, a description of the distribution and variability of particle con-centrations associated with the factors controlling the water circulation is required. Ocean colour remote sensing of suspended particles provides a much better spatial coverage, but is limited to surface waters during cloud free conditons during certain periods of the seasonal cycle. In situ optical techniques, most commonly involving a measurement of beam attenuation coeffi-cient, allow a significant increase in observational time and space scales. The beam attenuation at light wavelength of 660 nm has been typically used in these relationships. Because beam attenuation is sensitive not only to the concentration of particles but also their size and composition, numerous relationships have been developed to relate the particulate beam attenuation

coefficient, $c_p(\lambda)$ (where $\lambda$ is light wavelength in vacuo) to the dry mass concentration of suspended particulate matter (SPM) and particulate organic carbon (POC) (e.g., Bishop, 1986, 1999; Bunt et al., 1999; Gardner et al., 2006; Stramski et al., 2008; Jackson et al., 2010; Hill et al., 2011). The relationships are affected by the proportion of organic to inorganic material, because mineral particles typically have higher refractive index compared to organic particles, and thus generally produce higher

scattering per unit mass concentration (e.g., Babin et al., 2003b; Woźniak et al., 2010). Beam attenuation is also affected by variable absorption. In particular, at 660 nm the absorption by chlorophyll pigments may cause important distinctions between organic and inorganic material (Doxaran et al., 2012; Bélanger et al., 2013). Particle size is of importance because the scattering cross-section of individual particles typically increases as particle size increases (Morel and Bricaud, 1986; Stramski and Kiefer, 1991). However, particle concentration often decreases significantly with an increase in particle size so that relatively

small particles can have higher contribution to bulk scattering per unit mass concentration of particles than larger particles (Babin et al., 2003b; Reynolds et al., 2010; Hill et al., 2011).

Because of various origins and variable composition of particle assemblages in the southeastern Beaufort Sea, the feasibility of inferring SPM and POC from beam attenuation has been questioned for this region (Jackson et al., 2010). Nevertheless, in this study we use a comprehensive set of field data collected as part of the MALINA project in summer 2009 in waters

with diverse composition of particulate matter characterized by variation in the ratio of POC/SPM to determine statistical relationships between the particulate beam attenuation coefficient at 660 nm, $c_p(660)$, and SPM and POC. These relationships are then applied to infer the particle concentration fields from the measurements of $c_p(660)$. The distribution of SPM and POC on the Mackenzie Shelf displayed complex spatial variability that could not be explained in terms of a single parameter. The variability was found to be related to forcing and oceanographic conditions (wind speed and direction, sea ice coverage, and

freshwater content and source), both present and foregone, which control the circulation and water mass properties on the shelf. To gain a better contextual understanding of the effect of the forcing and oceanographic conditions on particle concentration fields, we compare and contrast the MALINA observations to two other expeditions to the southeastern Beaufort Sea during the open water season that also included beam attenuation measurements.

## 2   Materials and Methods

### 2.1   MALINA sampling overview

The MALINA expedition was conducted from 31 July to 24 August 2009 in the southeastern Beaufort Sea on the research icebreaker *CCGS Amundsen*. A total of 167 CTD/Rosette casts were carried out during the expedition with water sampling conducted at 28 station locations (Fig. 1). The locations, sampling times and bottom depths are provided in Table S1 in the Supplementary Materials. A small barge was launched to conduct coincident surface water sampling away from the ship's

influence on 26 of these stations. In addition, the barge visited 12 additional stations in coastal waters too shallow for the ship (Fig. 1; Doxaran et al., 2012). The CTD/Rosette onboard the icebreaker was equipped with 24 12-liter Niskin bottles for water collection and various in situ instruments including an SBE-911plus CTD (Sea-bird Electronics, Inc.), a C-Star 25-cm beam

transmissometer (Wetlabs, Inc.) for measuring particulate beam attenuation coefficient at 660 nm, $c_p(660)$ in units of m$^{-1}$, and a Wetstar fluorometer (Wetlabs, Inc.) for measuring fluorescence of chlorophyll-*a* (chl-*a*).

## 2.2  Determinations of SPM and POC

Niskin bottles were triggered during CTD/Rosette upcasts to collect water samples at 3 to 4 depths, which always included the near-surface water (1.5–3 m depth range) and subsurface chlorophyll-*a* maximum (SCM) if present. To ensure representative sampling of entire particle assemblages within Niskin bottles (including particles settled below the level of the spigot), the full content of the 12-liter Niskin bottles was drained directly into 20-liter HDPE carboys (Nalgene) by opening the bottom lid (Knap et al., 1996). Aliquots were then sampled from the carboys after mixing. If sufficient volume of water was available, filtration for SPM and POC determinations was made in triplicate for each examined depth. However, this was not always possible in clear waters with low particle concentrations, in which case either duplicates or single samples were prepared. Water samples for SPM and POC on the barge were collected by directly submerging a 20-liter HDPE carboy below the sea surface (Doxaran et al., 2012). Doxaran et al. (2012) reports on coefficient of variations for SPM and POC for these surface samples measured in triplicate. Additional near-surface water samples were occasionally collected by lowering a bucket from the side of the ship.

Water samples for SPM and POC were filtered through 25 mm diameter Whatman GF/F filters under low vacuum ($\leq$ 5 psi). Prior to the cruise the filters for both SPM and POC determinations had been rinsed with Milli-Q water, combusted at 450 °C for 1.5 hours to remove organic material, and weighed using a Mettler-Toledo MT5 balance ($\pm0.001$ mg precision) to obtain the blank measurement of the filter mass. Filters were stored individually in Petri dishes until the time of sample filtration. The volume of filtered seawater was adjusted to optimize particle load on the filter, but not to cause filters to clog. This volume ranged from 0.2 L for very turbid samples collected near the Mackenzie River mouth (station 697) to 5.8 L at station 780. Immediately following filtration, filters were rinsed with about 50 mL of Milli-Q water to remove salts, transferred back to the Petri slides, and dried for 6-12 h at 55 °C. The dried filters were stored at $-80$ °C until processing. After the cruise, filters were again dried at 55 °C in the laboratory for about 24 h before measuring their dry weight using the same Mettler-Toledo MT5 balance. The SPM (in units of g m$^{-3}$) was determined by subtracting the blank filter mass from the sample filter mass and dividing by the volume of water filtered. The relative humidity of the room was about or below 40 % during weighing of filters to minimize the effect of uptake of moisture by the filters during the measurements. The protocol used for SPM determinations is consistent with standard methodology (e.g., Babin et al., 2003a).

SPM and POC were determined on the same GF/F filters. After the weighing for SPM, POC content was determined with an Organic Elemental Analyser (PerkinElmer 2400 Series II CHNS/O) with a standard high-temperature combustion method as described in Doxaran et al. (2012). Prior to insertion of samples into the analyzer, the filters were acidified with 200-350 $\mu$L of 2N HCl to remove inorganic carbon and then dried at 60 °C. Filters were compacted into small ($\sim$5 mm diameter) rounded pellets within pre-combusted aluminum foil. Blank filters for POC determinations were treated and measured in the same way as sample filters. The combustion temperature was kept at 925 °C. The final POC values (in units of g m$^{-3}$) were calculated by dividing the mass of organic carbon measured (in units of $\mu$g) on the sample filter (corrected for blank filter) by the filtered

volume. In these calculations, the correction for blank filters was made using the average mass concentration of organic carbon determined on 9 blank filters, which was determined to be $21.2 \pm 8.1$ $\mu$g (corresponding to a range of $\sim$2 to 50 % of measured signal for the sample filters).

## 2.3 Particle size distributions

The particle size distribution (PSD) of 54 discrete seawater samples collected with the CTD/Rosette or from the barge were measured using a Beckman-Coulter Multisizer III analyser following the method described by Reynolds et al. (2016). In 40 of these samples, data were collected using both the 30 $\mu$m and 200 $\mu$m aperture sizes and merged into a single PSD ranging from 0.7 $\mu$m to 120 $\mu$m. Seawater filtered through a 0.2 $\mu$m filter was used as the diluent and blank, and multiple replicate measurements were acquired for each sample. Each aperture was calibrated using microsphere standards following recommendations by the manufacturer. The average number of particles per unit volume within each size class, $N(D)$ (in units of m$^{-3}$), where $D$ is the midpoint diameter of the volume-equivalent sphere in each size class, was obtained after subtracting the counts for the blank. The particle volume distribution, $V(D)$ (dimensionless), was then calculated from $N(D)$ by assuming spherical particles.

## 2.4 Beam attenuation measurements

C-Star transmissometer data were recorded at 24 Hz as raw voltages and merged with the depth recording from the CTD/Rosette. Downcasts were processed to 1-m vertical bins centered at integers by averaging the interquartile range of the voltages within bins. This method effectively removed spikes and noise from the data, if present. Time series of transmissometer data were also collected at selected depths and processed similarly to above, by taking the average of the interquartile range of the voltage values recorded over the periods when the rosette was stopped for water sampling during upcasts. These data were used for correlational analysis with SPM and POC data from discrete Niskin bottle water samples. The particulate beam attenuation coefficient at 660 nm, $c_\mathrm{p}(660)$ (in units of m$^{-1}$), was then calculated from the binned voltage signal, $V_\mathrm{signal}$, as

$$c_\mathrm{p}(660) = -\ln\left(\frac{V_\mathrm{signal} - V_\mathrm{dark}}{V_\mathrm{ref} - V_\mathrm{dark}}\right)/x \tag{1}$$

where $x$ is the pathlength of 0.25 m, $V_\mathrm{dark}$ is the dark voltage offset, and $V_\mathrm{ref}$ is the reference voltage associated with particle free pure seawater (cf. C-Star User's Guide, Wetlabs, Inc.). For MALINA, $V_\mathrm{ref}$ was taken as the highest $V_\mathrm{signal}$ reading observed during the expedition, i.e., it was determined to be 4.7362 V (lower than the factory supplied value of 4.8340 V) observed with the same instrument during the Geotraces cruise that followed immediately the MALINA cruise (cast 0903_26 on 4 September at depths between 1900 and 2500 m where water temperature and salinity averaged $-0.40\,^\circ$C and 34.94 PSU, respectively). This $V_\mathrm{ref}$ was only marginally higher than maximum values observed during the MALINA expedition. The above method also assumes a negligible contribution by CDOM to $c_\mathrm{p}$ at 660 nm (Bricaud et al., 1981), which is a reasonable assumption based on data shown in Matsuoka et al. (2012). $V_\mathrm{dark}$ was found to be 0.0517 V when measured immediately after a deep cast when the temperature of the instrument was equilibrated to seawater temperature. The factory supplied value was 0.061 V. However, discrepancies in $V_\mathrm{dark}$ are of little significance compared to $V_\mathrm{ref}$. For example, for relatively turbid conditions with $V_\mathrm{signal}$ as

low as 3.7 V (representing a $c_p(660)$ of 1 m$^{-1}$), the change from 0.0517 to 0.061, reduce the calculated $c_p(660)$ by only 0.2 %.

In this study we also use the C-Star transmissometer data obtained during CASES (2004) and IPY-CFL (2008) expeditions on the *CCGS Amundsen* (Ingram et al., 2008; Barber et al., 2010) to compare and contrast to the MALINA observations. The data were processed in the same way as the MALINA 2009 downcast data. One exception was that the factory supplied $V_{dark}$ values were used exclusively as they had not been determined onboard the vessels. The $V_{dark}$ values were 0.0570 V and 0.0586 V for the CASES and IPY-CFL expeditions, respectively. The highest $V_{signal}$ readings were 4.6783 V and 4.7902 V, respectively.

Four deep CTD casts were additionally collected in the Canada Basin during the Joint Ocean Ice Study (JOIS) on 21-23 September 2009 and the data were obtained from the Beaufort Gyre Exploration Program website (http://www.whoi.edu/beaufortgyre). These transmissometer data were processed as described above with a $V_{dark}$ value of 0.0633 V (factory calibration) and $V_{ref}$ value of 4.9408 V (maximum recorded value at station CB-21 on 9 October 2009).

## 2.5 Determination of surface water mass distributions

During the MALINA expedition, water samples were collected at 51 stations on the Mackenzie Shelf either by the CTD/Rosette or from the barge. Oxygen isotope ratio ($\delta^{18}O$) were analysed at the Light Stable Isotope Geochemistry Laboratory (GEOTOP-Université du Québec à Montréal) using a triple collector IRMS in dual inlet mode with a precision of $\pm$0.05 ‰. Total alkalinity (*TA*) was measured by open-cell potentiometric titration (TitraLab 865, Radiometer®) with a combined pH electrode (pHC2001, Red Rod®) and diluted HCl (0.03 M) as a titrant. Oxygen isotopes and *TA* collected during CASES 2004 are described, and partially published, in Lansard et al. (2012). We use salinity (*S*), $\delta^{18}O$ and *TA* data to estimate the fractional composition of sea ice meltwater ($f_{SIM}$) and meteoric water ($f_{MW}$) in the surface layer on the Mackenzie Shelf, following the protocol described in Lansard et al. (2012). The calculations follow Yamamoto-Kawai et al. (2008) and Lansard et al. (2012) with the sea ice melt (SIM) end-members 4.7 PSU, $-2.5$ ‰ and 415 $\mu$mol kg$^{-1}$, the meteoric water (MW) end-members 0 PSU, $-19.5$ ‰ and 1620 $\mu$mol kg$^{-1}$, and the saline Pacific Summer Water ($f_{PSW}$) end-members 31.5 PSU, $-3.0$ ‰, 2250 $\mu$mol kg$^{-1}$, for *S*, $\delta^{18}O$ and *TA*, respectively. The Mackenzie River represents the major source of meteoric water on the Mackenzie shelf.

## 2.6 Additional environmental data

To describe ocean currents, temperature, and salinity near the shelfbreak, in addition to CTD casts we used data from a current meter (RCM11, Aanderaa Instruments) moored at station CA05 near the center of Line 100 (Fig. 1). The locations where the mooring CA05 was deployed and the depth of the current meter varied slightly between years. During season 2003–2004, it was deployed in 250 m deep water (71.42° N, 127.37° W) at a depth of 202 m. In 2007–2008 and 2008–2009, the bottom depth was about 200 m (71.31° N, 127.60° W) and the instrument depth 178 m. In addition to current speed and direction, the instrument recorded water temperature, conductivity, turbidity, and dissolved oxygen content, all at 0.5 hour intervals. The conductivity sensor did not function in 2007–2008.

Annual estimates of Mackenzie River discharge and ice concentrations on the Canadian Beaufort Sea shelf for years 2004, 2008, and 2009 were obtained from publicly available data provided by Environment Canada. Daily discharge rates (in units of $m^3 \ s^{-1}$) for the Mackenzie River at the Arctic Red River location (10LC014) were obtained from Water Survey of Canada (Environment Canada) hydrometric data online archives. Ice coverage with a 1-week resolution for the Mackenzie Shelf area was calculated using the IceGraph 2.0 program (region: cwa01_02) provided online by the Canadian Ice Service (Environment Canada).

Estimates of wind speed over the shelf were obtained by averaging 10-m elevation wind data over grid points located over the shelfbreak in the southeastern Beaufort Sea obtained from National Centers for Environmental Prediction (NCEP) (Fig. 1). As pointed out by Williams et al. (2006), NCEP data are readily available and may be preferable over observations made at coastal stations because the latter may be affected by the presence of land. We use the NCEP wind data in a qualitative sense to identify conditions that may have induced upwelling or downwelling of seawater within the shelf area (e.g., Kirillov et al., 2016).

## 3 Results and Discussion

### 3.1 Water mass distributions and circulation during August 2009

During the MALINA cruise in August 2009, there was a distinct east-west gradient in the observed surface salinity on the shelf (Fig. 2a). To the west, surface salinities below 24 PSU were caused by the presence of the river plume that flowed along the coast and over the Mackenzie Trough in response to easterly winds during June 2009 (see section 3.3.3). The river plume formed a near-surface layer of about 15–20 m thickness, which covered the full extent of line 600 and line 700. To the east, water with salinity above 29 PSU was observed to reach the surface in the area north of Cape Bathurst. Williams and Carmack (2008) described such upwelling from within the Amundsen Gulf as topographically induced in response to easterly winds. Salinity values in excess of 32 PSU were measured near the shelf bottom at 30 m (Fig. 2c), which correspond to Pacific Waters in Amundsen Gulf at a depth of about 80 m (Fig. 2e). Generally, for the Arctic Ocean, salinity controls the vertical stratification such that higher salinity is found at greater depth. The water mass definition that ensues follow Carmack et al. (1989) and are consistent with descriptions in Lansard et al. (2012) and Matsuoka et al. (2012). The salinity range between 30.7 and 32.3 PSU corresponds to the Pacific Summer Water mass, which originates from waters flowing through Bering Strait during summer. Underneath, the Pacific Winter Water is characterized by salinity between 32.3 and 33.9 PSU and typically found from $\sim$180 to 220 m depth (e.g., Jackson et al., 2015). This is followed by a transition to waters of Atlantic-origin with salinity > 34.7 and temperature above 0 °C typically found between $\sim$220 and 800 m. Cold and dense deep water are found at greater depths and down to the bottom.

The relative contributions (%) of the two sources to the freshwater content, i.e., meteoric water $f_{MW}$ and sea ice meltwater $f_{SIM}$, in the surface layer is shown by the contours in Fig. 2a. The percent values are calculated as follows: $f_{MW}/(f_{MW} + f_{SIM}) \times 100$. Apart from the Mackenzie River mouth, the freshwater in the surface layer was a mixture between sea ice melt and river runoff. River water prevailed along the coastline, while sea ice melt had a larger contribution further offshore. A

larger river water fraction also extended further along the west coast with the northwest flowing river plume. In the upwelling region north of Cape Bathurst, river runoff and ice melt contributed about equal amounts to the relatively small freshwater content of ∼10 %. The high ice melt proportions in excess of 80 % were found in offshore waters with melting multiyear sea ice (Bélanger et al., 2013).

5     Geostrophic currents for the cross-shelf sections 100, 200, 300, and 600 were calculated using temperature and salinity data from August 2009 CTD casts (Fig. 3). The reference depth, where the current velocity was assumed to be zero, was selected as 500 m, corresponding to a water mass originating in the Atlantic in which geopotential gradients are small (McPhee, 2013). The sections reveal a westward mean flow of up to 9 cm s$^{-1}$ in the Canada Basin (Fig. 3b, c), which is consistent with the anticyclonic circulation of the Beaufort Gyre. Similarly, currents over the shelf were typically westward with speeds on the order of a few centimeters per second. A notable feature was the presence of the eastward flowing shelfbreak current centered between 100 and 150 m depth (Pickart, 2004). The shelfbreak current is an indicator for downwelling flow from the shelf to the basin (Dmitrenko et al., 2016). Both Dmitrenko et al. (2016) and Forest et al. (2015) present mooring data collected at Mackenzie Shelf shelfbreak location showing events of wind-driven shelfbreak current intensifications (with flow up to 1.2 m s$^{-1}$ in January 2005) during downwelling favorable winds. However, to our knowledge, the current intensification along the Mackenzie Shelf shelfbreak during summer has not been shown in the literature to date. The mean easterly flow was around 3 cm s$^{-1}$ (Fig. 3a–c), which is consistent with the observations of Pickart (2004) for the summertime period along the Alaskan Beaufort shelfbreak. The section along line 600 in the Mackenzie Trough captured an anticyclonic mesoscale eddy (∼50 km diameter) which impacted the patterns of $c_p(660)$ and chl-$a$ fluorescence (see below).

## 3.2   Characteristics of particles suspended in seawater in August 2009

20   Empirical relationships between the beam attenuation coefficient and SPM are dependent on the composition and size distribution of particle assemblages (Kitchen et al., 1982; Bunt et al., 1999; Babin et al., 2003b; Reynolds et al., 2010; Woźniak et al., 2010; Hill et al., 2011). In this section we present several water characteristics encountered in August 2009 that help understand the origin of suspended particles and composition of particle assemblages in the Canadian Beaufort Sea. The absorption associated with organic and inorganic material is described elsewhere (Doxaran et al., 2012; Bélanger et al., 2013). However, the measured particulate absorption at 660 nm was found to be smaller by 1–4 orders of magnitude than $c_p(660)$ and can thus be ignored (data not shown). Particle size distributions during MALINA and the relationship to backscattering are described in Reynolds et al. (2016). The environmental conditions encountered during MALINA showed large spatial variability; yet, a statistically significant and strong correlation was found between the particulate beam attenuation coefficient ($c_p(660)$) and SPM, as well as POC (see section 3.3). Although we recognize the possibility of interannual and seasonal variability in particle characteristics, the wide range of particle characteristics observed during the MALINA expedition gives us confidence in the applicability of the derived statistical relationships to infer suspended particle concentration fields on the Mackenzie Shelf and southeastern Beaufort Sea.

     Generally, $c_p(660)$ in the near-surface layer decreased from > 1 m$^{-1}$ in coastal waters to < 0.02 m$^{-1}$ in offshore Canada Basin waters (Fig. 2), reflecting the riverine and coastal sources of particulate matter. To the west, the fresher surface layer

influenced by the river plume featured relatively high $c_p(660)$ ranging from 0.1 to 0.4 m$^{-1}$ (Fig. 2b) and high coloured dissolved organic matter (CDOM) fluorescence (Fig. 4; Matsuoka et al., 2012). The highest ship-based observation of surface-water $c_p(660)$ of ~2.6 m$^{-1}$ was observed at station 394 in 13-m deep waters at the mouth of Kugmallit Bay; however, $c_p(660)$ reached 8.8 m$^{-1}$ at 10 m depth and presumably higher values near the seabed. The surface waters in the area of upwelling just

north of Cape Bathurst appear also to have been a hotspot in terms of particle concentration; $c_p(660)$ at the surface of station 170 reached values over 1.2 m$^{-1}$ (Fig. 2b).

The high $c_p(660)$ values near the shelf seafloor in August 2009 were accompanied by a strong chl-*a* fluorescence signal, both of which also extended from the shelf far into the Canada Basin as a subsurface chl-*a* maximum (SCM) layer (Figs. 4a, c, e). The SCM layer is a consistent feature in the southern Beaufort Sea during summer (Martin et al., 2010). The SCM was

centered at depths between the 31.5 and 32.3 PSU isohalines, which corresponds to the lower portion of the Pacific Summer Water. The underlying Pacific Winter Water is characterized by maxima in both nutrients and CDOM (Fig. 4; Matsuoka et al., 2012). The nutrient maximum is typically found at the center of the Pacific Winter Water near the 33.1 PSU isohaline (Martin et al., 2010).

Following Woźniak et al. (2010), the data representing discrete seawater samples were partitioned into three composition-

related groups based on the POC/SPM ratio: 1) mineral-dominated when POC/SPM < 6 %, 2) mixed when 6 % ≤ POC/SPM ≤ 25 %, and 3) organic-dominated when POC/SPM > 25 %. Only at station 394 (13 m bottom depth) near the entrance to Kugmallit Bay did the CTD/Rosette sampling from the *CCGS Amundsen* take place sufficiently close to the coast to reach the mineral-dominated water masses (Fig. 5a). However, the results from barge sampling in August 2009 show that mineral-dominated particle composition was mostly limited to shallow waters less than about 20 m deep near the two Mackenzie River

mouths where $f_{MW}$ contributed > 90 % of the freshwater content (Fig. 5a). This agrees with past observations suggesting that most mineral-dominated particles transported by the Mackenzie River plume settle to the bottom within the delta or shortly after reaching the shelf where the plume speed decreases (Macdonald et al., 1998). For the rest of the shelf and basin surface waters the particle composition in our collected samples showed considerable variability within the organic-dominated and mixed types (Fig. 5). The one exception was, however, the surface sample at station 110 located furthest east in the Amundsen

Gulf where the POC/SPM was less than 1.8 % (SPM = 3.56 g m$^{-3}$). Although the possibility of contamination of the sample from station 110 cannot be excluded, the high SPM load could also have been caused by the release of ice-rafted sediments as the ice melted (Bélanger et al., 2013). Deteriorated multiyear ice was observed in the vicinity of the station 110, which could have been the source of minerogenic material. Sea ice meltwater was found to have a slightly larger contribution at station 110 compared to other stations along line 100 (Table 1).

For a detailed description of the particle size distribution (PSD) data measured during MALINA, readers are referred to Reynolds et al. (2016). Here, we provide an overview of the spatial distribution of the PSD by calculating the volume fraction of particles less than 1 $\mu$m in diameter $D$ to the total particle volume between 0.7 $\mu$m and 120 $\mu$m. A notable feature in the particle volume distribution, $V(D)$, was the presence of high concentrations of < 1 $\mu$m volume fractions in surface waters and their reduced abundance in subsurface waters (Fig. 5c). The highest increase in the abundance of submicron particles

relative to larger particles was found in samples collected furthest to the west along lines 600 and 700 where surface water

salinity associated with the river plume was less than 24 PSU. A similar observation also pertains to the surface water sample from station 380 located near the Mackenzie River's Kugmallit Bay channel, even though the salinity was ∼28 PSU (Fig. 5c). However, the fraction of meteoric water was similar to station 620 (Fig. 5d). The PSD measurements for low salinity, highly turbid samples nearest to the river mouth (stations 390, 394, and 690) were not possible due to limitations of the Coulter technique. Station 110 stands out among line 100 stations with < 1 $\mu$m volume fractions of 0.29 at the surface (salinity of 29.1 PSU) and 0.09 at 60 m depth (31.6 PSU).

To conclude, from the data in Fig. 5 we find that (1) when $f_{MW}$ increased in the surface waters of southeast Beaufort Sea, POC/SPM ratios decreased while the < 1 $\mu$m particle fraction increased, and conversely (2) when the $f_{SIM}$ influence increased, POC/SPM increased while the < 1 $\mu$m particle fraction decreased in surface waters.

## 3.3 Relationships between SPM, POC and particulate beam attenuation

The SPM of the samples examined during the MALINA cruise ranged from 0.04 to 140 g m$^{-3}$ with associated POC from 0.007 to 1.5 g m$^{-3}$ (Doxaran et al., 2012). Organic-dominated and mixed particle assemblages were predominant in the portion of the data set obtained from ship-based sampling, with SPM extending to 5.6 g m$^{-3}$. The mineral-rich particle assemblages were more common in turbid estuarine waters located close to shore (Fig. 5a). These waters were sampled using a small barge with an optical package that included a Wetlabs AC-9 meter (Doxaran et al., 2012), but no Wetlabs C-Star 660-nm. The nearest wavelength band on the AC-9 was 676 nm. It thus provided $c_p(676)$. Note that much higher sediment loads were observed in the region in the past. For example, Carmack and Macdonald (2002, their Fig. 10) reported on near bottom SPM values of 3000 g m$^{-3}$ due to resuspension of bottom sediments during a storm in September 1987.

Data from all 28 stations with coincident measurements were used in the development of relationships between $c_p(660)$ and SPM and between $c_p(660)$ and POC. The particulate beam attenuation coefficient correlated well with both SPM and POC (Fig. 6a, b).

$$\mathrm{SPM} = 1.933\, c_p(660)^{0.9364} \tag{2}$$

and

$$\mathrm{POC} = 0.2071\, c_p(660)^{0.6842} \tag{3}$$

where SPM and POC are in [g m$^{-3}$] and $c_p(660)$ in [m$^{-1}$], with $r^2$ of 0.71 and 0.74, respectively. Further details on the evaluation of the regression fits are provided in the Supplementary Material. In some instances, for example in biogeochemical modelling studies, the objective may be to estimate light transmission from SPM or POC that has either been measured or is available as model output. For this case the best-fit regression functions are $c_p(660) = 0.4267\, \mathrm{SPM}^{0.9068}$ and $c_p(660) = 3.088\, \mathrm{POC}^{1.098}$, respectively.

The slopes of the best fit lines (with intercepts set to zero) obtained through linear fitting to all pairs of $c_p(660)$ vs. SPM and $c_p(660)$ vs. POC data were 0.404 m$^2$ g$^{-1}$ ($r^2 = 0.70$) and 3.39 m$^2$ g$^{-1}$ ($r^2 = 0.72$), respectively. These slope values represent average SPM-specific and POC-specific particulate beam attenuation coefficients, respectively, for the examined data set. Our

average SPM-specific particulate beam attenuation coefficient at 660 nm is consistent with the range 0.2–0.6 m$^2$ g$^{-1}$ reported by Boss et al. (2009) and Hill et al. (2011) for a 12-m deep coastal site in the North Atlantic Ocean (Martha's Vineyard, MA, USA). Our average POC-specific value is near the middle of the range from 2.31 m$^2$ g$^{-1}$ at $c_\mathrm{p}(660)$ = 0.45 m$^{-1}$ to 4.10 m$^2$ g$^{-1}$ at $c_\mathrm{p}(660)$ = 0.07 m$^{-1}$ observed by Stramska and Stramski (2005) in the north polar Atlantic. Jackson et al. (2010)

reported beam attenuation vs. SPM and POC correlations for measurements in the Arctic Ocean in 2006–2007, from which we estimate SPM-specific values of 0.34-0.50 m$^2$ g$^{-1}$ and POC-specific values of 3.4–3.7 m$^2$ g$^{-1}$ for the $c_\mathrm{p}(660)$ range from 0.07 to 0.45 m$^{-1}$, respectively. The slopes calculated from our data within this same $c_\mathrm{p}(660)$ range were 0.46 m$^2$ g$^{-1}$ ($r^2$ = 0.57) for $c_\mathrm{p}(660)$ vs. SPM and 2.47 m$^2$ g$^{-1}$ ($r^2$ = 0.69) for $c_\mathrm{p}(660)$ vs. POC, with the latter being consistent with other datasets (e.g., Cetinić et al., 2012) but notably smaller than the Jackson et al. (2010) value.

The data of SPM used in fitting the relationship of SPM vs. $c_\mathrm{p}(660)$ range from about 0.04 g m$^{-3}$ to 5.6 g m$^{-3}$ (Fig. 6a). This corresponds to $c_\mathrm{p}(660)$ values up to about 3.1 m$^{-1}$; however, the highest measured $c_\mathrm{p}(660)$ where Wetlabs C-Star measurements were made (but not accompanied by SPM sampling) was 8.8 m$^{-1}$ (at 10 m depth at station 394), which according to Eq. 2 would correspond to SPM of about 14.8 g m$^{-3}$. For the purpose of examining SPM patterns we extend the use of Eq. 2 beyond the maximum measured SPM. A similar non-linear least squares regression analysis that included the

highest observed SPM values and corresponding beam attenuation values measured at 676 nm using a Wetlabs AC-9 resulted in a very good fit and a trend line approximating that of the extrapolation of Eq. 2 (Fig. 6c). This supports the assumption that the estimation of SPM from beam attenuation measurements can be reasonably well extended to cover the broader range of values measured with the Wetlabs C-Star, thus being valid from the very clear open ocean to the highly turbid estuarine waters.

The situation is different for the POC vs. $c_\mathrm{p}(676)$ regression. Coincident observations of POC and $c_\mathrm{p}(676)$ reveal a tendency

of POC to level off at the very high attenuation values (Fig. 6d). These high $c_\mathrm{p}(676)$ values were all observed from the barge in the shallow estuarine waters of the Mackenzie River mouth (Doxaran et al., 2012). As the particle assemblages within these coastal waters are dominated by mineral particles, a weak relationship between POC and $c_p$ is expected. However, within the POC range up to about 0.45 g m$^{-3}$ and $c_\mathrm{p}(660) \leq 3$ m$^{-1}$ covered by ship-based observations (Fig. 6b), which included only organic-dominated and mixed particle assemblages (POC/SPM $\leq$ 25 %), both $c_\mathrm{p}(660)$ and $c_\mathrm{p}(676)$ are well represented by

Eq. 3. This covers the range of $c_\mathrm{p}(660)$ observed along all the ship-based transects (Fig. 1).

### 3.4   SPM distributions on the shelf, slope and beyond

The large range in concentration and composition of suspended particle assemblages (Figs. 5 and 6) collected as a part of the MALINA dataset allowed the determination of empirical relationships for estimating SPM and POC from $c_\mathrm{p}(660)$ (Eqs. 2–3) in Canadian Beaufort Sea waters. In the following, SPM distributions in the Canadian Beaufort Sea are investigated by

applying the SPM algorithm to $c_\mathrm{p}(660)$ data collected during three cruises in the Canadian Beaufort Sea. These cruises include the two year-long projects CASES (2003-2004) and IPY–CFL (2007-2008), and the MALINA project in August 2009, which altogether cover a wide range of conditions encountered during the open water season in Canadian Beaufort Sea. Additionally, $c_\mathrm{p}(660)$ data from four deep casts in Canada Basin collected during the JOIS expedition in September 2009 are examined to show conditions further away from the shelfbreak (Fig. 1).

Here, we focus on the cross-section plots for transect lines 100, 300 and 600 only (Fig. 1). These transect lines have been also repeatedly measured during other field campaigns (e.g., Carmack et al., 1989; Tremblay et al., 2011; Lansard et al., 2012; Mol et al., 2018). Line 100 crosses the Amundsen Gulf near its entrance from north of Cape Bathurst towards the southwestern point of Banks Island. Line 300 is a south-to-north transect located approximately along $134°$ W, and associated with Kugmallit

Valley. Line 600 follows the Mackenzie Trough and provides the western border to the Mackenzie shelf. The Mackenzie River delta is a maze of tributaries; however, the main discharge channel exits at Mackenzie Bay near the end of line 600, while the second largest channel exits at Kugmallit Bay near line 300.

Figure 7 shows the SPM fields from the three expeditions, derived from $c_{\mathrm{p}}(660)$ profiles using Eq. 2. Figure 8 provides the supporting temperature and salinity fields. Black contour lines show SPM values up to 10 g m$^{-3}$ (Fig. 7f). We recall from

section 3.3 that both Eq. 2 and Eq. 3 are derived from ship-data and are strictly valid for $c_{\mathrm{p}}(660)$ values up to 3.1 m$^{-1}$ (Fig. 6). Thus, this excludes the most mineral-dominated waters on the shelf with SPM over 5.6 g m$^{-3}$ and POC over about 0.5 g m$^{-3}$. However, comparisons against near-shore data collected with the barge indicates that Eq. 2 for SPM is reasonably valid for a wider range (Fig. 6c). This is not the case for POC. Within the valid range ($c_{\mathrm{p}}(660) < 3.1$ m$^{-1}$) the presented SPM [g m$^{-3}$] fields can be converted to POC [g m$^{-3}$] according to POC $= 0.1279$ SPM$^{0.7307}$, which is derived from the regression analysis

of POC vs. SPM data.

Elevated SPM values were generally present in shelf surface waters, and associated with a lower salinity surface layer or plume. Highest values were seen nearest to the shore in shallow waters, indicating the riverine origin. SPM in the surface layer decreased past the shelfbreak often reaching very low values, except within the northwest flowing Mackenzie River plume during the 2004 CASES and 2009 MALINA expeditions (Figs. 7g, h). Clear waters with SPM ranging between 0.04 and 0.06

g m$^{-3}$ were found offshore on line 300 in each of the three expeditions (Figs. 7d–f ). The corresponding POC ranged from 0.01 to 0.02 g m$^{-3}$. The low SPM values were especially widespread in August 2009 likely related to the high $f_{SIM}$ content (Table 1).

Wedges of very clear water are seen extending far onto the shelf particularly during 2009. The extension of clear waters onto the shelf as a wedge between the surface plume and the turbid near bottom layer has been described by Carmack et al. (1989).

It appears that neither particle settling from the surface plume nor the upward mixing of bottom sediments were sufficient in August 2009 to increase these clear-water values of $c_{\mathrm{p}}(660)$ above those found in deep basin surface waters. The landward extension of the clear-water layer was particularly noticeable on line 600 (Fig. 7h) which corresponds to the Mackenzie Trough, the main river channel and the most distinct surface plume feature of the transects.

### 3.4.1 Subsurface nepheloid layers

Figure 7 reveals a ubiquitous presence of intermediate nepheloid layers (INL) extending from the Beaufort Sea continental slope. These INLs are produced primarily by resuspension of bottom sediments previously settled onto the shelf or slope, and provide clear evidence for the lateral transport of suspended particles, and water, away from the shelf. Numerous INLs are seen in the upper 500 m of the water column throughout the Amundsen Gulf and extending into Canada Basin, however, the variability in their depth locations is large between the profiles revealing the complex SPM dynamics of the region (Fig. 9).

Generally, the SPM of INLs in offshore waters was an order of magnitude smaller than in the benthic nepheloid layer (BNL) on shelf and particle concentrations decreased with distance from the shelf.

In the Mackenzie Trough (line 600), in addition to the surface river plume, two notable INLs were observed in 2004 and 2009 to extend from the shelf at depths of 100–130 m and 200–250 m (Fig. 7g, h). These two INLs formed near where the 33.1 PSU isohalines intersected the shelf seafloor and immediately above and below a slightly less sloping section of the Mackenzie Trough bottom. Only the upper INL was accompanied by relatively high chl-*a* fluorescence (Fig. 4e). However, the depths of 100 m and greater are beneath the euphotic layer rendering *in situ* primary production negligible. Thus, these chl-*a* containing particles likely represent laterally transported particles that originated from resuspension in shallower shelf waters.

It is important to differentiate subsurface nepheloid layers from the mainly locally formed subsurface chl-*a* maximum (SCM) layer that is commonly present in the Canadian Beaufort Sea at depths between the 31.5 and 32.3 PSU isohalines (Martin et al., 2010; Tremblay et al., 2011). As the SCM seems to intersect with the shelf bottom before extending into the Canada Basin (Fig. 4), the presence of relatively high chl-*a* concentrations within subsurface nepheloid layers may however conceal the presence of minerogenic particles at the same depth. As suggested by Tremblay et al. (2011), the patterns of salinity, $c_p(660)$ and chl-*a* fluorescence indicate that biological production on the shelf bottom was enhanced by upwelled nutrient-rich waters and, at the time of our measurements, biogenic material was being transported seaward in a BNL, and then INL, across the shelfbreak at 50–70 m depth (Figs. 4 and 7c, f, g). The shelf circulation at play makes it conceivable that the transport of biogenic material produced on the shelf, including resuspension of settled particles originating from an earlier bloom (e.g. ice algae), could play some role in the formation and maintenance of the SCM in the off-shelf region.

Beneath 500 m depth, the vertical profiles of SPM still showed numerous INLs and the decrease in SPM beneath or above INLs (Fig. 9), which rules out turbulent mixing and suggests lateral advection in the formation of this SPM structure. Generally, however, the particle concentration at specific depths decreased as bottom depth increased as it also relates to the distance from the shelfbreak. This decrease is approximately exponential with distance from the shelfbreak indicating horizontal spreading. In waters less than 3000 m deep located on the continental slope and rise, the SPM began to increase with depth from about the mid depth of the water column which had the clearest waters. The thickness of these BNLs ranged from ~200 m (station 340) to over 1000 m (Fig. 9). Past the 3000 m bottom depth, BNLs were essentially absent with the clearest waters found close to the bottom as may also be the case for the Canada Basin abyssal plain (Hunkins et al., 1969). Near-bottom SPM values based on $c_p(660)$ were ~$2\times10^{-3}$ g m$^{-3}$ at the station CB-27, and decreased to ~$1\times10^{-3}$ g m$^{-3}$ at 3500 m at CB-21 (74.0042° N, 139.8699° W, i.e., 113 km north of CB-27) on 9 October. Thus, basin waters agreed with the two types of profiles described in Hunkins et al. (1969), first, in waters with bottom depths less than about 3000 m the SPM had minimum values roughly at mid-depths of the water column and then increased towards the bottom forming a c-shaped profile, and second, in waters exceeding the 3000 m depth the SPM reached minimum values near the bottom.

A notable INL at stations CB-23, CB-27, and CB-21 was spreading in the layer immediately below the isopycnal surface where the potential density anomaly $\sigma_\theta$ reached 28.096 kg m$^{-3}$ or the salinity reached 34.956 (Fig. 9). This was the deepest INL (below which no INLs were observed) extending to the Canada Basin abyssal plain at the top of the adiabatic Canada Basin bottom water layer at ~2500–2700 m depth (Timmermans et al., 2003). The depth where the INL occurred varied between the

stations. The maximum SPM within the INL at 2470 m depth at station CB-23 was 0.0126 g m$^{-3}$. At CB-27 the maximum was $8.2 \times 10^{-3}$ g m$^{-3}$ at 2600 m (Fig. 9). The SPM levels above the INLs (with $\sigma_\theta$ = 28.095) were 0.010 and 0.027 g m$^{-3}$, respectively. Given that the INL depth increased by 130 m over the 128 km distance that separated the two stations, the INL descent rate was about 1 m km$^{-1}$. A thinner (50 m thick) and weaker INL with a maximum SPM of $3.2 \times 10^{-3}$ g m$^{-3}$ at 2656

m was observed at CB-21 (Fig. 9d). Beneath this interface the potential temperature was uniform with depth, thereby marking a transition to the adiabatic Canada Basin bottom water layer (e.g., Timmermans et al., 2003). Assuming that the particles in the INL were from the bottom layer of CB-31 ( 1920 m depth with $\sigma_\theta$ = 28.093 kg m$^{-3}$), then the transport of particles from the bottom of station CB-31 to the INL at station CB-23 requires a 560 m increase in depth over a 100 km distance. This assumption of a northwestward flow direction of particles may not be valid, because the deep circulation in Canada Basin is

thought to be a cyclonic gyre that follows along bathymetric contours (e.g., Holland et al., 1996). Nevertheless, such transport of particles crosses isopycnal surfaces, suggesting ongoing particle settling in addition to advective transport.

### 3.5 Environmental forcing and oceanographic conditions

As is evident, SPM is not a conservative property of a water mass, but undergoes settling or resuspension at rates that are dependent on particle composition and size, and water dynamics. Consequently, in this section, the environmental forcing and

oceanographic conditions during each of the three expeditions are first described and contrasted. Then, in the next section 3.6, the observed patterns of the SPM fields are compared and discussed in the context of these oceanographic conditions, and in particular as these patterns relate to river runoff, sea ice melt, and wind.

### 3.5.1 River discharge and sea ice conditions

The Mackenzie River discharge has large seasonal and interannual variability (e.g., McClelland et al., 2012). Similarly, sea

ice concentration on the shelf undergoes large variability (Galley et al., 2008). This is also evident when comparing daily Mackenzie River discharge rates and ice concentrations on the shelf for years 2004, 2008 and 2009 (Fig. 10). Although the seasonal trend follows a predictable overall pattern, discharge rates during the open water season show significant day-to-day variation, while the timing of landfast ice break-up, wind forcing, and the large-scale circulation in the Beaufort Sea affect ice concentrations. The three field expeditions were conducted during different times of the annual cycle with noticeable

differences in the Mackenzie River discharge (Fig. 10). The CASES 2004 cross-shelf transects were conducted a few weeks after ice break-up and the freshet. The spring freshet occurred later in 2004 with a sharp peak pulse that reached a higher level than during the other years considered. In 2004, the discharge decreased rapidly after the freshet so that the lowest (of the four years) annually averaged discharge occurred. The condition with the highest discharge rates was encountered during the IPY-CFL 2008 transect cross-section sampling as late as in early July, when ice concentrations on the shelf were unusually low

(around 10 %). In contrast, the MALINA 2009 sampling occurred later in the season (August) with conditions characterized by comparatively high (30 %) sea ice concentrations on the shelf.

     The buoyant freshwater released from the melting sea ice competed for surface space with river water, thus affecting plume dynamics and the ability of the plume to keep particulate matter in suspension. As was also the case during CASES in June–July

2004 (Lansard et al., 2012), the freshwater composition in the surface layer on the Mackenzie Shelf during MALINA was a mixture between river runoff (meteoric water) and sea ice meltwater (Fig. 2a). Table 1 provides information on surface salinity and the contribution of freshwater sources measured at the same geographical locations during both CASES and MALINA. Compared to MALINA, river runoff during CASES resulted in lower surface salinity and contributed to a much larger fraction

of the freshwater in the southern half of the Mackenzie Shelf. The one station 320 located past the shelfbreak, however, indicates fresher conditions during MALINA due to a higher sea ice meltwater contribution. In contrast to the river waters, sea ice meltwater typically contains little particulate matter and CDOM (e.g., compare Fig. 4b, d, f). However, significant near-surface particle enrichment was observed, which was associated with meltwater originating from multi-year ice (Bélanger et al., 2013). During MALINA, numerous multi-year ice floes had drifted into the southeastern Beaufort Sea where they were

melting in place (Figs. S1 and S4 in Supplementary Material).

### 3.5.2 Wind forcing

The large freshwater inputs to the Mackenzie Shelf during summer result in strong vertical stratification and a vertically sheared two-layer circulation (Fig. 3) (Carmack and Macdonald, 2002; Carmack and Chapman, 2003; Mol et al., 2018). This estuarine circulation is reflected in the patterns of SPM across the shelf (Fig. 7). Sustained easterly along-shelf winds, particularly when

strong, are known to cause offshore Ekman transport of shelf surface waters, thereby generating upwelling of deeper nutrient rich water of Pacific-origin onto the shelf (Carmack and Kulikov, 1998; Williams et al., 2006, 2008; Yang, 2009). The high salinity observed during the MALINA expedition in Kugmallit Valley (line 300), Mackenzie Trough (line 600) and near the coast west of 140° W indicated the occurrence of upwelling (Fig. 2). During westerly winds, onshore Ekman transport will generate downwelling flow on the shelf (Dmitrenko et al., 2016). During westerly or weak winds, the river plume turns right to

flow eastward along the coast of Tuktoyaktuk Peninsula (Carmack and Macdonald, 2002). Relaxation or reversal of either of these winds will cause return flow to occur towards or from the shelf. Furthermore, strong winds, and brine released from ice formation during late fall and winter, promote vertical mixing and may mix shallow shelf waters to the bottom (e.g., Carmack and Macdonald, 2002; Forest et al., 2007).

The wind vectors reveal a predominance of easterly winds during our study periods in 2004, and 2008–2009, with often

a southward component resulting in along-shelf wind component (Fig. 11a). The predominance of easterly winds is also a driving force behind the large-scale anticyclonic circulation of the Beaufort Gyre and its ice cover. The occasional reversals of the Beaufort Gyre are related to transient synoptic weather patterns (Asplin et al., 2012) that also affect the circulation on the shelf. Two notable periods dominated by westerly winds occurred during the month of October 2008, and during December 2008 to the end of January 2009. Typically the westerly wind episodes were characterized by relatively low wind speeds.

The wind conditions prior to the ship-based expeditions (marked by blue circles) are shown in Fig. 11a. During June–July 2004 (CASES) the wind speed ranged from 2 to 8 m s$^{-1}$ with a variable direction. IPY-CFL sampling (late June and early July 2008) overlapped with CASES in terms of time of year; however, winds were notably different with a month of easterly winds prior to the sampling. The conditions leading up to the MALINA expedition in August 2009 are characterized by <10 m s$^{-1}$ winds at directions inducing the upwelling in June and most of July, but with a turn to northerly winds during the first part of

July, which probably were a contributing factor keeping sea ice on the shelf. Winds turned to easterly (upwelling inducing) for the last week of July with wind speed $> 8$ m s$^{-1}$. Winds during the MALINA expedition were comparatively weak ($< 6$ m s$^{-1}$) with variable direction.

### 3.5.3 Evidence of upwelling and relaxation

Current speeds and directions were measured at 178 m depth on the CA05 mooring in 2008–2009 (and at 250 m in 2003–2004) (Fig. 11b). This depth corresponded to the location of the base of the eastward flowing shelfbreak current (Fig. 3a). The currents at this depth on the slope were found to have two distinct modes: (i) along-shelf current that followed the isobaths towards southwest (i.e., $\sim140°$), and (ii) northeastward cross-shelf current ($\sim300°$). Interestingly, the shift between the two modes was very brief occurring within only a few hours.

The long periods (i.e., from October to November 2008, from December 2008 to April 2009, and from May to June 2009) of along-shelf southwestward currents at the 178 m depth at mooring CA05 (Fig. 11b) were related to periods with either weak winds or westerly winds (Fig. 11a). Episodes with cross-shelf currents occurred on five occasions in the period between August 2008 and October 2009 (Fig. 11b). In addition, a brief period of change in direction occurred during late July and the first few days of August 2009, likely associated with the change in wind direction to easterly during the last week of July. The time series collected during 2003–2004 show only a minor cross-shelf flowing event around the beginning of November. Each episode with cross-shelf currents, with the exception of November 2003 (the location of the moored instrument was deeper and further east compared to 2008–2009), was associated with increases in salinity, temperature, or both, which is an indication of upwelling. All of these events are directly linked to periods with strong easterly along-shelf winds (Fig. 11a) highlighting the likely role of the wind in forcing upwelling. During 2009, the salinity reached up to 34.5 PSU (Fig. 11d), which corresponds to an "effective depth" (see Fig. 3 in Carmack and Kulikov, 1998) of about 300 m indicating a vertical displacement of $\sim120$ m compared to a representative offshore location. Note, however, that the recorded salinity rarely decreased below 33.5 PSU, which in itself corresponds to an "effective depth" of more than 200 m. After the abrupt termination of each upwelling event, temperature and salinity decreased towards pre-upwelling values. Some of the lowest salinity values at 178 m were encountered at the time of the MALINA expedition during August 2009, and associated with downwelling return flow on the Mackenzie Shelf (Fig. 11d).

Episodes of high along-shelf current speeds (dark green in Fig. 11c), such as at the end of the MALINA expedition in late August 2009, but also in November 2008, February, May and July 2009, were generally associated with reductions in salinity and temperature at the CA05 mooring, and perhaps also linked to shelfbreak transport of SPM with downwelling flow.

## 3.6 Effects of river runoff and sea ice melt on SPM distributions on the shelf

### 3.6.1 River plume variability

Wind-forcing largely controls the flow direction of the Mackenzie River plume. Due to the size and shape of the Mackenzie Shelf, the most likely direction for the Mackenzie River plume to spread significant distances past the shelfbreak is to the

northwest (Doxaran et al., 2012). During the spring freshet in June 2009, sustained easterly along-shelf winds caused the flaw-lead polynya to widen along the Mackenzie Shelf and a turbid river plume extended northwestward from the landfast ice to the pack ice (Fig. S2). The MALINA sampling occurred during a time of transition from a northwestward plume (during easterly winds) towards a Coriolis-forced right turning plume flowing eastward along the coast. Plumes of both directions are visible in MODIS satellite images for the period of the MALINA expedition (Doxaran et al., 2012; Forest et al., 2013). By 26 July 2009, the plume was clearly seen extending out past the tip of Cape Bathurst. The sampling along lines 600 and 700 was conducted during the first half of August 2009, following a two-week period of easterly winds (Fig. 11a). By mid-August only very weak features remained from the northwestward plume. Notably, both river discharge and ice concentrations on the shelf were reduced by half during the period of one month (Fig. 10).

Figures 7g, h and 8g, h show the river plume extending northwest along the Mackenzie Trough (line 600). The Mackenzie River plume occupied an about 15 m thick layer at the sea surface both in July 2004 and August 2009. A sharp decrease in SPM was found immediately below this layer. The surface plumes had low salinity, high meteoric water fractions (Table 1 and Fig. 2a), and high CDOM fluorescence (Fig. 4f), at least in 2009, and a high < 1 $\mu$m particle volume fraction (Fig. 5c, d), indicating a riverine origin. Interestingly, particle concentrations differed markedly for the two years compared. In 2004, high levels of SPM extended the full length of the transect with values reaching 4 g m$^{-3}$ as far as 70° N. In contrast, in 2009 the SPM values observed in the plume were only about 10 % of the 2004 values but still distinctly noticeable because the plume overlaid a layer of very clear water. Also, the waters beneath the river plume in 2004 were significantly more turbid compared to 2009, probably due to settling of particles from the plume.

Although the timing of the transect measurements in 2004 and 2009 was a month apart, overall conditions on the shelf were not markedly different. Easterly winds were weak in both cases (Fig. 11), ice coverage on the shelf was 30–40 %, and the river discharge was ~13,000-14,000 m$^3$ s$^{-1}$ during both years (Fig. 10). Moreover, the cross-sections along lines 100 (Fig. 7a, c) and 300 (Fig. 7d, f) show very similar features and particle concentrations during the two years. The differences between the two situations can be attributed to the seasonal timing. The 2004 transects were measured in early July soon after the break-up of the landfast sea ice cover and the surge of backed-up river waters across the delta and estuary. In contrast, the 2009 measurements were conducted much later in the season after landfast ice break-up. Consequently, in 2004 the surface plume was likely conditioned by a greater initial SPM discharge at the river mouth and by a higher momentum compared to 2009 so that it was capable of keeping more particles in suspension for a longer distance, including larger-sized particles if present. MODIS imagery of sea-surface temperature for 2 July 2004 (Fig. S3 in Supplementary Material) highlights this river plume inertia.

### 3.6.2 Surface versus near-bottom cross-shelf SPM distributions

Comparatively high levels of SPM were found along line 300 (Kugmallit Valley) near the shelf bottom in August 2009 with particularly high values extending across the shelf (Fig. 7f). On line 600 (Fig. 7h), a nepheloid layer with SPM > 0.001 g m$^{-3}$ formed near the 33.1 PSU isohaline at ~100 m depth. It was accompanied by a strong chl-*a* fluorescence signal (Fig. 4e). Elevated near-bottom and shelfbreak SPM values were also observed during CASES and IPY-CFL (Fig. 7d, g). Such

SPM patterns are indicative of downwelling return flow from the shelf after upwelling-inducing wind conditions relaxed. The presence of sea ice and its meltwater on the shelf during August 2009, as seen from the low surface temperatures and salinities at ∼70.9° N (Fig. 8f) and high meltwater fractions (Fig. 2a and Table 1), can explain the containment of the spreading of the plume along line 300 (Fig. 7f). High particle settling rates from a slow moving or stagnant river plume may in turn explain the elevated near bottom SPM which then could be transported along the shelf bottom with the return flow of the upwelled waters.

A contrasting situation is provided by the conditions observed along line 300 during June-July 2008 (IPY-CFL study) (Figs. 7e and 8e). During the IPY-CFL, ice coverage on the shelf was low (Fig. 10b) and upwelling-inducing winds prevailed throughout June and early July (Fig. 11a). Consequently, the two compared SPM sections along line 300 differed markedly (Fig. 7e, f). As seen in Fig. 7e, in 2008 the turbid surface river plume spread northward past the shelfbreak. At the same time, the near-bottom turbidity was low likely owing to conditions resulting from upwelling, evidenced by the high salinity of the shelf bottom water and the extent of the surface plume (see Fig. 8e). This offshore surface flow was made possible by the absence of sea ice and ice meltwater (buoyancy forcing) and wind-driven Ekman transport.

The low SPM values were especially widespread in August 2009 (MALINA) with wedges of very clear water extending far onto the shelf. The conditions encountered during MALINA differed from expeditions in previous years particularly in terms of sea ice coverage (Fig. 10b). The break-up of the landfast ice on the shelf occurred relatively late and ice floes were not readily transported away from the shelf due to the northerly and, then later, weak winds (Fig. 11a). Furthermore, multiyear ice extended further south compared to the two other years considered in this study (Fig. S4). At around 70.5° N on line 300, which coincides with northward extent of the river plume and rapid decrease in water column SPM levels (Fig. 7f), the surface salinity decreased below 27 PSU and temperature was <5 °C (Fig. 8f) with over >70 % $f_{SIM}$ fraction of the freshwater (Fig. 2a). As sea surface salinity remained low for the length of line 300 (Fig. 8f), we argue that the meltwater from this ice influenced the low SPM levels in the shelf waters by increasing the stratification, reducing vertical mixing, and hindering the northward spread of the particle-rich river plume.

Another contrasting situation is seen in the Amundsen Gulf along line 100 (Fig. 7a–c) where differences in conditions between the years can be explained by the presence or absence of sea ice, and the history of wind forcing as it relates to SPM transport from the shelf. Whereas ice free and comparatively clear surface waters were present in 2008 (Fig. 7b), turbid (i.e., high $c_p(660)$ and SPM) surface waters extended across Amundsen Gulf in 2004 and 2009 (Figs. 7a, c), and the surface was furthermore partially ice covered in June 2004 (Fig. S4a). The temperature and salinity fields, however, showed only modest differences between conditions in 2004, 2008, and 2009, and the water column remained vertically stratified throughout the transect line (Fig. 8a–c). This suggests that the turbid surface waters in 2004 and 2009 were caused by the advection of shelf waters containing particles that originated from the Mackenzie River and/or from bottom resuspension in shallow shelf waters closer to the shore. This is corroborated by the observed high meteoric water fractions in 2004 and 2009 (Table 1), and the high fraction of < 1 $\mu$m particles in the surface waters in 2009 (Fig. 5d). The equally fresh but clear surface layer in July 2008, after a long period of easterly winds (Fig. 11a) and consequent westward circulation on the shelf (Mol et al., 2018), was however associated with sea ice meltwater with relatively low concentration of particles. The observations that $f_{MW}$ at stations 110 and

140 in July 2008 (IPY-CFL) were of similar magnitude to those observed during CASES and MALINA may be an indication of the importance of resuspension in the supply of SPM to surface water.

Tremblay et al. (2011) discussed the conditions in 2008, as well as nutrient dynamics, leading up to the high primary productivity observed in the Amundsen Gulf during the summer of 2008. The productivity of the SCM is generally proportional to the concentration of chl-*a* and limited by light and nutrient availability (Martin et al., 2010). Tremblay et al. (2011) proposed that the unusually early clearing of sea ice in 2008 was the key factor in increasing the subsurface light availability and primary productivity. However, the influence of the optical water clarity of the surface water layer was not considered. For example, Figs. 7a–c reveal that in July 2008, beneath the low turbidity surface layer, a higher SPM in the SCM centered at the 31.5 PSU isohaline ( 50 m depth) was observed compared to June 2004 and August 2009 when surface water layers were more turbid. Thus, we suggest that the cross-shelf transport of SPM in surface plumes may additionally influence primary productivity in Amundsen Gulf by reducing light penetration.

## 4 Conclusions

The data collected in the southeastern Beaufort Sea during the MALINA in August 2009 enabled the development of relationships for estimating SPM and POC from measurements of optical beam attenuation coefficient. These relationships provided, in turn, a means for obtaining a comprehensive view of particle concentration fields covering the full expanse of the Canadian Beaufort Sea continental margin on the basis of optical data collected during several expeditions in this region. Accompanying water sampling enabled us to conduct a detailed assessment of oceanographic conditions and particle characteristics, including freshwater sources, particle size and composition. Our analysis revealed temporal and spatial variations in particle concentration and dynamics which could be attributed to (i) discharge of the Mackenzie River, (ii) ice coverage and meltwater, and (iii) wind forcing. These three factors control the estuarine-like two-layer circulation on the shelf during summer, and are reflected in cross-shelf SPM patterns that suggest transport occurring mainly within a buoyant surface river plume and the bottom boundary layer. SPM on the shelf exceeded 1 g m$^{-3}$ in each of these cases. A clear water layer was also found at mid-depths on the outer shelf. Similar features were noted by Carmack et al. (1989).

The wind-driven shelfbreak upwelling and downwelling signals were clearly present in the CA05 mooring record for the base of the Pacific Water layer (Fig. 11b) on the continental slope at the mouth of Amundsen Gulf (Fig. 1) at a depth corresponding to an eastward flowing shelfbreak jet (Fig. 3). At 178 m depth, the current was seen to follow isobaths during quiescent and downwelling favorable conditions, but switched to move cross-shelf during upwelling favorable winds (Fig. 11a, b). Interestingly, there appeared to be two very distinct modes of flow at this depth and location along the slope. In 2009, the salinity at 178 m reached 34.5 PSU during the upwelling events (Fig. 11d), which corresponds to an effective depth of about 300 m (Carmack and Kulikov, 1998). However, in all cross-shelf transects shown in Fig. 8, salinities of at least 32.3 PSU were found on the shelf at 60–80 m depth. This salinity corresponded to the transition between Pacific Summer Water and Winter Water, which is typically found at 100 m depth in the Canada Basin (e.g., Carmack et al., 1989). The salinity on the shelf was higher than in corresponding Canada Basin waters at all times and all observed sections. Thus, this modest 20–40 m of

(depth equivalent) upwelling onto the shelf may represent a steady state condition linked to the generally easterly wind and anticyclonic circulation of the Beaufort Gyre.

Freshwater inputs from the Mackenzie River and the melting of sea ice resulted in surface waters being a varying mixture $f_{MW}$, $f_{SIM}$ and $f_{PSW}$, where $PSW$ refers to Pacific Summer Water with a core salinity of 31.5 PSU. We found that the

buoyant sea ice meltwater competed for space with the river plume, and in contrast contained little particulate matter (and CDOM; Fig. 4), which had a significant effect on SPM distributions within the surface layer. When ice meltwater was present on the shelf during years with high ice coverage, it appeared to restrict the expansion of the surface river plume, and cross-shelfbreak transport of particles was consequently found to occur mainly along the shelf bottom in a benthic nepheloid layer (Fig. 7). This was a consequence of two factors: (i) the reduction in plume buoyancy driving force by the sea ice meltwater layer

such that more particles carried by a slower moving or stagnant plume were settled to the bottom, and (ii) weak or westerly winds that allowed sea ice and meltwater to remain on the shelf and to initiate downwelling return flow (after relaxation of wind-induced upwelling) that could transport particles in the bottom boundary layer towards the shelfbreak.

Particle characteristics in surface waters differed considerably depending on the relative contributions of river runoff and sea ice meltwater. Compared to sea ice meltwater, river runoff carried significantly higher SPM loads (Fig. 7), had a particle size

distribution with a higher fraction of submicron particles, a smaller POC to SPM ratio (i.e., more minerogenic particles), and a high CDOM content (Figs. 4–5). These differences have implications on the optical properties of the water, and consequently affect the propagation of sunlight and primary productivity during the open water season.

As the Arctic continues to warm, the open water season is expected to become increasingly longer and the extent of multiyear ice further decline (Stroeve et al., 2014). The reduction in ice coverage in the Beaufort Sea implies an increase in SPM dynamics

on the continental margin due to the associated changes in wind forcing and river discharge (Carmack et al., 2006). Greater wind and wave forcing on open waters is expected to increase particle concentrations on the shelf. However, the presence of both clear intermediate waters and highly turbid bottom waters observed on the shelf in this study highlighted interesting linkages to the effect of sea ice on river water and particle transport on the shelf, which need further study. The processes that operate within subsurface layers and ice-covered waters cannot be deciphered through satellite remote sensing, so their quantification requires

in-situ monitoring. Optical beam transmission is a simple yet efficient tool for mapping SPM distributions. The relationship between SPM and $c_{\mathrm{p}}(660)$ developed in this study can be applied to past and future transmissometer observations to monitor changes in SPM. Vertical measurements reaching all the way to the seafloor would be very beneficial when attempting to determine lateral SPM transport. This is typically not done due to the risk to the instruments. Furthermore, ongoing research that considers current speeds together with particle size distributions are needed in order to shed more light on particle transport

and settling processes across the Beaufort Sea continental shelf and slope, which are experiencing considerable change in response to river discharge, sea ice coverage, and wind forcing. The results from this study can help evaluate numerical models which may be used to investigate sensitivities of SPM dynamics associated with oceanographic and forcing conditions on the Mackenzie Shelf.

*Data availability.* All data used in this study are available in the following online data bases: Polar Data Catalogue, Water Survey of Canada (Environment Canada), Canadian Ice Service (Environment Canada), National Centers for Environmental Prediction (NCEP), and the French national IMBER/SOLAS data base.

*Author contributions.* JKE drafted the manuscript, analysed the data and prepared the figures. JKE and DD collected and analysed the SPM and POC data, while RR conducted the particle size distribution sampling. BL conducted all $\delta^{18}O$ sampling. All coauthors contributed to writing the manuscript.

*Competing interests.* The authors declare that they have no conflict of interest.

*Acknowledgements.* This study was conducted as part of the MALINA Scientific Program funded by ANR (Agence Nationale de la Recherche), INSU-CNRS (Institut National des Sciences de l'Univers – Centre National de la Recherche Scientifique), CNES (Centre National d'Etudes Spatiales), ESA (European Space Agency), and the ArcticNet Canadian Network of Centres of Excellence. Additional funding for JKE, RAR, and DS was provided by the US National Aeronautics and Space Administration (Grant NNX07AR20G). JKE is funded by the Natural Sciences and Engineering Research Council of Canada (NSERC) Discovery Grant program. We thank the National Aeronautics and Space Administration (NASA), National Centers for Environmental Prediction (NCEP), and Environment Canada for providing free access to data. We thank all participants on the CASES, ArcticNet, IPY-CFL and MALINA cruises for their help, and in particular all of the crew members on *CCGS Amundsen*.

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

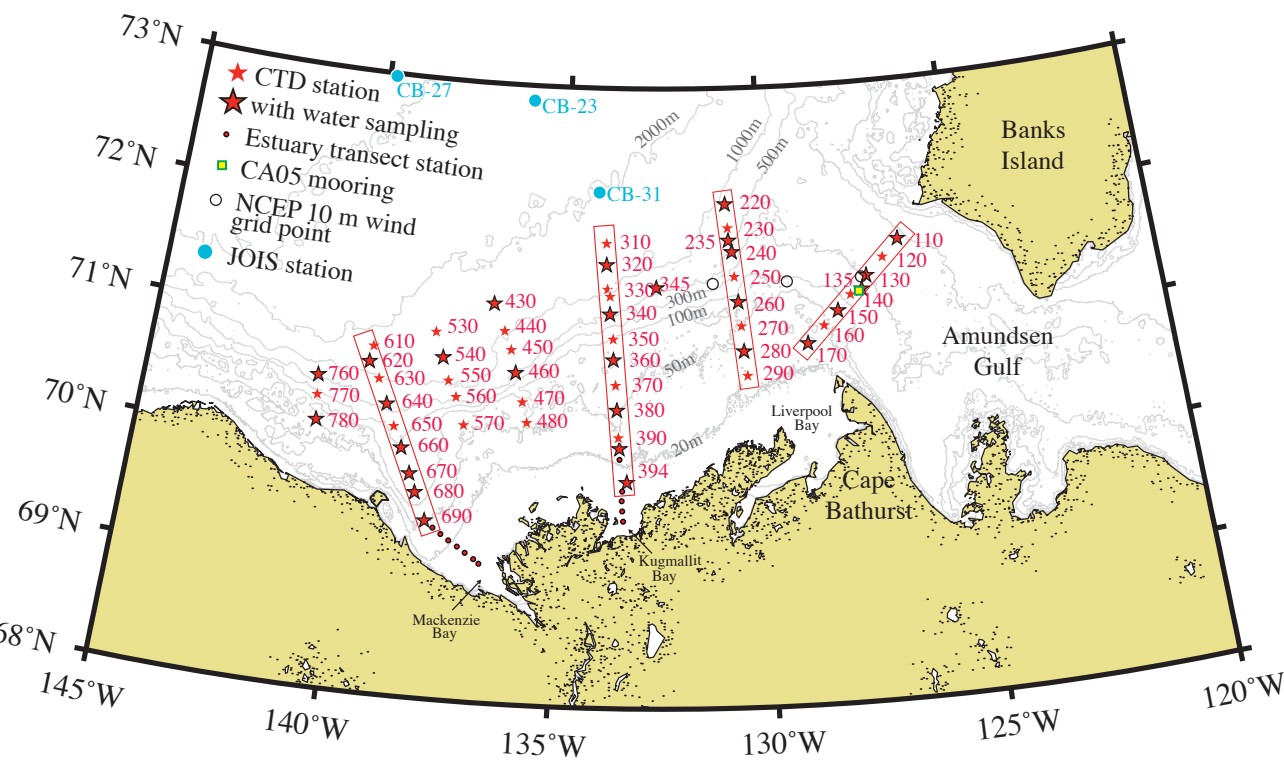

**Figure 1.** Map of study area with stations sampled along transect lines 100 to 700 during the MALINA expedition in 2009. Stars indicate stations visited by *CCGS Amundsen*, and small circles indicate the estuarine stations sampled by the barge. CTD/Rosette water sampling was conducted on the 28 stations marked by stars with black borders. Black circles are the three locations selected for NCEP 10 m winds. The green-yellow square near station 140 indicates the location of the long-term mooring CA05 with a current meter at 178 m. The cyan circles mark the locations for three of the profiles shown in Fig. 9. The fourth station, CB-21, was located 1° north of CB-27.

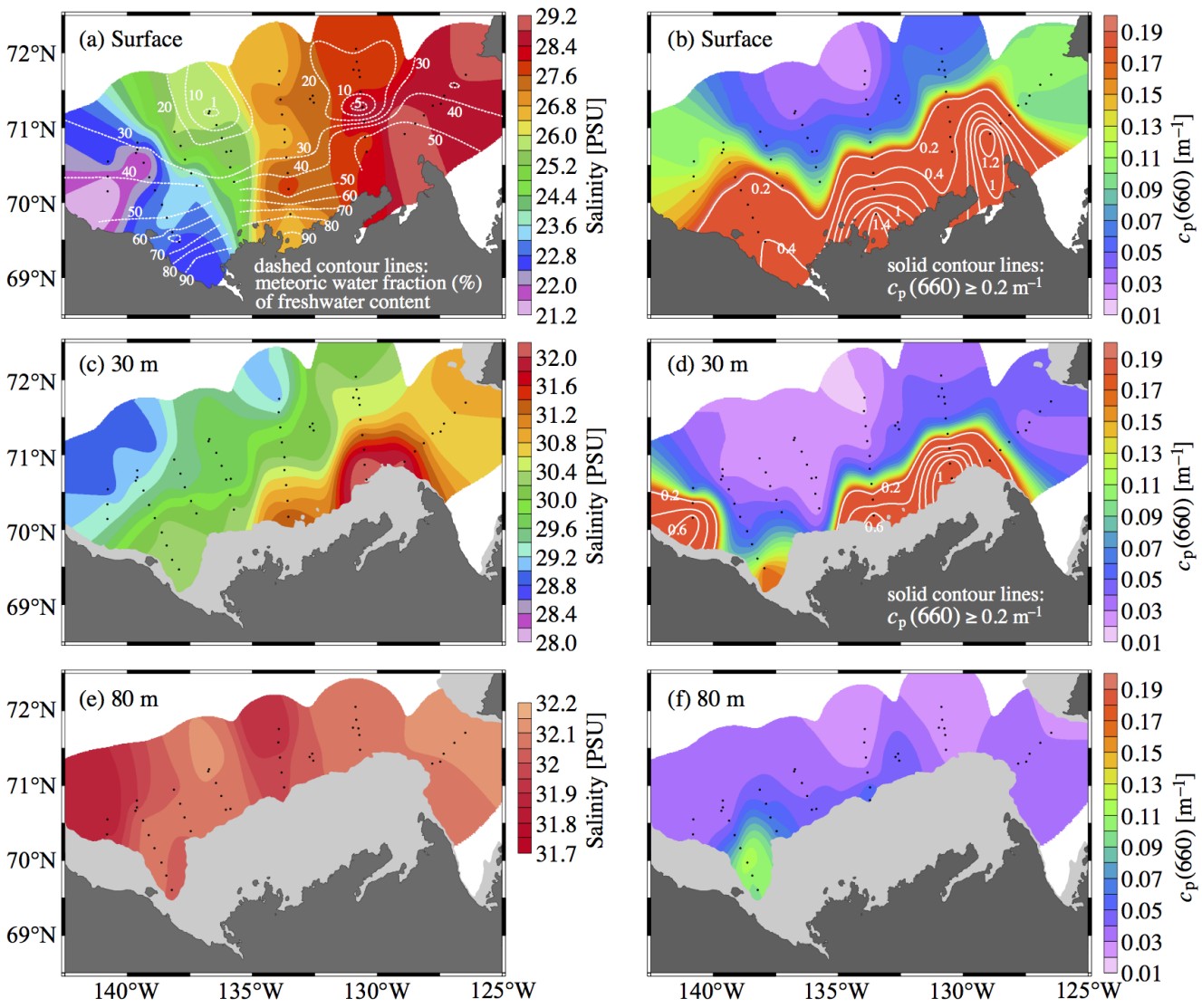

**Figure 2.** Fields of water salinity (left panels) and particulate beam attenuation coefficient at 660 nm, $c_p(660)$, (right panels) for (a–b) sea surface, (c–d) 30 m depth, and (e–f) 80 m during the MALINA 2009 expedition. The presented results include only CTD/Rosette measurements from the *CCGS Amundsen* marked by black dots. Dashed contour lines in (a) are the fraction of meteoric water (%) of the freshwater calculated using samples collected both by barge and CTD/Rosette.

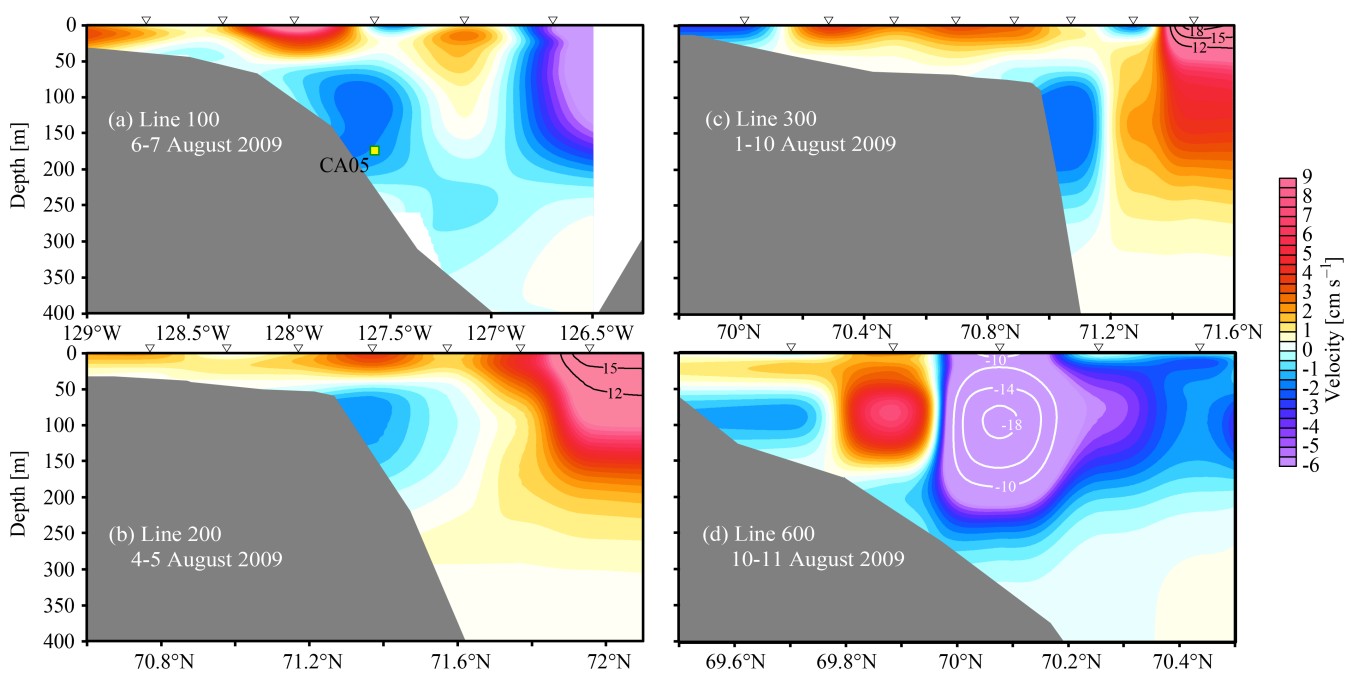

**Figure 3.** Sections of geostrophic current velocity (colours and white contours) perpendicular to transect lines 100 (a), 200 (b), 300 (c), and 600 (d). Geopotential heights were referenced to 500 m. Positive current values are generally for the direction perpendicular to the transect lines (see Figure 1) either towards northwest (a) or west (b–c) or southwest (d). The green-yellow square in (a) indicates the location of the current meter on the CA05 mooring.

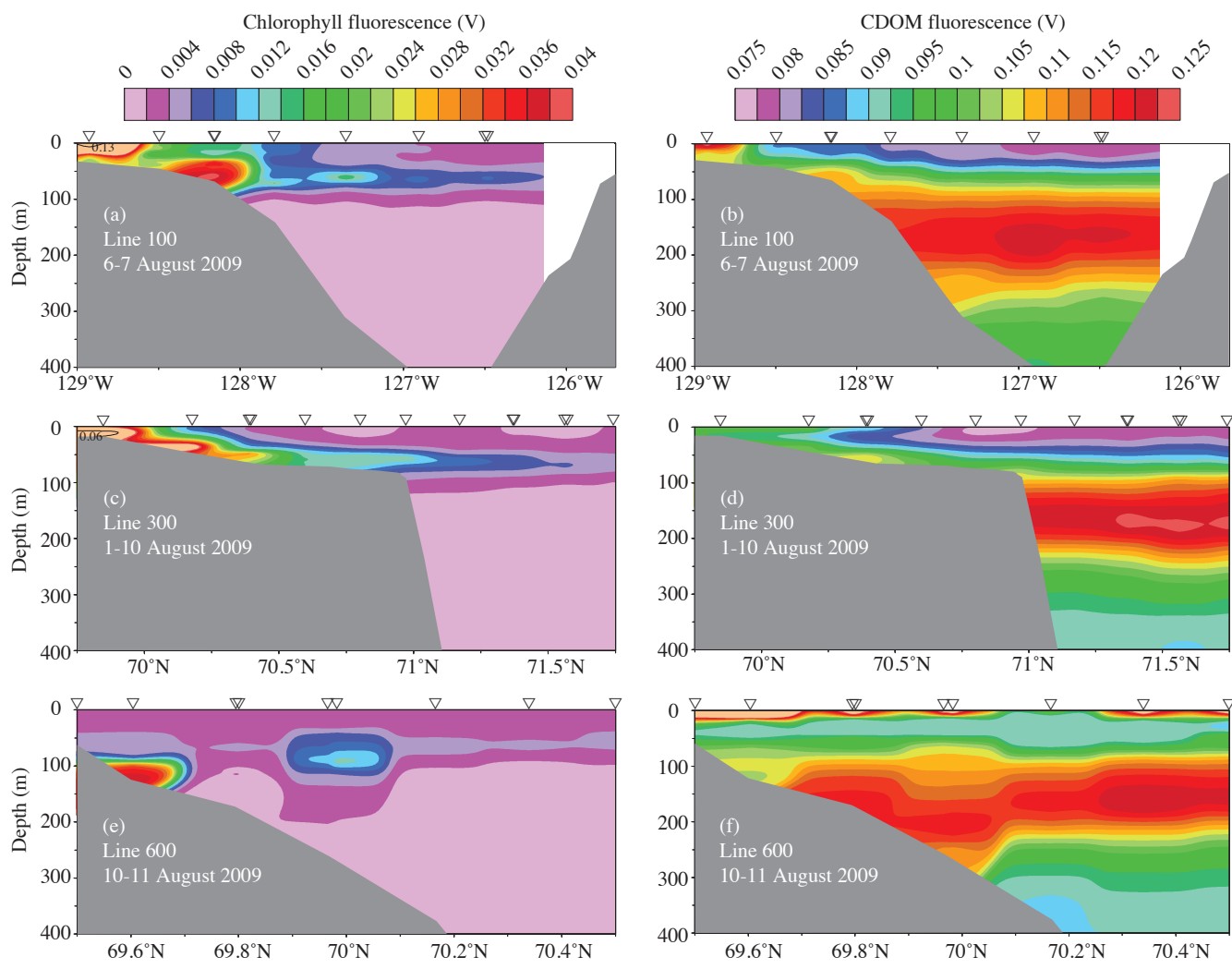

**Figure 4.** Voltage readings from the chlorophyll fluorometer (left panels) and CDOM fluorometer (right panels) for transects (a–b) 100, (c–d) 300 and (e–f) 600.

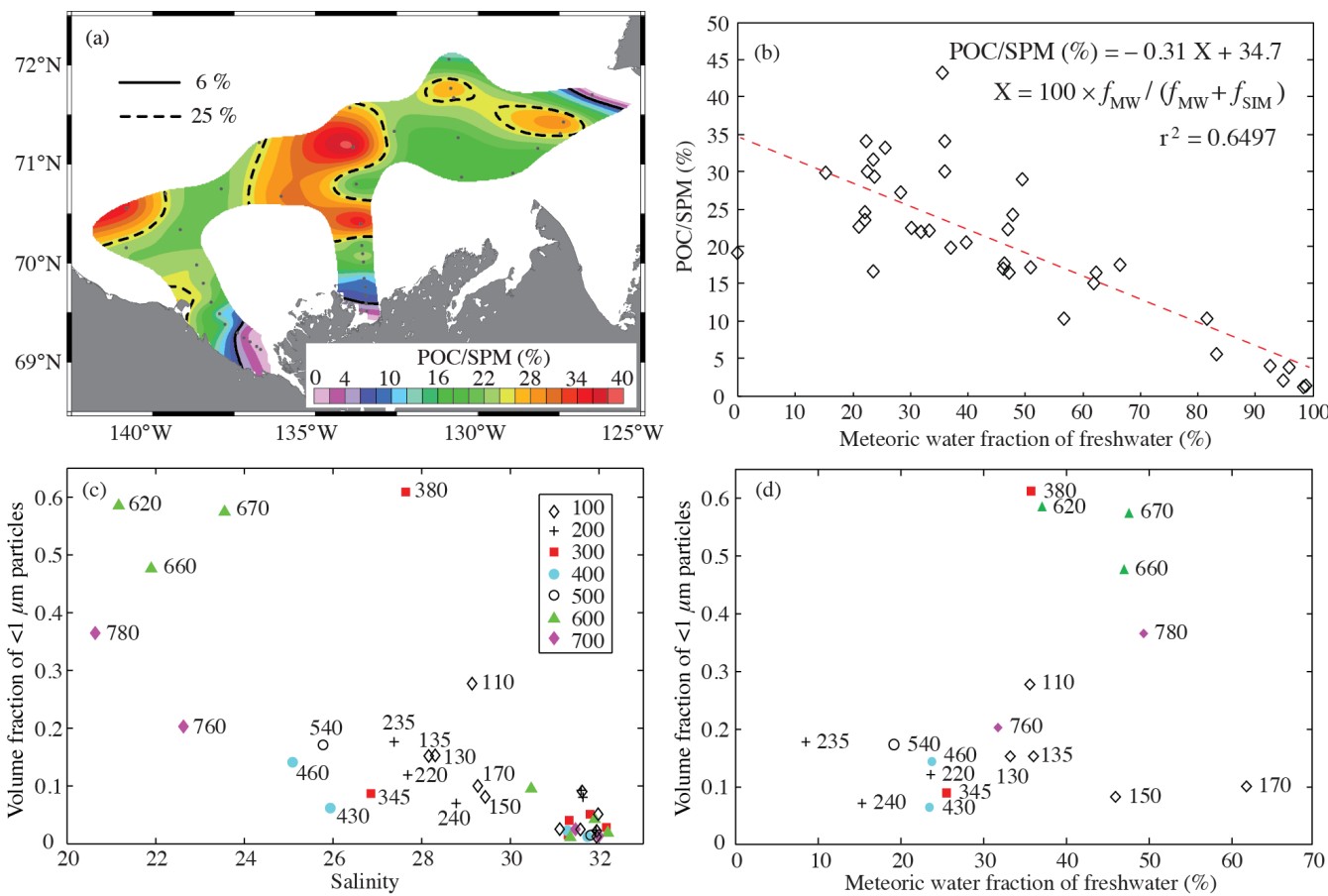

**Figure 5.** (a) POC to SPM ratio for surface samples within the study area, (b) relationship between POC to SPM ratio and meteoric water fraction of freshwater in surface waters (see Fig. 2a), and relationship between volume fraction of particles less than 1 $\mu$m in diameter and (c) salinity, and (d) meteoric water fraction of freshwater. Values in (a), (b) and (d) are limited to surface water samples, while data points in (c) with salinities over 30 PSU represent subsurface samples.

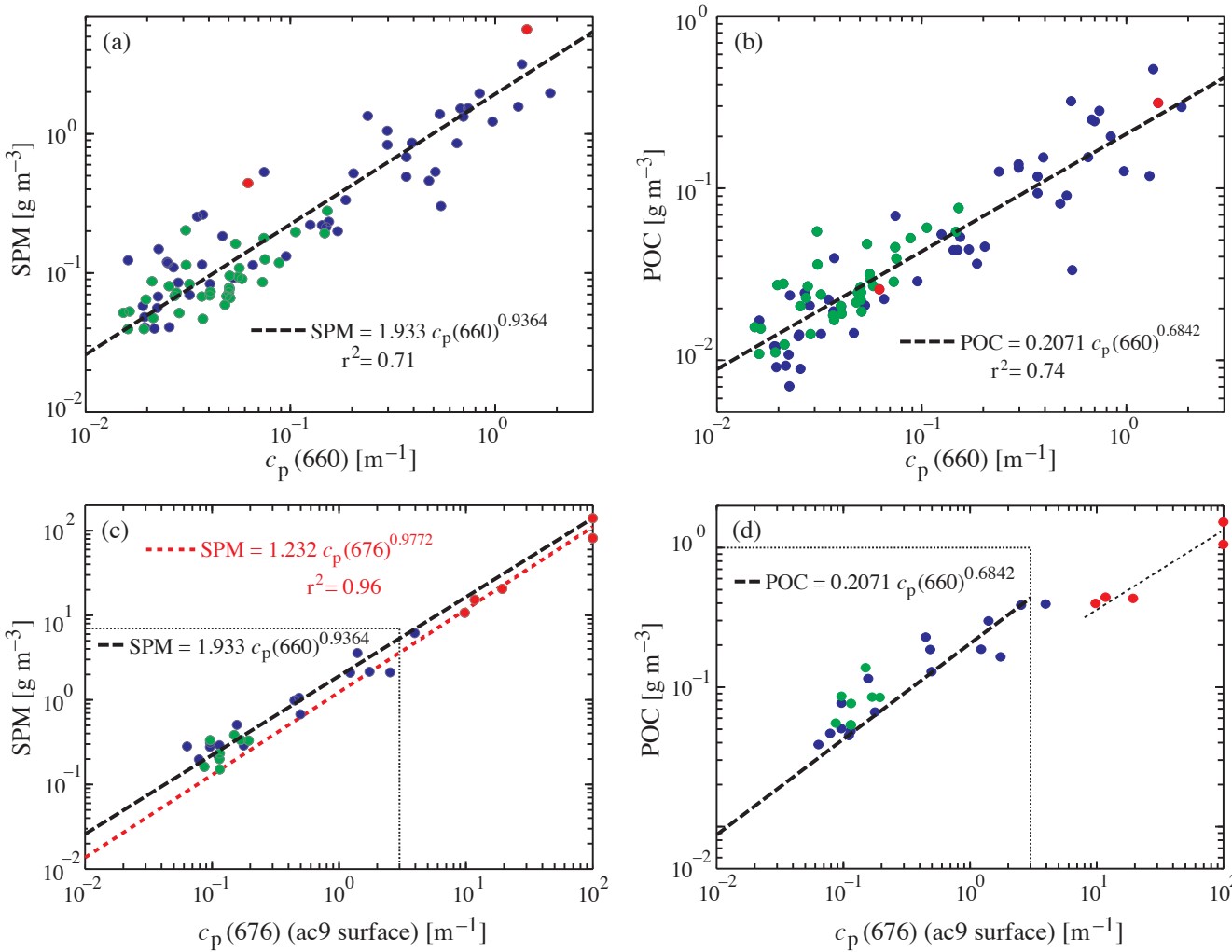

**Figure 6.** SPM and POC as a function of particulate beam attenuation coefficient at 660 nm based on measurements from *CCGS Amundsen* during the MALINA expedition in 2009 (a–b), and as a function of the particulate beam attenuation coefficient at 676 nm measured with the AC-9 from the barge (c–d) (the latter data contain only surface samples). The dotted squares in (c) and (d) indicate axes limits in (a) and (b), respectively. The colours of the data points indicate POC/SPM categories: mineral-dominated (red), mixed (blue), and organic-dominated (green).

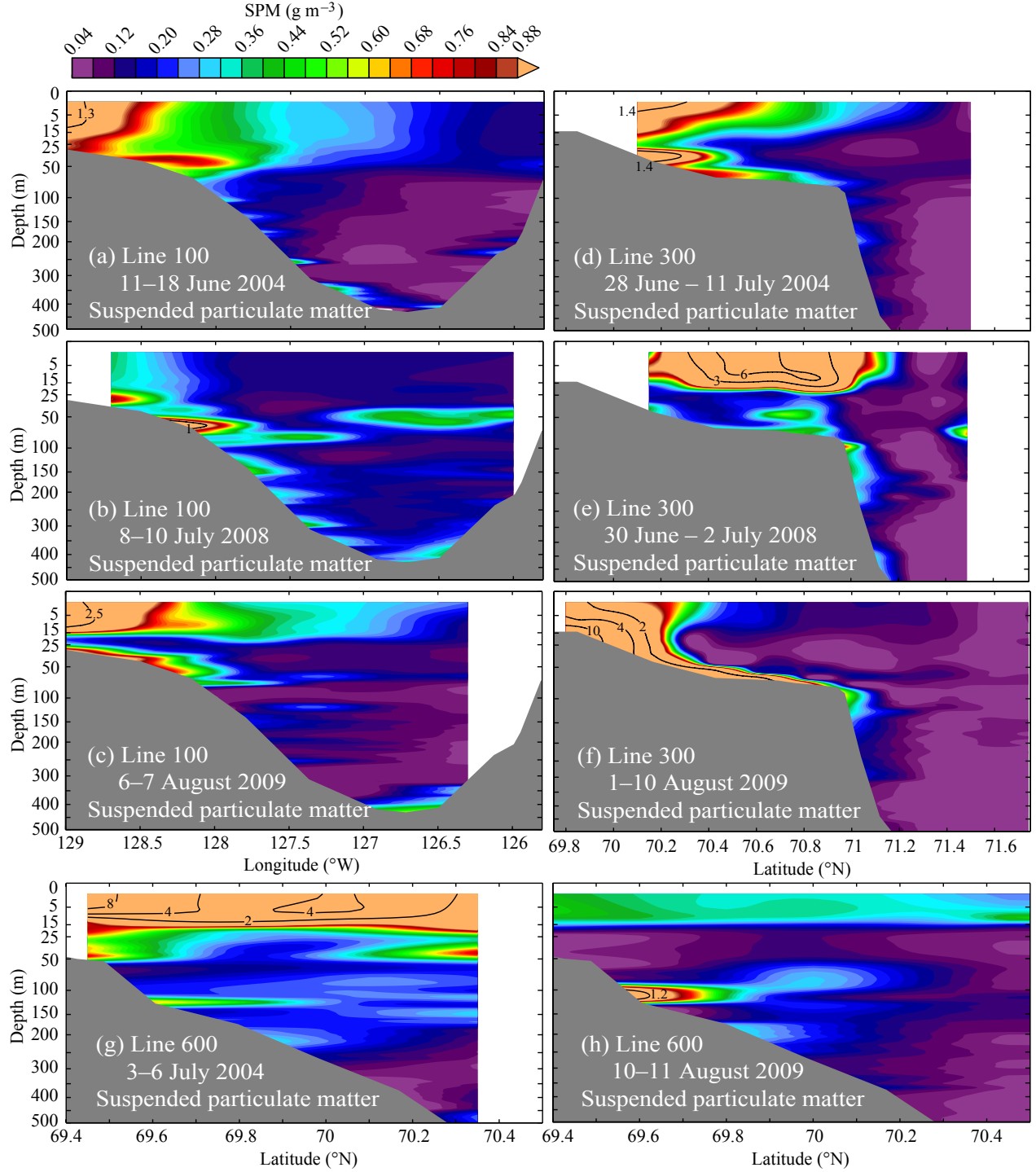

**Figure 7.** Concentration of suspended particulate matter, SPM, calculated from measurements of particulate beam attenuation coefficient at 660 nm, $c_p(660)$, using Eq. 2 for lines 100, 300 and 600 during different field campaigns in 2004, 2008, and 2009, as indicated.

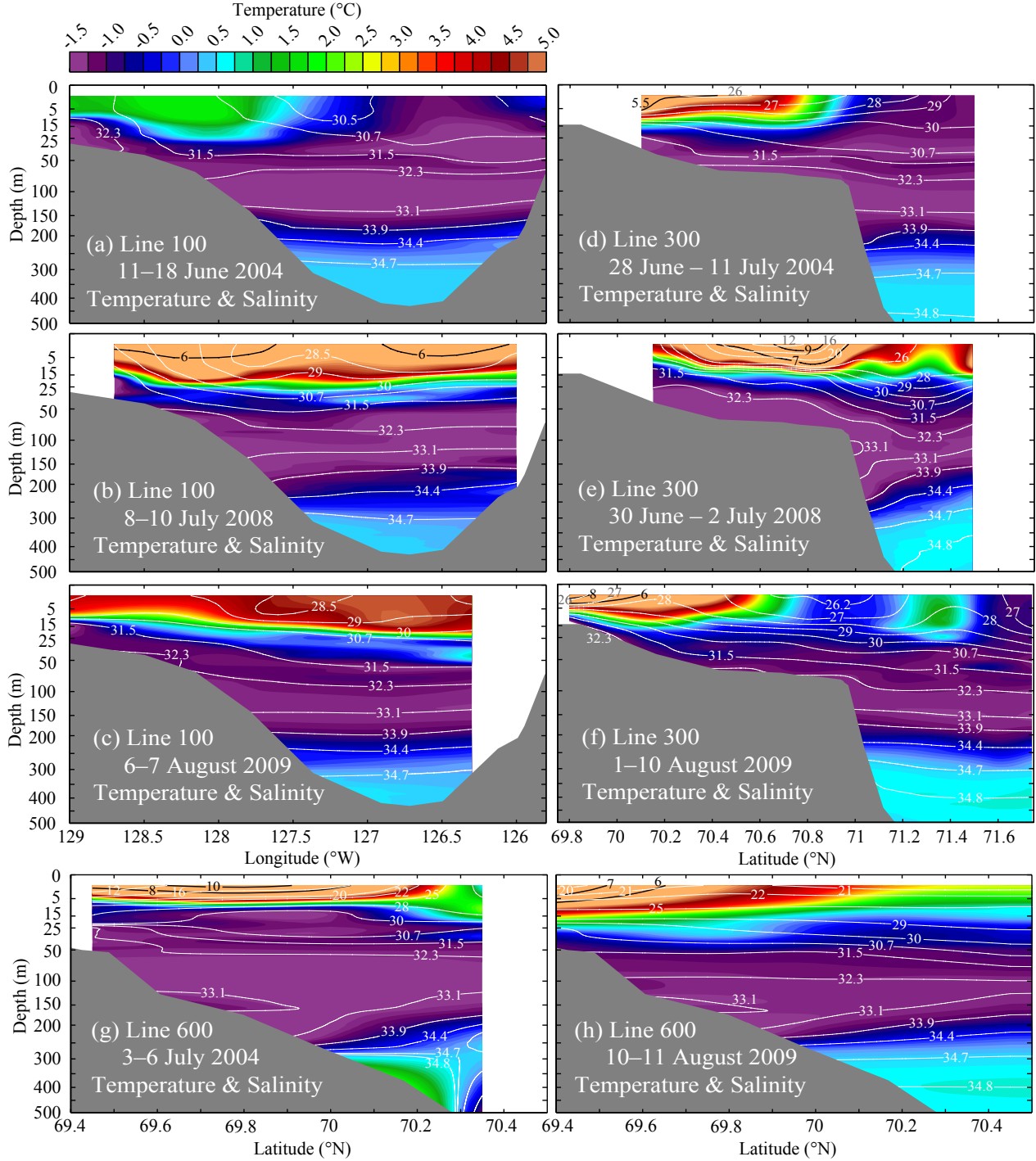

**Figure 8.** As Fig. 7 but for measurements of water temperature (colours) and salinity (contour lines).

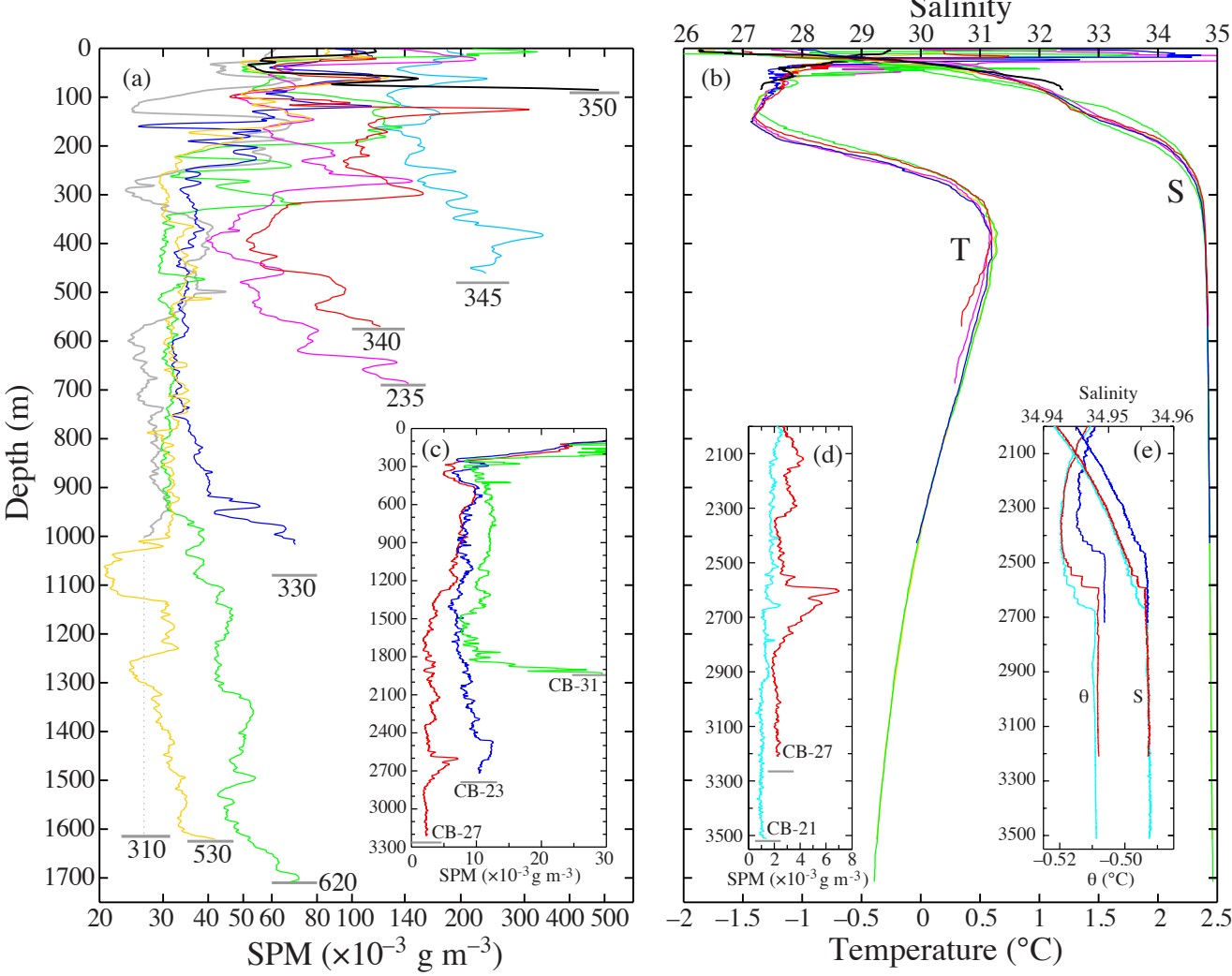

**Figure 9.** Vertical profiles of (a) suspended particulate matter, SPM, calculated from particulate beam attenuation coefficient at 660 nm, $c_p(660)$, and (b) temperature, $T$, and salinity, $S$, at selected "deep" stations. Inserts (c) and (d) show transmissometer data (converted to SPM using Eqs. 1–2) that were collected in Canada Basin during 21–23 September 2009 and made available by the Beaufort Gyre Exploration Program based at the Woods Hole Oceanographic Institution (http://www.whoi.edu/beaufortgyre) in collaboration with researchers from Fisheries and Oceans Canada at the Institute of Ocean Sciences. Insert (e) shows a close up of the potential temperature, $\theta$, and $S$ for CB-23, CB-27 and CB-21 at the interface to the Canada Basin Bottom Water layer. Grey horizontal lines indicate bottom depths and are underlain by station numbers (see Fig. 1 for locations).

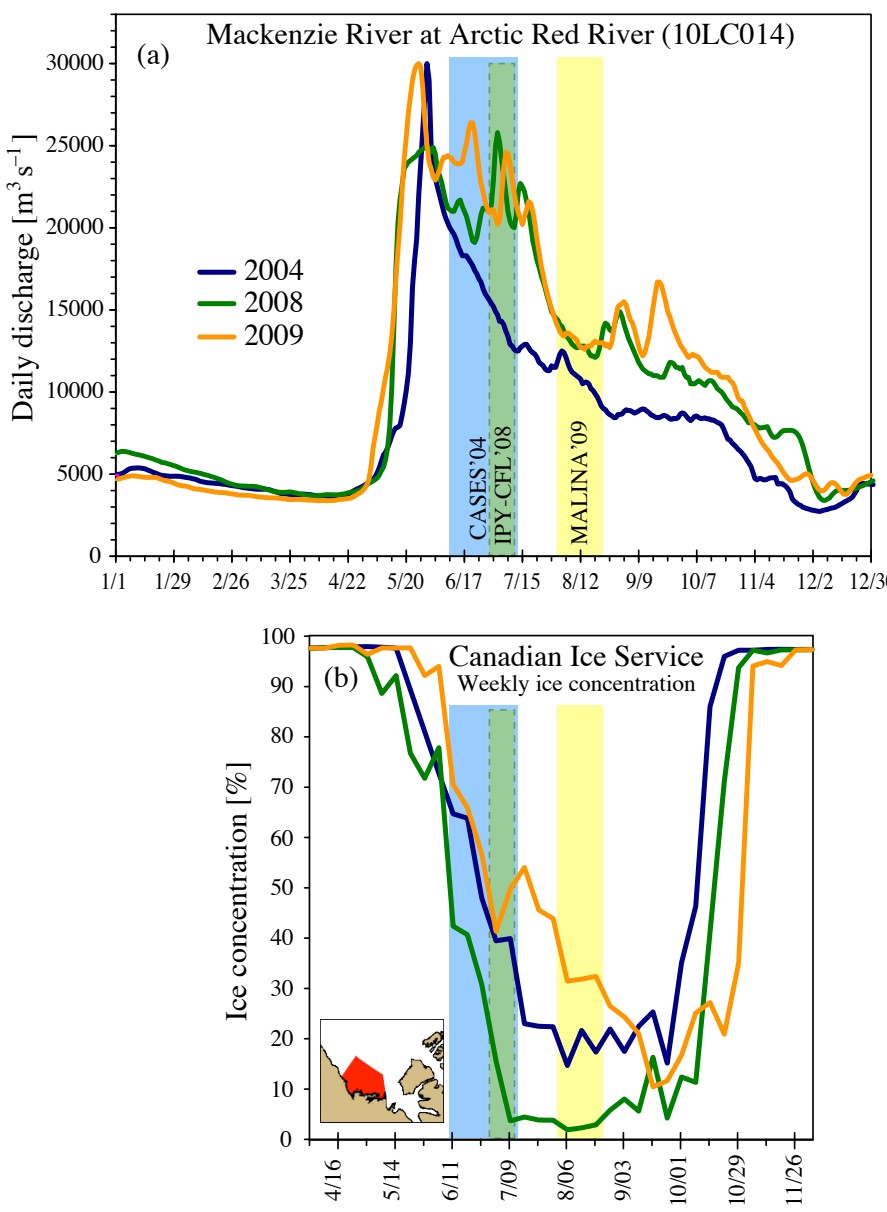

**Figure 10.** (a) Daily discharge for the Mackenzie River at the Arctic Red River location (10LC014). Data obtained from Environment Canada. (b) Weekly ice coverage for the Mackenzie Shelf area calculated using IceGraph 2.0 provided online by the Canadian Ice Service. The distributions of sea ice types are provided in Fig. S4 in the supplementary material. Time periods for the three expeditions considered in this study are also indicated in colour shades. The x-axis labels are dates expressed in the format of month/day, and are spaced 4 weeks apart.

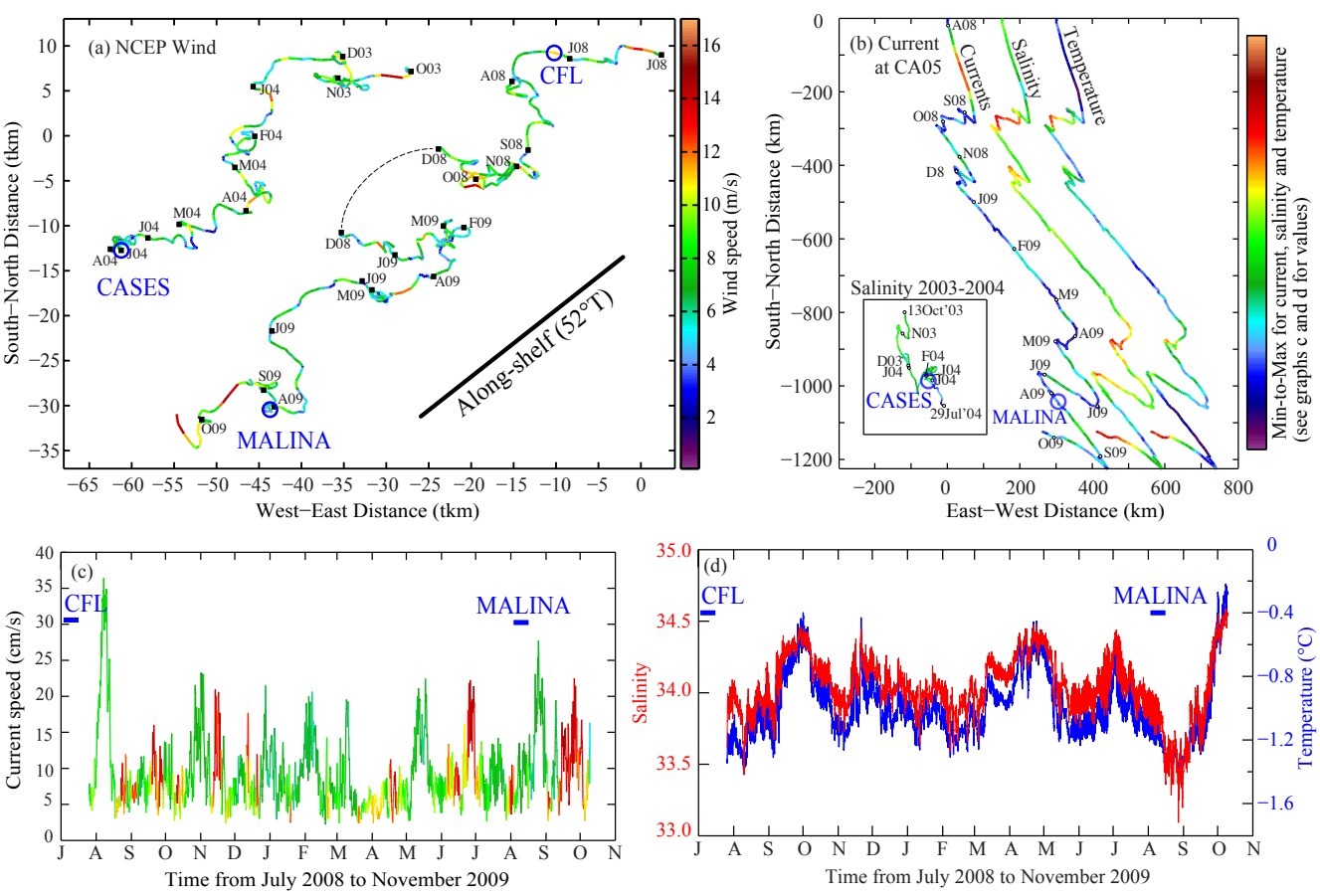

**Figure 11.** Progressive vector plots of (a) daily average wind (NCEP, 10 m) and (b) currents from mooring CA05, with inset for 2003–2004 CASES data. Colours in (a) indicate daily average wind speeds shown in colour bar. The black squares in (a) and (b) indicate the start of each month (the first letter of month followed by year, e.g., D03 stands for December 2003), while the blue circles show the approximate times of the ship-based transect sampling across the Mackenzie shelf break used in this study. The black line shows the direction along the shelfbreak referenced to True North. In (b), the same vector plot for currents is shown three times, but the colours of each plot indicate either current speed, salinity or temperature as denoted next to the lines and shown in more detail in (c) and (d). For the inset in (b) showing 2003–2004, the colours indicate salinity as denoted. Time series for 2008–2009 for (c) current speed (colours are current directions), (d) salinity (red) and temperature (blue) at 178 m depth.

**Table 1.** Salinity, saline end-member, meteoric and sea ice melt fractions (%), and meteoric water percentage of freshwater content (MW% = $f_{MW}/(f_{MW} + f_{SIM}) \times 100$) for surface seawater samples obtained at matching station locations during CASES 2004 and MALINA 2009. A few matching station were also sampled during IPY-CFL 2008. The samples were collected with Niskin bottles on a CTD-Rosette. The fraction of the saline end-member ($f_{PSW}$) represents Pacific Summer Water with a salinity of 31.5 PSU.

| Cruise | Station | Salinity | $f_{PSW}$ | $f_{MW}$ | $f_{SIM}$ | MW% |
|--------|---------|----------|-----------|----------|-----------|------|
| CASES | 110 | 25.4 | 80.1 | 6.6 | 13.3 | 34.8 |
| 2004 | 140 | 27.8 | 84.6 | 5.1 | 10.4 | 33.2 |
| | 150 | 29.3 | 89.1 | 7.5 | 3.4 | 68.6 |
| | 170 | 29.8 | 91.0 | 7.0 | 2.1 | 77.5 |
| | 320 | 29.4 | 89.2 | 1.3 | 9.4 | 12.2 |
| | 340 | 27.1 | 81.4 | 3.4 | 15.2 | 18.3 |
| | 360 | 25.1 | 79.9 | 19.6 | 0.5 | 97.5 |
| | 380 | 24.8 | 76.2 | 19.5 | 4.3 | 82.0 |
| | 390 | 25.4 | 78.8 | 20.4 | 0.8 | 96.3 |
| | 660 | 15.9 | 48.5 | 40.9 | 7.7 | 79.3 |
| | 670 | 16.9 | 52.6 | 40.2 | 7.2 | 84.7 |
| | 690 | 8.8 | 26.0 | 60.5 | 13.5 | 81.8 |
| IPY-CFL | 110 | 28.2 | 85.1 | 4.9 | 10.1 | 32.5 |
| 2008 | 140 | 28.2 | 85.1 | 4.3 | 10.6 | 28.8 |
| | 160 | 30.3 | 91.6 | 3.7 | 4.7 | 44.4 |
| | 320* | 26.3 | 79.5 | 12.4 | 8.1 | 60.5 |
| | 340* | 25.0 | 75.5 | 17.5 | 7.1 | 71.1 |
| | 390 | 29.4 | 88.9 | 11.0 | 0.2 | 98.6 |
| MALINA | 110 | 28.9 | 86.6 | 4.8 | 8.6 | 35.6 |
| 2009 | 150 | 29.4 | 89.4 | 4.9 | 5.7 | 46.1 |
| | 170 | 29.3 | 89.9 | 6.3 | 3.9 | 62.0 |
| | 320 | 26.5 | 79.2 | 6.3 | 14.5 | 30.2 |
| | 340 | 26.9 | 79.7 | 4.5 | 15.6 | 22.2 |
| | 360 | 26.5 | 78.4 | 4.6 | 17.0 | 21.2 |
| | 380 | 27.7 | 83.1 | 6.1 | 10.8 | 36.0 |
| | 390 | 27.2 | 83.5 | 7.8 | 8.7 | 47.2 |
| | 660 | 21.9 | 63.9 | 17.0 | 19.1 | 47.1 |
| | 670 | 23.4 | 68.0 | 15.3 | 16.7 | 47.8 |
| | 690 | 27.2 | 67.2 | 18.6 | 14.2 | 56.7 |

*Approximately 5 nm south of station, which is half way to the next station