# Peer review of "Patterns of suspended particulate matter across the continental margin in the Canadian Beaufort Sea during summer"

_Biogeosciences, 2018_

## Referee Comment (RC1) · Anonymous Referee #1 · 12 Jul 2018

REVIEW of the manuscript:

**"Patterns of suspended particulate matter across the continental margin in the Canadian Beaufort Sea", Jens K. Ehn, Rick A. Reynolds, Dariusz Stramski, David Doxaran, and Marcel Babin**

General comments

The present manuscript reports on the distribution and patterns of suspended particulate matter (SPM) and associated optical properties in the Canadian Beaufort Sea. Specifically, the authors demonstrate the correlation between the particulate beam attenuation and the dry mass concentration of SPM and use it to extend the SPM data to stations where only beam attenuation measurements were done. The obtained SPM distribution is discussed in relationship with environmental forcing such as wind, river discharge and sea ice coverage. The authors show that these forcings result in different circulation modes, upwelling onto the shelf, downwelling return flow across the shelf and vertical mixing due to strong wind conditions.

The manuscript is clearly written, the methods well explained and the graphs mostly illustrate the data accordingly.

My major concern about this manuscript relates to its structure. The authors present in a first step the optical (beam attenuation) and SPM data obtained during MALINA 2009 cruise and use these data to develop the SPM algorithm. The algorithm is then applied to beam attenuation data obtained from 4 other cruises in the Canadian Beaufort Sea, in order to extend the SPM data set. The second step consists in presenting almost a new manuscript with a first description of environmental parameters and then of different patterns of SPM distribution.

Although the presented structure is clear, the different pieces (paragraphs) are rather isolated and their contribution to the scientific question remains unclear.

I would therefore propose a different approach, which consists in keeping the first part with the MALINA data and use the data of the second part to do a statistical analysis relating the SPM patterns to the different environmental scenarios. Not only would the findings be more robust by being "statistically" supported compared to the only descriptive presentation in the present manuscript, but also the manuscript as a whole would appear more coherent with respect to SPM patterns related to environmental forcing.

I will give more detailed arguments in the specific comments in order to better explain my proposition.

Specific comments

Introduction: The scientific context is well presented. Particle origin and transport ways, as well as the different factors to which beam attenuation is sensitive (concentration, size and composition of particles) are introduced, and one would expect that these factors would be discussed accordingly within the manuscript. Even if at the end of the introduction, the authors solely talk about particles, the reader would suppose that they mean organic and mineral but also different sizes of particles.
Also, clearly, the authors admit temporal variations of particle characteristics

but intend to relate the distribution patterns to oceanographic conditions (last sentence of the introduction).

This is exactly what could be answered by my above proposition: A robust, statistical relationship between environmental parameters and particle distribution takes into account the different variabilities and overcomes at least at a certain probability level such uncertainties.

Paragraph 3.1.2.: The paragraph could be removed and the beam attenuation results presented together with the data from the other cruises.

The fluorescence is certainly an important parameter for the particle characteristics, but the authors do not discuss these data (paragraph 3.4.5.) very extensively. E.g. they could use them to see how autochthonous production of particles and the related difference in distribution dynamics influences the general particle distribution pattern. Also, there is no discussion on its influence on the beam attenuation data, although the authors clearly state it (line 30, page 7).

Paragraph 3.1.3.: As said before, several characteristics of particles that influence the beam attenuation are presented, but this aspect is not really included in later discussions.

Some interesting findings are presented about mineral and organic dominated particle composition, but none of this is being considered when it comes to a general discussion on the particle distribution patterns, unless I have overseen this point.

The same accounts for the particle volume distribution and the particle size distribution (PSD). The data of the former are not so much of a surprise to me and I do not think that they contribute substantially to the science of this manuscript. However, the data about PSD deserve more attention than given by the authors. The description (lines 1-6, page 9) is rather confusing and a table or a graph would shed much more light on them. Also, the authors could use these data to discuss points like optical properties of different size spectra, is the chosen wavelength (660 nm) appropriate for all types of spectra etc. Some of the co-authors (Reynolds, Stramski) have signed a very nice article in L&O, 61, 2016, which I would consider as a model case of thorough discussion related to the same subject. I could imagine that this opens many possibilities of parameters to be used for statistical treatment.

Paragraph 3.2.: The relationships and different regressions are presented in much detail. While some of them are not necessary, others add more confusion than clarity. E.g. what do the two measurements, RMSE and MNB, add to the regression coefficient? The latter is rather well known, but the former may need some explanation in order to be evaluated by the reader, e.g. reference values for the two (0, 1) would permit an evaluation of the presented results.

The explanation of the regressions of the $c_p$ (660) and $c_p$ (676) vs. SPM data (lines 18-24, page 10) are confusing. It is not clear which points were used for the two analyses, red points for red regression? but red stands also for mineral-dominated, i.e. are there only mineral dominated data for 676 nm measurements? In this case, it is maybe worth to explore if the measurements for the two wavelengths can be merged, which would at the same time better justify the argument that equation 2) is used for high SPM values (lines 21,22,

page 10).

Lines 20-23, page 9: If differences in $r^2$ are not significant, there should be a better argument than just "appears to best match" for choosing a linear power function fit, unless the RMSE and MNB measures are better explained.

In the same sense, what conclusion can be drawn from the fact that a non-linear power function fit is best for SPM data and a linear regression to log-transformed data best for POC data? This brings me to a general question about establishing relationships between optical and biological measurements. Is it possible to attribute some functional meaning to a given class of data fits? For example, if the fit is a power function, is this related to growth rates of phytoplankton and if it is a linear fit, is it related to cell density etc.?

Paragraphs 3.3. and further: It appears as if the SPM data from the other cruises are used to discuss the patterns from MALINA by choosing the contrasting or similar situations. Examples:

1) Wedges of clear water found over the shelf due to near meltwater from extensive ice coverage, as opposed to low ice coverage in other cruises where clear water is absent (paragraph 3.4.1.).

2) High near bottom SPM concentration during MALINA related to downwelling return flow as opposed to 2008 upwelling situation with high river plume extension and low bottom SPM concentration (paragraph 3.4.2.).

3) Similar SPM patterns between MALINA and CASES 2004, but higher SPM concentration during CASES due to timing of the year (recent break up of land fast ice cover) (paragraph 3.4.3.).

These examples together with the points discussed in paragraphs 3.4.4. (high SPM concentrations in a well-mixed water column due to upwelling) and 3.4.5 (primary production depends on sea ice coverage (light availability), nutrient availability and river plume extension related to wind conditions) are all criteria which could be generalized and chosen as parameters for a statistical analysis to explore relationships between the main environmental factors sea ice coverage, river discharge and wind and the typical patterns of SPM distribution quantified by the dry mass concentration of SPM across the shelf and into the Canada Basin.

Since the descriptions given in these paragraphs are rather clear, I could well imagine that a statistical analysis will yield significant results, which is in my view the ideal way to apply statistical analyses to environmental data: First, you inspect the data in a rather subjective manner, then you are able to apply the appropriate statistical analysis to obtain an objective result with a given amount of error.

Finally, the discussion in paragraph 3.4.6. was the least convincing. Examples:

1) line 6, page 18: Fig. 12 does not show the cast-to-cast variability.

2) line 13, page 18: it is rather difficult to define the bottom layer thickness from the presented profiles.

3) lines 20-26: the authors may be able to see flow patterns of INLs, but the reader may as well see other patterns.

Again, a statistical analyses would (or not) remove any doubt about the proposed explanations of the different patterns of nepheloid layers.

Figures: By consequence of my proposition, the figures 6, 7, 8, 9, 12 and maybe 11 would need to be modified or even removed and figure 10 remains the key figure.

Technical corrections

- Lines 1, 4, page 2: Mass units are generally given in g, i.e. Tg instead of Mt
- Lines 5-6, page 2: If 50% are deposited in the delta and 40% on the shelf then the fraction
  across the shelf break is not poorly known, but should most likely be 10%
- Line 30, page 2: …part of the MALINA project…
- Line 26, page 4: The blank value seems a bit high to me. Is this common for the used
  instrument?
- Line 6, page 7: Instead of the questioned Matsuoka reference, I would suggest: McDonald
  et al., 1989, JGR and/or Carmack et al., 1989, JGR, which are the refs.
  mentioned in Matsuoka.
- Line 8, page 8: …Only at station 394….
- Line 33, page 10: which transect is meant?
- Line 14, page 14: …which corresponds to the Mackenzie….
- Line 17, page 14: …to the northerly and rather weak….
- Line 6, page 20: …at a depth corresponding to an…..
- Line 10, page 20: …(Fig. 8d)….
- Line 15, page 23: The reference Guay et al. is not cited in the manuscript
- Line 25, page 24: Timmermans et al. should appear after Stroeve et al.

---

## Referee Comment (RC2) · Anonymous Referee #2 · 13 Jul 2018

Review of 'Patterns of suspended particulate matter across the continental margin in the Canadian Beaufort Sea' by Jens Ehn, Rick Reynolds, Dariusz Stramski, David Doxaran and Marcel Babin for Biogeosciences

I was very interested to read this manuscript, whose main goal is to develop a relationship between beam attenuation data collected by transmissometers and suspended particulate matter and particulate organic carbon. Much archived transmissometer data exist for this region so finding such relationships could give valuable historic information on suspended particles. This is a very complex region and others have struggled to find statistically significant relationships between these properties in such complex regions. In general, I think that the authors did a convincing job of showing that there are robust relationships.

It seemed like a secondary goal of this manuscript was to describe the physical forcing responsible for the high or low particulate concentrations. Unfortunately, this is where I think this manuscript fell apart. The interpretation of the physical data was vague and few solid conclusions could be made from the very long discussion. I got the sense that the authors had limited understanding of the physical oceanography of this region.

Perhaps a better approach would be to choose only one physical process that is related to suspended particles. I see the main storyline of the manuscript as a comparison between the attenuation and bottle data. Proof of concept of this relationship could be shown by focusing on only one process, such as the Mackenzie River plume. There is room here for a very thorough study of this process and much new information could be gained on by coming up with concrete conclusions related to one physical process.

I recommend that this manuscript undergoes major revision before it can be reconsidered for publication. Below are several other concerns and suggestions that I have:

- Page 1, line 6 – Several times throughout the manuscript, the authors state that the surface layer is a mixture of sea ice melt and river runoff. While this may be true, the composition of the surface layer hasn't been quantified so the source of the particles in the freshwater can't be determined. The authors could attempt a freshwater budget as was done by Yamamoto-Kawai et al. (2008, doi:10.1029/2006JC003858) or they can acknowledge that they don't know the source of the freshwater or the particles therein.

- The Introduction is in general quite confusing. I think that a more clear description of the region would greatly help readers not familiar with this area. In addition, a stronger literature review of previous SPM and POC work in this region, and the mechanisms that transport these particles, would set the stage nicely for focusing this manuscript.

- Page 2, second paragraph – This paragraph is quite confusing and needs more focus

- Page 3, lines 14-15 – Why weren't lines 400 and 500 analyzed in this study?

- Page 3, lines 25 -27 – What other depths were sampled in addition to the surface and

SCM?

- Page 3, line 29 – What is an aliquot?

- Page 3, line 29 – What is considered a sufficient volume of water? Was this based on the time it took to filter or something else? What determines whether a duplicate or triplicate was sampled?

- Page 4, lines 10 to 12 – Some other studies sampling SPM rinse the filters with ammonium bicarbonate or ammonium formate. Could the authors please explain why they didn't do this?

- Page 5, line 8 – Please describe the interquartile range method, with a reference if applicable

- Section 2.4 – Comparison of data between different transmissometers is notoriously difficult due to different calibration values and instrument drift. Is the use of dark voltage offset to allow for comparison of tranmissometer data between cruises? If so then could the authors please state the accuracy of this method, with references if applicable.

- Page 6, line 7 – What depth were the other sensors located at on the mooring?

- Section 2.5 – Why were data from only CA05 shown? This mooring is at the edge of the Cape Bathurst upwelling region, which is not particularly representative of the region. Several other moorings have been deployed along the Canadian Beaufort during the study period, 2004 to 2009. Why was only this mooring selected to represent the region?

- Page 6, line 28 – where is the proof that there were strong easterly winds in June 2009?

- Section 3.1.1 – Please add some references to the different water mass definitions.

- Page 7, lines 5-6 – Please see Jackson et al (2015, 10.1002/2015JC010812 ) for information on Pacific winter water in this region

- Page 7, lines 10 – 17 – The c_p values in this paragraph don't appear to match those in Figure 2

- Page 7, line 18 – what does the 'strong chl-a fluorescence signal mean? Couldn't they be quantified by discrete chlorophyll samples?

- Page 7, line 22 – What is the source of CDOM in Pacific Winter water? Perhaps more information can be added from Guegen et al., 2012 (doi:10.1016/j.dsr2.2011.05.004 )

- Page 8, line 10 – There is no information about the location or methods used for barge sampling

- Page 8, lines 24 – 26 – Why is it not possible to measure PSD using the Coulter technique in low salinity, turbid waters?

- Page 8, line 32 – I disagree with this statement. The relationships shown in Figure 4b are not very convincing. Is the relationship statistically significant?

- Page 9, lines 8-10 – What is the difference between the MALINA and Amundsen data?

- Page 9, lines 18 -20 – Of these three regression analyses, why is only ii) shown in Figure 5?

- Page 9, line 26 – do the authors mean 'nonlinear power function' instead of 'nonlinear least squares regression'?

- Page 12, lines 20 – 26 – Figure 8 is very unclear and possibly incorrect. I can't see the statements in this paragraph supported in Figure 8

- Page 12, line 28 – I can't see the wind speeds in Figure 8a

- Page 12, lines 31-35 – Don't these lines contradict lines 11-13 on page 12?

- Section 3.3.3. This section need much more work. How was cross-shelf defined? What depths were influenced by cross shelf currents? How do we know that the currents observed at CA05 were representative of the rest of the region? How was a cross-shelf episode defined? I'm not entirely sure how this section is giving evidence of upwelling and relaxation

- Section 3.3.4 – I don't see the point of this section. What new information does it tell us about SPM and POC in the Canadian Beaufort Sea?

- Page 13, line 29 – Could the authors please be more clear with where the current intensification is observed? I refer the authors to Forest et al (2015, http://dx.doi.org/10.1016/j.csr.2015.03.009 ) for discussion of other strong shelfbreak currents in the region

- Page 14, lines 10-11 – What causes this clear water extension onto the shelf?

- Page 14, lines 24-25 – Is there proof of downwelling return flow and after upwelling?

- Section 4 – I am unclear exactly how the physical observations described in section 3 lead to the listed conclusions in section 4. Much more work needs to be done to understand the physical processes before they can be related to the particle concentrations

- Figure 1 – Please make the CA05 mark larger and easier to see

o It is difficult to distinguish between the different colored stars

- Figure 2 – Why is the very freshest water on the western shelf away from the Mackenzie River? It doesn't appear that this very fresh water is correlated with the highest attenuation

o It would help the reader understand the text if the stations could be marked on these figures

o What does the grey area mean?

- Figure 3 – Why are the error bars backwards?

- Figure 4 – It would help the reader interpret this if boundaries were drawn around the

3 different defined areas

o Figure 4b – I don't see very strong, statistically significant relationships here. Also, is the salinity from the same depth that the water was sampled from?

- Figure 6 – It is difficult to see the writing of the different cruises

o The data look smoothed. Can the authors please state how they smoothed the data?

- Figure 8 – I really struggled with this figure. It is difficult to interpret, has very small writing, and has a huge amount of information.

o The wind and current data in particular were difficult to distinguish. It was near impossible to see upwelling or downwelling as this figure was laid out

o I think that the depths in the temperature and salinity plots were mislabeled – the shallower water shouldn't be saltier

o There should be some explanation as to why such salty water was observed in figure 8d. Water of this salinity would have to be upwelled from several hundred meters depth, and a significant upwelling event would need to be evident in the wind data.

o I don't understand what the different colours mean in 8c

- Figure 9 – Please mark the mooring location on line 100 (CA05)

o I couldn't distinguish between the different contour lines

o Please include the station locations

o The current values don't make sense to me. General definitions in oceanography are that northward and eastward currents are positive and southward and westward currents are negative. Having different definitions makes this figure very confusing

---

## Author Response (AR1)

RESPONSES to the review of the manuscript:

"Patterns of suspended particulate matter across the continental margin in the Canadian Beaufort Sea", Jens K. Ehn, Rick A. Reynolds, Dariusz Stramski, David Doxaran, Bruno Lansard, and Marcel Babin

We greatly appreciate the constructive comments from both reviewers. Here we provide our detailed point-by-point responses and any description of action taken in regards to the comments by Referee #1. The Referees' comments are shown in regular font; our responses follow each comment in blue font.

**Response to Referee #1**

General comments

The present manuscript reports on the distribution and patterns of suspended particulate matter (SPM) and associated optical properties in the Canadian Beaufort Sea. Specifically, the authors demonstrate the correlation between the particulate beam attenuation and the dry mass concentration of SPM and use it to extend the SPM data to stations where only beam attenuation measurements were done. The obtained SPM distribution is discussed in relationship with environmental forcing such as wind, river discharge and sea ice coverage. The authors show that these forcings result in different circulation modes, upwelling onto the shelf, downwelling return flow across the shelf and vertical mixing due to strong wind conditions.

The manuscript is clearly written, the methods well explained and the graphs mostly illustrate the data accordingly.
My major concern about this manuscript relates to its structure. The authors present in a first step the optical (beam attenuation) and SPM data obtained during MALINA 2009 cruise and use these data to develop the SPM algorithm. The algorithm is then applied to beam attenuation data obtained from 4 other cruises in the Canadian Beaufort Sea, in order to extend the SPM data set. The second step consists in presenting almost a new manuscript with a first description of environmental parameters and then of different patterns of SPM distribution.

Although the presented structure is clear, the different pieces (paragraphs) are rather isolated and their contribution to the scientific question remains unclear.
I would therefore propose a different approach, which consists in keeping the first part with the MALINA data and use the data of the second part to do a statistical analysis relating the SPM patterns to the different environmental scenarios. Not only would the findings be more robust by being "statistically" supported compared to the only descriptive presentation in the present manuscript, but also the manuscript as a whole would appear more coherent with respect to SPM patterns related to environmental forcing.

I will give more detailed arguments in the specific comments in order to better explain my proposition.

REPLY:  We would like to thank Referee #1 for the insightful comments that spurred us to take a critical look at the structure of our manuscript. We agree mostly with the revisions that have been suggested by Referee #1. We agree that the link between the SPM algorithm development using MALINA data and the second part that involved comparisons to forcing conditions required clarifications. In the revised manuscript, we have put much effort into focusing the paper by rearranging its structure and removing unnecessary descriptions. However, considering unavoidable limitations in the available data sets, the possibility of conducting a statistical analysis of the kind suggested by Referee #1 appeared to us highly problematic. Instead of attempting such statistical analysis, we have followed the advice of Referee #2 and increased a focus of the manuscript on particle characteristics associated with freshwater inputs. This involved including a new data set of water oxygen isotopic composition. See also our reply to the comment on Paragraph 3.3.

Regarding the lack of exploitation of the effects of particle size and composition on the SPM vs. cp relationship, we point out that we have to rely on one relationship regardless of particle size and composition because we apply the relationship to the cp data measured during different field experiments when no ancillary data on particle size and composition were available. In contrast to cp, which is routinely collected in the field as a part of CTD casts, the particle size and composition data are rarely collected except during focused/dedicated field experiments. Thus, in this paper we use the particle size and composition data gathered on MALINA primarily to indicate that our relationships between cp, SPM, and POC are robust over a broad range of variability in the particle assemblage.

Given the quite extensive scope of revisions that we made with regard to restructuring the mansucript and making changes in the content of various sections, it would be impractical to describe each and every change related to the restructuring in this response. We believe, however, that these main changes are easily identifiable in the revised manuscript.

Specific comments

Introduction: The scientific context is well presented. Particle origin and transport ways, as well as the different factors to which beam attenuation is sensitive (concentration, size and composition of particles) are introduced, and one would expect that these factors would be discussed accordingly within the manuscript. Even if at the end of the introduction, the authors solely talk about particles, the reader would suppose that they mean organic and mineral but also different sizes of particles. Also, clearly, the authors admit temporal variations of particle characteristics but intend to relate the distribution patterns to oceanographic conditions (last sentence of the introduction).
This is exactly what could be answered by my above proposition: A robust, statistical relationship between environmental parameters and particle distribution takes into account the different variabilities and overcomes at least at a certain probability level such uncertainties.

REPLY: To improve the description of the effects of different factors on beam attenuation we included a more detailed description in paragraph 3 of the Introduction section. Earlier this text was part of the first paragraph of the original section 3.1.2, now 3.2.2, which has now been shortened. As mentioned above, with regard to statistical analysis, we have followed the advice

of Referee #2 and increased a focus of the manuscript on particle characteristics associated with freshwater inputs. We believe this is an important aspect which improved the manuscript. See also our reply to the comment on Paragraph 3.3.

Paragraph 3.1.2.: The paragraph could be removed and the beam attenuation results presented together with the data from the other cruises.

The fluorescence is certainly an important parameter for the particle characteristics, but the authors do not discuss these data (paragraph 3.4.5.) very extensively. E.g. they could use them to see how autochthonous production of particles and the related difference in distribution dynamics influences the general particle distribution pattern. Also, there is no discussion on its influence on the beam attenuation data, although the authors clearly state it (line 30, page 7).

REPLY: We have considerably rearranged this part of the text. The original sections 3.1.2 and 3.1.3 have been combined into a new section 3.2. The new sections 3.1 and 3.2 still focus on MALINA observations to show the ranges in water and particle characteristics, which underlie the development of statistical relationships presented in the section 3.3. Measurements of chl-a fluorescence are used throughout the revised manuscript (e.g., sections 3.2, 3.4) as an indicator of particle origin and characteristics.

Paragraph 3.1.3.: As said before, several characteristics of particles that influence the beam attenuation are presented, but this aspect is not really included in later discussions. Some interesting findings are presented about mineral and organic dominated particle composition, but none of this is being considered when it comes to a general discussion on the particle distribution patterns, unless I have overseen this point.

Routine beam attenuation measurements during the Arctic expeditions used in our analyses have not been accompanied with specialized analysis aimed at determining the particle composition and PSD characteristics (with the exception of a subset of MALINA data set). Because of this lack, we feel that speculations regarding the potential effects of particle assemblage properties (such as composition and PSD) on the discussion of general SPM patterns is unwarranted in the context of these additional cruises. The subset of MALINA data is used to indicate that our developed relationships are applicable over a wide range of variability in the particle assemblage.

The same accounts for the particle volume distribution and the particle size distribution (PSD). The data of the former are not so much of a surprise to me and I do not think that they contribute substantially to the science of this manuscript. However, the data about PSD deserve more attention than given by the authors. The description (lines 1-6, page 9) is rather confusing and a table or a graph would shed much more light on them. Also, the authors could use these data to discuss points like optical properties of different size spectra, is the chosen wavelength (660 nm) appropriate for all types of spectra etc. Some of the co-authors (Reynolds, Stramski) have signed a very nice article in L&O, 61, 2016, which I would consider as a model case of thorough discussion related to the same subject. I could imagine that this opens many possibilities of parameters to be used for statistical treatment.

REPLY: We have included a reference to Reynolds et al. (2016) and indicated that this study

includes a detailed discussion of the PSD data collected in Arctic waters, including results from MALINA. In the revisions we have focused on improving the presentation of the relationship between the particle composition and size characteristics and freshwater composition. In Fig. 5 (formerly Fig. 4) we have added two graphs illustrating how POC/SPM and PSD shape are related to meteoric water fractions present in surface water samples. In our view, these revisions address the points made above and focus the discussion on differentiating particle characteristics between sources (fluvial, sea ice melt, and pelagic).

Paragraph 3.2.: The relationships and different regressions are presented in much detail. While some of them are not necessary, others add more confusion than clarity. E.g. what do the two measurements, RMSE and MNB, add to the regression coefficient? The latter is rather well known, but the former may need some explanation in order to be evaluated by the reader, e.g. reference values for the two (0, 1) would permit an evaluation of the presented results.

The explanation of the regressions of the $c_p$ (660) and $c_p$ (676) vs. SPM data (lines 18-24, page 10) are confusing. It is not clear which points were used for the two analyses, red points for red regression? but red stands also for mineral-dominated, i.e. are there only mineral dominated data for 676 nm measurements? In this case, it is maybe worth to explore if the measurements for the two wavelengths can be merged, which would at the same time better justify the argument that equation 2) is used for high SPM values (lines 21,22, page 10).

Lines 20-23, page 9: If differences in $r^2$ are not significant, there should be a better argument than just "appears to best match" for choosing a linear power function fit, unless the RMSE and MNB measures are better explained.
In the same sense, what conclusion can be drawn from the fact that a non-linear power function fit is best for SPM data and a linear regression to log-transformed data best for POC data? This brings me to a general question about establishing relationships between optical and biological measurements. Is it possible to attribute some functional meaning to a given class of data fits? For example, if the fit is a power function, is this related to growth rates of phytoplankton and if it is a linear fit, is it related to cell density etc.?

REPLY: In response to these comment and to clarify the issues related to regression analysis, we have moved the detailed description of the regression fits to the Supplementary Materials, where we also provide the RMSE and MNB equations. In the revised manuscript only the two final chosen regressions are shown, which are then applied to beam attenuation data from the three Arctic expeditions. We have also made it clear that all data points, regardless of their "colour", are used in the final regression fits. For more details about the regression analysis, the readers are referred to the Supplementary Materials. This additional material also includes results for different types of regression fits. With the regards to functional meaning, we point out that the various models (linear, power) were not statistically different, and we chose the power function as this has been the most common approach used in the past. These are simply empirical best-fits to the relationship.

Paragraphs 3.3. and further: It appears as if the SPM data from the other cruises are used to discuss the patterns from MALINA by choosing the contrasting or similar situations. Examples: 1) Wedges of clear water found over the shelf due to near meltwater from extensive ice

coverage, as opposed to low ice coverage in other cruises where clear water is absent (paragraph 3.4.1.).

2) High near bottom SPM concentration during MALINA related to downwelling return flow as opposed to 2008 upwelling situation with high river plume extension and low bottom SPM concentration (paragraph 3.4.2.).

3) Similar SPM patterns between MALINA and CASES 2004, but higher SPM concentration during CASES due to timing of the year (recent break up of land fast ice cover) (paragraph 3.4.3.).

These examples together with the points discussed in paragraphs 3.4.4. (high SPM concentrations in a well-mixed water column due to upwelling) and 3.4.5 (primary production depends on sea ice coverage (light availability), nutrient availability and river plume extension related to wind conditions) are all criteria which could be generalized and chosen as parameters for a statistical analysis to explore relationships between the main environmental factors sea ice coverage, river discharge and wind and the typical patterns of SPM distribution quantified by the dry mass concentration of SPM across the shelf and into the Canada Basin.

Since the descriptions given in these paragraphs are rather clear, I could well imagine that a statistical analysis will yield significant results, which is in my view the ideal way to apply statistical analyses to environmental data: First, you inspect the data in a rather subjective manner, then you are able to apply the appropriate statistical analysis to obtain an objective result with a given amount of error.

REPLY: We agree that a statistical analysis would be ideal, however, it is unclear to us how to implement a statistical analysis with the actual limited availability of field data in this particular study to make this analysis quantitatively meaningful. Although the transect lines we have chosen are probably the most sampled in the Canadian Beaufort Sea, this is still a limited number of data. We note that past studies, including recent studies using extensive mooring timeseries such as Forest et al. (2015) and Jackson et al. (2015), take a similar approach to our study and use inference to understand processes on the shelf. A numerical model sensitivity analysis of different factors affecting SPM distributions would, in our opinion, provide probably the best way forward to deduce statistical relationships. This would, however, constitute a separate study on its own and is beyond the scope of our study. Our result could be useful for evaluating such model and we have added this statement at the end of Conclusions section in the revised manuscript.

Finally, the discussion in paragraph 3.4.6. was the least convincing. Examples: 1) line 6, page 18: Fig. 12 does not show the cast-to-cast variability.

2) line 13, page 18: it is rather difficult to define the bottom layer thickness from the presented profiles.

3) lines 20-26: the authors may be able to see flow patterns of INLs, but the reader may as well see other patterns.

Again, a statistical analyses would (or not) remove any doubt about the proposed explanations of the different patterns of nepheloid layers.

REPLY: We have made significant modifications of this section. We no longer mention cast-to-cast variability. The text in lines 20-26 has been deleted. We have, however, kept the figure (originally Fig. 11, now Fig. 9) and a brief discussion of this figure, as we want to show one

graph with individual cast (other cp data are shown only as contour plots) and illustrate the SPM concentrations on the shelf within a context of what is observed in offshore Canada Basin waters.

Figures: By consequence of my proposition, the figures 6, 7, 8, 9, 12 and maybe 11 would need to be modified or even removed and figure 10 remains the key figure.

Technical corrections

- Lines 1, 4, page 2: Mass units are generally given in g, i.e. Tg instead of Mt
REPLY: We have changed to Tg although we note that the source reference Macdonald et al. (1998) uses the units of Mt.

- Lines 5-6, page 2: If 50% are deposited in the delta and 40% on the shelf then the fraction across the shelf break is not poorly known, but should most likely be 10%
REPLY: The sentence has been rewritten as follows: "Macdonald et al (1998) recognize that sedimentation rates on the shelf are poorly known, but estimate that about 40% of the sediment input to the shelf is deposited while about 13 % is transported across the shelfbreak either in surface river plumes, near the bottom in nepheloid layers, or by ice rafting." The cited paper includes large ranges in these values.

- Line 30, page 2: ...part of the MALINA project...
REPLY: Added "the".

- Line 26, page 4: The blank value seems a bit high to me. Is this common for the used instrument?
REPLY: We have not been able to ascertain the typical blank values for this instrument. Note that our POC measurements were made on the same filters as were used for SPM. Thus, the blank filter preparation also followed the SPM protocol steps such as weighting, rinsing with milli-Q, etc. This may have contributed to higher blank values than what might be otherwise expected. Nevertheless, we filtered sufficient volumes of sample water such that the carbon signal on the filter was significantly higher than the blank values.

- Line 6, page 7: Instead of the questioned Matsuoka reference, I would suggest: McDonald et al., 1989, JGR and/or Carmack et al., 1989, JGR, which are the refs. mentioned in Matsuoka.
REPLY: Indeed, we did have the reference to Carmack et al. 1989 but unfortunately misspelled the citation reference in LaTex (hence the **?**). We have kept the reference of Matsuoka et al because it uses the same dataset as our study. This is now corrected.

- Line 8, page 8: ...Only at station 394....
REPLY: Corrected.

- Line 33, page 10: which transect is meant?
REPLY: Changed to "all the ship-based transects (Fig. 1)"

- Line 14, page 14: ...which corresponds to the Mackenzie....
REPLY: Corrected.

- Line 17, page 14: ...to the northerly and rather weak....
REPLY: Changed to: "to the northerly and, then later, weak winds".

- Line 6, page 20: ...at a depth corresponding to an.....
REPLY: Corrected.

- Line 10, page 20: ...(Fig. 8d)....
REPLY: Corrected.

- Line 15, page 23: The reference Guay et al. is not cited in the manuscript - Line 25, page 24:
Timmermans et al. should appear after Stroeve et al.
REPLY: Thank you. We removed Guay et al. and moved Timmermans et al.

RESPONSES to the review of the manuscript:

"Patterns of suspended particulate matter across the continental margin in the Canadian Beaufort Sea", Jens K. Ehn, Rick A. Reynolds, Dariusz Stramski, David Doxaran, Bruno Lansard, and Marcel Babin

We greatly appreciate the constructive comments from both reviewers. Here we provide our detailed point-by-point responses and any description of action taken in regards to the comments by Referee #2. The Referees' comments are shown in regular font; our responses follow each comment in blue font.

**Response to Referee #2**

I was very interested to read this manuscript, whose main goal is to develop a relationship between beam attenuation data collected by transmissometers and suspended particulate matter and particulate organic carbon. Much archived transmissometer data exist for this region so finding such relationships could give valuable historic information on suspended particles. This is a very complex region and others have struggled to find statistically significant relationships between these properties in such complex regions. In general, I think that the authors did a convincing job of showing that there are robust relationships.

It seemed like a secondary goal of this manuscript was to describe the physical forcing responsible for the high or low particulate concentrations. Unfortunately, this is where I think this manuscript fell apart. The interpretation of the physical data was vague and few solid conclusions could be made from the very long discussion. I got the sense that the authors had limited understanding of the physical oceanography of this region.

Perhaps a better approach would be to choose only one physical process that is related to suspended particles. I see the main storyline of the manuscript as a comparison between the attenuation and bottle data. Proof of concept of this relationship could be shown by focusing on only one process, such as the Mackenzie River plume. There is room here for a very thorough study of this process and much new information could be gained on by coming up with concrete conclusions related to one physical process.

REPLY: We would like to thank Referee #2 for insightful comments and suggestions for revisions that we feel have helped improve this manuscript. Specifically, we have followed the Referee's suggestion and used oxygen isotope data to determine freshwater sources. We have also reduced the discussion and focused on the summer season by removing data from the fall 2007.

Regarding choosing only one physical process: As stated by the Referee, the Mackenzie Shelf is a very complex region. We find that it is difficult to interpret SPM distributions in regards to a single physical process, as they respond to multiple and interrelated physical and biological processes occurring at different temporal and spatial scales. The goal is to simply to describe the SPM distributions, and attempt to interpret them within the context of all these processes.

I recommend that this manuscript undergoes major revision before it can be reconsidered for publication. Below are several other concerns and suggestions that I have:

- Page 1, line 6 – Several times throughout the manuscript, the authors state that the surface layer is a mixture of sea ice melt and river runoff. While this may be true, the composition of the surface layer hasn't been quantified so the source of the particles in the freshwater can't be determined. The authors could attempt a freshwater budget as was done by Yamamoto-Kawai et al. (2008, doi:10.1029/2006JC003858) or they can acknowledge that they don't know the source of the freshwater or the particles therein.
REPLY: Thank you for this advice. We have added an analysis of oxygen isotope data to help link the observed particle characteristics to freshwater sources. Furthermore, CDOM fluorescence in Fig. 3 also qualitatively indicates where ice meltwater (low CDOM) vs. river water (high CDOM) dominated the surface layer. We feel that these additions have helped achieve a better focus and improved the manuscript.

- The Introduction is in general quite confusing. I think that a more clear description of the region would greatly help readers not familiar with this area. In addition, a stronger literature review of previous SPM and POC work in this region, and the mechanisms that transport these particles, would set the stage nicely for focusing this manuscript.
REPLY: We have improved the introduction with a broader description of the study area. In particular, we have added the description of the submarine valleys and the effects of wind-forcing on shelf-basin exchange. We have added references for past SPM and POC studies in the region and other relevant studies within the world's ocean.

- Page 2, second paragraph – This paragraph is quite confusing and needs more focus
REPLY: The introduction has been substantially reworked. As a part of this, we expanded paragraph 2 as shown below (with the italic portion representing the original 2nd paragraph) to create on concise paragraph on SPM sources:

"The significance of sediment discharge to the region is underscored by the fact that this sediment load from the Mackenzie River surpasses the combined load of all other major rivers discharging into the Arctic Ocean. Additional sediment sources of minerogenic sediment to the shelf include coastal and bottom erosion, and other rivers, which have been estimated to provide ~9 Tg per year (Macdonald et al., 1998). This makes the Mackenzie Shelf the most turbid shelf sea in the Arctic Ocean. Biological production, *by both marine phytoplankton and sea ice associated algae towards the end of the ice-covered season, is a major authochthonous source of biogenous sediments in the Beaufort Sea during summer (Forest et al., 2007; Forest et al., 2010; Tremblay et al., 2008), although the ice and turbid seawater are thought to greatly limit primary production on the Mackenzie Shelf (Carmack and Macdonald, 2002). The particulate sinking flux therefore comprises highly variable fractions of allochthonous and autochthonous origins (Sallon et al., 2011), making particle characterization in the area a complex task. The vertical export of autochtonous organic material to the deep waters of Canada Basin is found to be surprisingly small, however (Honjo et al., 2010). As the organic material reaching the deep ocean layers is thousands of years old it must be transported there laterally from the shelf or slope reservoirs of highly refractory material (Honjo et al., 2010). This highlights the*

*importance of understanding the distribution and lateral transport of particulate material from the shelf.*"

- Page 3, lines 14-15 – Why weren't lines 400 and 500 analyzed in this study?
REPLY: Unfortunately, there was not enough time during the MALINA expedition so that we needed to prioritize some stations and transects over others.

- Page 3, lines 25 -27 – What other depths were sampled in addition to the surface and SCM?
REPLY: There were large variations in other depths and their choice depended on the features seen on the CTD Rosette cast.

- Page 3, line 29 – What is an aliquot?
REPLY: Aliquot is a commonly used term defined as "a portion of a larger whole, especially a sample taken for chemical analysis or other treatment."

- Page 3, line 29 – What is considered a sufficient volume of water? Was this based on the time it took to filter or something else? What determines whether a duplicate or triplicate was sampled?
REPLY: Water from the Niskin bottles were in high demand on the cruise and used for analysis of many variables. We used as much water as was available and appropriate for our analysis. Seawater was passed through the filter to collect sufficient amount of particles but to avoid clogging of the filter. If the first sample filter required, for example, 4 L of water and only 6 L in total were available, then we did not collect a duplicate sample. We think this has been sufficiently described in the text, however, we added the information about "near-clogging" in the text.

- Page 4, lines 10 to 12 – Some other studies sampling SPM rinse the filters with ammonium bicarbonate or ammonium formate. Could the authors please explain why they didn't do this?
REPLY: We chose to rinse our filters with milli-Q to remove as much salt as possible but also avoid potential cell lysis. This is the most common procedure in SPM determination. Milli-Q water was readily available on the ship. We followed the method used in Babin et al. 2003 (as cited in the text), which essentially followed the JGOFS protocol described in Van der Linde, Protocol for determination of total suspended matter in oceans and coastal zones, Tech. Note I.98.182, Joint Res. Cent., Brussels, 1998. We also determined POC using the same sample filters.

- Page 5, line 8 – Please describe the interquartile range method, with a reference if applicable
REPLY: We have described it with the following sentence: "Time series of transmissometer data were also collected at selected depths and processed similarly to above, by taking the average of the interquartile range of the voltage values recorded over the periods when the rosette was stopped for water sampling during upcasts." There are no references for this method.

- Section 2.4 – Comparison of data between different transmissometers is notoriously difficult due to different calibration values and instrument drift. Is the use of dark voltage offset to allow for comparison of tranmissometer data between cruises? If so then could the authors please state the accuracy of this method, with references if applicable.
REPLY: The Vdark and Vref (representing particle-free seawater) are the calibration parameters

supplied with a C-Star and used to obtain beam transmittance and attenuation. The Vdark voltage offset is always done (but mostly directly within the Seabird CTD software). We have cited the Wetlabs C-Star user manual for how these are calculated, and the accuracy is stated by the manufacturer. It is worth noting that manufacturer calibrations are typically done at water temperatures of ~20 °C, but both Vdark and Vref values are sensitive to temperature. In brief, our processing of the C-Star raw voltage data is exactly the same as is typically done in the CTD software. However, as noted, there can be significant drift of the C-Star over time, and the ambient temperature can further affect the readings. To minimize these uncertainties, we have taken the approach to not to rely on the calibrated values (and here we note that the Vdark is of less importance compared to Vref in the relatively clear marine waters) but to determine them ourselves from the raw voltages measured at ambient temperature as described in the manuscript. For determining Vref  we have used the clearest waters that we observed in the Arctic Ocean during the field campaigns, rather than pure water in the lab.

- Page 6, line 7 – What depth were the other sensors located at on the mooring?
REPLY: These sensors were all associated with the Aanderaa RCM11 and thus located at the same depth of 178 m.

- Section 2.5 – Why were data from only CA05 shown? This mooring is at the edge of the Cape Bathurst upwelling region, which is not particularly representative of the region. Several other moorings have been deployed along the Canadian Beaufort during the study period, 2004 to 2009. Why was only this mooring selected to represent the region?
REPLY: This mooring was located on our line 100, at the shelfbreak, so it is representative of our observations. These data also show well the change in the direction of the current at 178 m associated with upwelling.

- Page 6, line 28 – where is the proof that there were strong easterly winds in June 2009?
REPLY: Figure 8a (now Fig. 12a) shows that the wind direction during June was persistently from northeast (along the shelf).

- Section 3.1.1 – Please add some references to the different water mass definitions.
REPLY: We think that Carmack et al (1989) is a pertinent citation for water mass definitions for the study region. We have added the following sentence: "The water mass definitions that ensue follow Carmack et al (1989) and are consistent with descriptions in Lansard et al. (2012) and Matsuoka et al. (2012)." We removed these citations from the next sentence. We have also added a reference to Jackson et al. (2015).

- Page 7, lines 5-6 – Please see Jackson et al (2015, 10.1002/2015JC010812 ) for information on Pacific winter water in this region
REPLY: Thank you for suggesting this reference. We have included it.

- Page 7, lines 10 – 17 – The c_p values in this paragraph don't appear to match those in Figure 2
REPLY: The range is correct. Note that the higher cp values are represented with white contour lines. The colour bar represents the full range.

- Page 7, line 18 – what does the 'strong chl-a fluorescence signal mean? Couldn't they be quantified by discrete chlorophyll samples?
REPLY: In principle, this could be done by 'calibrating' the fluorescence sensor with discrete chl-a measurements, but, as far as we know, this calibration has not been done on the data from the MALINA cruise. There are uncertainties associated with this calibration. We simply use the chl-a fluorescence (and CDOM fluorescence) in a qualitative sense to gain a better understanding of particle characteristics and patterns on the shelf. A 'strong chl-a fluorescence signal' simply means that the instruments detected higher fluorescence at these depths/locations than elsewhere, which is generally indicative of higher concentration of particles containing chl-a.

- Page 7, line 22 – What is the source of CDOM in Pacific Winter water? Perhaps more information can be added from Guegen et al., 2012 (doi:10.1016/j.dsr2.2011.05.004 )
REPLY: The paper by Matsuoka et al (2012) describes the CDOM observations during MALINA. Therefore, we think that the addition of more information overlapping with the Matsuoka et al. study is not necessary in our manuscript. However, the CDOM source is likely associated with the decomposition of organic materials in the Bering and Chukchi Seas.

- Page 8, line 10 – There is no information about the location or methods used for barge sampling
REPLY:  Barge sampling is described in Doxaran et al. (2012). We cited that study in that context. We have mentioned that barge stations were in the vicinity of the CCGS Amundsen, and also mentioned the transects to the Mackenzie River mouths.

- Page 8, lines 24 – 26 – Why is it not possible to measure PSD using the Coulter technique in low salinity, turbid waters?
REPLY:  The Coulter Counter counts and sizes particles suspended in an electrolyte by aspirating sample through a small aperture and recording the change in the electrical impedance as particles pass through the aperture. In our case, the electrolyte is seawater. Reductions in salinity decrease the conductivity across the aperture and increases the noise associated with the measurement; below a certain threshold of salinity the uncertainties of the measurement become untenable.

Although turbid waters may pose a challenge because of the need for coincidence correction (accounting for multiple particles passing simultaneously through the aperture), this can generally be handled through appropriate dilution of the sample. In the present case, the statement "low salinity, turbid waters" is made simply because the waters near the river mouth were associated with both low salinity and high turbidity, not because of any limitation in the technique associated with turbidity per se.

- Page 8, line 32 – I disagree with this statement. The relationships shown in Figure 4b are not very convincing. Is the relationship statistically significant?
REPLY: We have deleted the first sentence. We did not test the statistical significance.

- Page 9, lines 8-10 – What is the difference between the MALINA and Amundsen data?
REPLY: No difference. It was used to specify the contrast between sampling from Amundsen

and barge during MALINA. For clarity we have changed "CCGS Amundsen" to "ship-based sampling".

- Page 9, lines 18 -20 – Of these three regression analyses, why is only ii) shown in Figure 5?
REPLY: We decided to clarify the results of regression analysis in the revised manuscript by moving details of this analysis to the Supplementary Materials, and keeping only the most essential regression fits actually utilized in the manuscript.

- Page 9, line 26 – do the authors mean 'nonlinear power function' instead of 'nonlinear least squares regression'?
REPLY: Firstly, the description containing this sentence has been deleted and moved to the Supplementary Materials. Secondly, in the Supplementary Materials, we have now written: "Therefore, we selected the SPM vs. $c_p(660)$ relationship obtained from the power function fit using nonlinear least squares regression to ordinary (non-transformed) variables as the algorithm for estimating SPM in $[\text{g m}^{-3}]$ from $c_p(660)$ in $[\text{m}^{-1}]$ in the rest of this study".

- Page 12, lines 20 – 26 – Figure 8 is very unclear and possibly incorrect. I can't see the statements in this paragraph supported in Figure 8
REPLY: In our opinion, progressive vector plots are the best way to display the overall wind and current regimes. We have improved the clarity of the figure and we see no problems that could make it incorrect. We have marked the timing of the expeditions on all plots, removed the fall 2007 period, and removed the observations from 54 m depth in an effort to clarify the figure.

- Page 12, line 28 – I can't see the wind speeds in Figure 8a
REPLY: The colour of the progressive vector plot indicates wind speed.

- Page 12, lines 31-35 – Don't these lines contradict lines 11-13 on page 12?
REPLY: Yes, indeed. We did not mention the one week period of southeasterly winds during the last week of July. We have added the sentence "Winds turned to southwesterly for the last week of July with wind speed > 8 m s$^{-1}$".

- Section 3.3.3. This section need much more work. How was cross-shelf defined? What depths were influenced by cross shelf currents? How do we know that the currents observed at CA05 were representative of the rest of the region? How was a cross-shelf episode defined? I'm not entirely sure how this section is giving evidence of upwelling and relaxation
REPLY: As indicated cross-shelf current was defined by the direction of 300 degrees as indicated by the change in direction in the progressive vector current plot (original Fig. 8b, now Fig. 12b). We have not tried to expand this analysis of currents to cover the full Mackenzie Shelf region. This is beyond the scope of our study. More information on modelled currents can be found, for example, in Mol et al. (2018) which is cited. The increase in salinity and temperature during upwelling episodes are an indication of upwelling of deeper waters that are more salty and warm. The upwelling shelfbreak flow is also linked with seaward Ekman transport of surface waters during which the plume extends further north and northwest. This reverses during downwelling inducing winds or relaxation of upwelling. We show that the SPM patterns on the shelf reflect this circulation.

- Section 3.3.4 – I don't see the point of this section. What new information does it tell us about SPM and POC in the Canadian Beaufort Sea?
REPLY: Section 3.3.4 describes the geostrophic currents which were used to detect the shelf break jet and the overall circulation on the shelf. This information clearly has bearing on SPM distributions on the shelf. Beam attenuation values were noticeably elevated at the shelf break jet, so the jet plays a role in resuspending particles and/or keeping particles in suspension. The occurrence of the shelf break jet is an indication for downwelling flow (e.g. Dmitrenko et al. 2016; Forest et al. 2015). We argue that this plays a role in SPM patterns on the shelf.

- Page 13, line 29 – Could the authors please be more clear with where the current intensification is observed? I refer the authors to Forest et al (2015, http://dx.doi.org/10.1016/j.csr.2015.03.009 ) for discussion of other strong shelfbreak currents in the region
REPLY: We have rewritten this section and it is now incorporated in section 3.1 (page 8). We have included a reference to Forest et al 2015. We have been more specific by saying "along the Makenzie Shelf shelfbreak" instead of "at this location".

- Page 14, lines 10- 11 – What causes this clear water extension onto the shelf?
REPLY: The particles originate from the river and shelf bottom. Therefore, the absence of particles in the clear water layer reflects the water column structure and dynamics, with the river plume in a stratified surface layer from which particle settling is limited, and bottom resuspension reaching only a certain level. This leaves a clear layer in between. We have described this in the text on page 12 (4th paragraph).

- Page 14, lines 24-25 – Is there proof of downwelling return flow and after upwelling?
REPLY: Yes, the mooring record, its link to wind speed and direction, the geostrophic current sections, and SPM patterns are all evidence for Ekman upwelling/downwelling.

- Section 4 – I am unclear exactly how the physical observations described in section 3 lead to the listed conclusions in section 4. Much more work needs to be done to understand the physical processes before they can be related to the particle concentrations
REPLY: We feel that the addition to the revised manuscript of freshwater source analysis has strengthened this link. We have also removed data from the fall 2007, rearranged and tried to better focus the text in order to address this issue.

- Figure 1 – Please make the CA05 mark larger and easier to see o It is difficult to distinguish between the different colored stars
REPLY: We added a green arrow pointing to the CA05 mark. We have furthermore added a list of station locations in the Supplementary Materials.

- Figure 2 – Why is the very freshest water on the western shelf away from the Mackenzie River? It doesn't appear that this very fresh water is correlated with the highest attenuation o It would help the reader understand the text if the stations could be marked on these figures o What does the grey area mean?
REPLY: The grey area is below the bottom so no data at the depth exists. The wind direction in 2009 was such that the Mackenzie River plume was pushed eastward. This was also linked to upwelling, as can be seen from the current data. Attenuation values were elevated in the plume,

however, seasonal timing play a role here as, for example, the SPM values in July 2004 (Fig. 7g) were an order of magnitude higher than in August 2009 (Fig. 7h). In August 2009, the highest cp(660) on the shelf are indeed not in the freshest part of the plume, but towards the east. Interestingly, this is also linked to higher salinity. Thus, we draw the conclusion that the wind, the upwelling and tides (which are not discussed in detail) caused resuspension of sediment in the shallow eastern portion of the shelf. Note also that prior to MALINA, during the first part of July, there was a period of northerly winds that pushed the river plume along the shore towards east. Some of the suspended particles may have been remnants from the earlier eastward-moving alongshore plume.

- Figure 3 – Why are the error bars backwards?
REPLY: Figure 3 does not include error bars. If the reviewer is referring to the color bar scale, we have reversed the direction (lowest to highest values going from left to right) in the revised manuscript.

- Figure 4 – It would help the reader interpret this if boundaries were drawn around the 3 different defined areas o Figure 4b – I don't see very strong, statistically significant relationships here. Also, is the salinity from the same depth that the water was sampled from?
REPLY: We have redone this figure to show the POC/SPM boundaries of 0.06 and 0.25 between mixed and organic-dominated particle assemblages. 4b: We agree that there is not a strong statistical relationship here. The point of the figure is to show the range of PSDs. However, there is a strong relationship between the river runoff fraction and POC/SPM (as shown in new Fig. 5b).

- Figure 6 – It is difficult to see the writing of the different cruises
o The data look smoothed. Can the authors please state how they smoothed the data?
REPLY: We have increased the size of the figure. The data has not been smoothed by us. What is plotted is the data downloaded from Environment Canada (http://wateroffice.ec.gc.ca/) and the Canadian Ice Service.

- Figure 8 – I really struggled with this figure. It is difficult to interpret, has very small writing, and has a huge amount of information.
o The wind and current data in particular were difficult to distinguish. It was near impossible to see upwelling or downwelling as this figure was laid out
o I think that the depths in the temperature and salinity plots were mislabeled – the shallower water shouldn't be saltier
o There should be some explanation as to why such salty water was observed in figure 8d. Water of this salinity would have to be upwelled from several hundred meters depth, and a significant upwelling event would need to be evident in the wind data.
o I don't understand what the different colours mean in 8c
REPLY: We are sorry about the figure presentation and agree it is a complex figure with lots of information. In the revised manuscript, we have inserted a larger figure which is intended as a full page width figure. When the paper is prepared in final form for publication we will make additional improvements, if needed, to ensure that all details are clear. We have also simplified this figure by removing the extra depth of 57 m and the turbidity, to just focus on the 178 m data..

- It was not mislabeled. However, the plots were on different scales. This is no longer an issue.
- The timing of upwelling inducing winds (a) and upwelling events (b-d) are consistent. Same is true for downwelling events.
- Colours are current direction which is more easily seen in (b).

- Figure 9 – Please mark the mooring location on line 100 (CA05)
REPLY: Done
o I couldn't distinguish between the different contour lines
REPLY: Figure is now larger which hopefully helps.

o Please include the station locations
REPLY: We have not included station locations, however, the small black dots indicate the profiles taken at the station locations.

o The current values don't make sense to me. General definitions in oceanography are that northward and eastward currents are positive and southward and westward currents are negative. Having different definitions makes this figure very confusing
REPLY: Geostrophic current calculations can only give speeds that are perpendicular to the transect line. We could divide this speed up in U and V components, however, since our transect lines are nearly perpendicular to the shelf break, the calculated geostrophic currents represent along-shelf current magnitude. Note that this is consistent with what has been done in other studies (e.g., Forest et al., 2015).

%% Copernicus Publications Manuscript Preparation [...] Submissions
%% ~~
%% ~~
%% This template should be used for copernicus.c[...]
%% The class file and some style files are bundl[...]
Latex Package, which can be downloaded from the d[...]
webpages.
%% For further assistance please contact Copernic[...]
production@copernicus.org
%%
https://publications.copernicus.org/for_authors/manuscript_preparation.ht
ml

%% Please use the following documentclass and journal abbreviations for
discussion papers and final revised papers.

%% 2-column papers and discussion papers
\documentclass[journal abbreviation, manuscript]{copernicus}

%% Journal abbreviations (please use the same for discussion papers and
final revised papers)

% Advances in Geosciences (adgeo)
% Advances in Radio Science (ars)
% Advances in Science and Research (asr)
% Advances in Statistical Climatology, Meteorology and Oceanography
(ascmo)
% Annales Geophysicae (angeo)
% Archives Animal Breeding (aab)
% ASTRA Proceedings (ap)
% Atmospheric Chemistry and Physics (acp)
% Atmospheric Measurement Techniques (amt)
% Biogeosciences (bg)
% Climate of the Past (cp)
% Drinking Water Engineering and Science (dwes)
% Earth Surface Dynamics (esurf)
% Earth System Dynamics (esd)
% Earth System Science Data (essd)
% E&G Quaternary Science Journal (egqsj)
% Fossil Record (fr)
% Geographica Helvetica (gh)
% Geoscience Communication (gc)
% Geoscientific Instrumentation, Methods and Data Systems (gi)
% Geoscientific Model Development (gmd)
% History of Geo- and Space Sciences (hgss)
% Hydrology and Earth System Sciences (hess)
% Journal of Micropalaeontology (jm)
% Journal of Sensors and Sensor Systems (jsss)
% Mechanical Sciences (ms)

Latexdiff produced errors which I could not solve. Therefore, I used microsoft office to perform a comparison between the originally submitted LaTeX tex file and the revised manuscript tex file.

Deletions are in red with strikethrough
Additions in blue
Moved text in blue underlined and red doublestriked text

```latex
% Natural Hazards and Earth System Sciences (nhess)
% Nonlinear Processes in Geophysics (npg)
% Ocean Science (os)
% Primate Biology (pb)
% Proceedings of the International Association of Hydrological Sciences
(piahs)
% Scientific Drilling (sd)
% SOIL (soil)
% Solid Earth (se)
% The Cryosphere (tc)
% Web Ecology (we)
% Wind Energy Science (wes)

%% \usepackage commands included in the copernicus.cls:
%\usepackage[german, english]{babel}
%\usepackage{tabularx}
%\usepackage{cancel}
%\usepackage{multirow}
%\usepackage{supertabular}
%\usepackage{algorithmic}
%\usepackage{algorithm}
%\usepackage{amsthm}
%\usepackage{float}
%\usepackage{subfig}
%\usepackage{rotating}

\begin{document}

\title{Patterns of suspended particulate matter across the continental
margin in the Canadian Beaufort Sea

 during summer}

% \Author[affil]{given_name}{surname}
\Author[1]{Jens K.}{Ehn}
\Author[2]{Rick A.}{Reynolds}
\Author[2]{Dariusz}{Stramski}
\Author[3]{David}{Doxaran}
\Author[4]{Bruno}{Lansard}
\Author[5]{Marcel}{Babin}

\affil[1]{Centre for Earth Observation Science, University of Manitoba,
Winnipeg, Manitoba, Canada.}
\affil[2]{Marine Physical Laboratory, Scripps Institution of
Oceanography, University of California San Diego, La Jolla, California,
U.S.A.}
\affil[3]{Sorbonne Université, CNRS, Laboratoire d,ÄôOcéanographie de
Villefanche, Villefranche-sur-mer 06230, France.}
\affil[4]\affil[4]{Laboratoire des Sciences du Climat et de
l,ÄôEnvironnement, LSCE/IPSL, CEA-CNRS-UVSQ-Université Paris Saclay,
91198 Gif-sur-Yvette, France}
```

\affil[5]{Joint International ULaval-CNRS Laboratory Takuvik, Qu√©bec-
Oc√©an, D√©partement de Biologie, Universit√© Laval, Qu√©bec, Qu√©bec,
Canada.}

%% The [] brackets identify the author with the corresponding
affiliation. 1, 2, 3, etc. should be inserted.

\runningtitle{SPM in Canadian Beaufort Sea}

\runningauthor{Ehn et al.}

\correspondence{Jens K. Ehn (jens.ehn@umanitoba.ca)}

\received{}
\pubdiscuss{} %% only important for two-stage journals
\revised{}
\accepted{}
\published{}

%% These dates will be inserted by Copernicus Publications during the
typesetting process.

\firstpage{1}

\maketitle

\begin{abstract}
The particulate beam attenuation coefficient at 660 nm,
$c_\mathrm{p}(660)$, was measured in conjunction with properties of
suspended particle assemblages in August 2009 within the Canadian
Beaufort Sea continental margin, a region heavily influenced by
freshwater and sediment discharge from the Mackenzie River., but also by
sea ice melt. The mass concentration of suspended particulate matter mass
concentration (SPM) ranged from 0.04 to 140 g m$^{-3}$, its composition
varied from mineral to organic-dominated, and the median particle
diameter ranged determined over the range 0.7--120 $\mu$m varied from
0.78 to 9.45 $\mu$m, with the fraction of particles < 1 $\mu$m highest in
surface layerswaters reflecting the degree influenced by river water or
ice melt. A.  Despite this range in particle characteristics, a strong
relationship between SPM and $c_\mathrm{p}(660)$ was developedfound, and
used to determine SPM distributions across the shelf based on
measurements of $c_\mathrm{p}(660)$ taken during summer seasons of 2004,
2008, and 2009, as well as fall 2007.. SPM spatial patterns on the shelf
are explained by an interplay between wind forcing, river discharge, and
sea ice coverage resulting in three stratified shelf reflected the
vertically sheared two-layer estuarine circulation modes:and SPM sources
(i.e., fluvial inputs, bottom resuspension, and biological productivity).

[revised manuscript text omitted]
 \citet{Carmack_etal_1989} and are consistent with descriptions in \citet{Lansard_etal_2012} and \citet{Matsuoka_etal_2012}.
The salinity range between 30.7 and 32.3 PSU corresponds to the Pacific Summer Water mass, which  originates from waters flowing through Bering Strait during summer . Underneath, the Pacific Winter Water is characterized by  salinity between 32.3 and 33.9 PSU and typically found from $\sim$180 to 220 m depth \citep[e.g.,][]{Jackson_etal_2015}. This is followed by a transition to waters of Atlantic-origin with salinity > 34.7 and temperature above 0 $^{\circ}$C typically found between $\sim$220 and 800 m. Cold and dense deep water are found at greater depths and down to the bottom.

The relative contributions ($\%$) of the two sources to the freshwater content, i.e., meteoric water $f_{MW}$ and sea ice meltwater $f_{SIM}$, in the surface layer is shown by the contours in Fig. \ref{fig:SSURF}a. The percent values are calculated as follows: $f_{MW}/(f_{MW}+f_{SIM})\times100$. Apart from the Mackenzie River mouth, the freshwater in the surface layer was a mixture between sea ice melt and river runoff. River water prevailed along the coastline, while sea ice melt had a larger contribution further offshore. A larger river water fraction also extended further along the west coast with the northwest flowing river plume. In the upwelling region north of Cape Bathurst, river runoff and ice melt contributed about equal amounts to the relatively small freshwater content of $\sim$10 \%. The high ice melt proportions in excess of 80 \% were found in offshore waters with melting multiyear sea ice \citep{Belanger_etal_2013}.

Geostrophic currents for the cross-shelf sections 100, 200, 300, and 600 were calculated using temperature and salinity data from August 2009 CTD casts (Fig. \ref{fig:GEOSTROPHIC}). The reference depth, where the current velocity was assumed to be zero, was selected as 500 m, corresponding to a water mass originating in the Atlantic in which geopotential gradients are small \citep{McPhee 2013}. The sections reveal a westward mean flow of up to 9 cm s$^{-1}$ in the Canada Basin (Fig. \ref{fig:GEOSTROPHIC}b, c), which is consistent with the anticyclonic circulation of the Beaufort Gyre. Similarly, currents over the shelf were typically westward with speeds on the order of a few centimeters per second. A notable feature was the presence of the eastward flowing shelfbreak current centered between 100 and 150 m depth \citep{Pickart_2004}.
The shelfbreak current is an indicator for downwelling flow from the shelf to the basin \citep{Dmitrenko_etal_2016}.
Both \citet{Dmitrenko_etal_2016} and \citet{Forest_etal_2015} present mooring data collected at Mackenzie Shelf shelfbreak location showing events of wind-driven shelfbreak current intensifications (with flow up to 1.2 m s$^{-1}$ in January 2005) during downwelling favorable winds. However, to our knowledge, the current intensification along the Mackenzie Shelf shelfbreak during summer has not been shown in the literature to date. The mean easterly flow was around 3 cm s$^{-1}$ (Fig. \ref{fig:GEOSTROPHIC}a--c), which is consistent with the observations of \citet{Pickart_2004} for the summertime period along the Alaskan Beaufort shelfbreak. The section along line 600 in the Mackenzie Trough captured

an anticyclonic mesoscale eddy ($\sim$50 km diameter) which impacted the patterns of $c_\mathrm{p}(660)$ and chl-\textit{a} fluorescence (see below).

[revised manuscript text omitted]

For a detailed description of the particle size distribution (PSD) data measured during MALINA, readers are referred to \citet{Reynolds_etal_2016}. Here, we provide an overview of the spatial distribution of the PSD by calculating the volume fraction of particles less than 1 $\mu$m in diameter $D$ to the total particle volume between 0.7 $\mu$m and 120 $\mu$m. A notable feature in the particle volume distribution, $V(D))$, was the presence of high concentrations of ~~small particles with diameters D < 1 ~μm4bSimilar.with the Coulter technique. Interestingly, the offshore samples from stations 110 (surface) and 240 (55 m depth) with low POC/SPM ratios were also associated with high concentrations of < 1 $\mu$m particles, which is consistent with multiyear ice (suspension freezing) and shelf bottom (resuspension) origins. The volume fraction of < 1 $\mu$m particles for the surface water samples at station 240 and the nearby station 235, was 0.12 and 0.18, respectively. At these stations, the near-surface salinity was close to 27.5 PSU.~~due to limitations of the Coulter technique. Station 110 stands out among line 100 stations with < 1 $\mu$m volume fractions of 0.29 at the surface (salinity of 29.1 PSU) and 0.09 at 60 m depth (31.6 PSU).

~~In general, the samples for which PSD was measured can be separated based on salinity of the sampled water. The samples with salinity < 30 PSU were collected in the surface layer while the samples with salinity > 30 PSU were collected at a depth of 20 m or deeper. Percentile statistics of V(D) show that small sized particles dominated the particle assemblages within surface waters. The subsurface waters were characterized by larger variability with generally increased contribution of larger particles (data not shown). The particle diameters corresponding to the 10th, 50th, and 90th percentiles of V(D), i.e., $D_V^{10}$, $D_V^{50}$, and $D_V^{90}$, respectively, can be summarized as follows: $D_V^{10}$ was~~

To conclude, from the data in Fig. \ref{fig:POC2SPM} we find that (1)
when $f_{MW}$ increased in the surface waters of southeast Beaufort Sea,
POC/SPM ratios decreased while the < 1 $\mu$m particle fraction
increased, and conversely (2) when the $f_{SIM}$ influence increased,
POC/SPM increased while the < 1 $\mu$m particle fraction decreased in
surface waters.

\subsection{Relationships between SPM, POC and particulate beam
attenuation}
 \label{relationship}
The SPM of the samples examined during the MALINA cruise ranged from 0.04
to 140 g m$^{-3}$ with associated POC from 0.007 to 1.5 g m$^{-3}$
\citep{Doxaran_etal_2012}. Organic-dominated and mixed particle
assemblages were predominant in the portion of the data set obtained from
ship-based sampling, with SPM extending to 5.6
g m$^{-3}$. The mineral-rich particle assemblages were more common in
turbid estuarine waters located close to shore (Fig.
\ref{fig:POC2SPM}a). These waters were sampled using a small barge with
an optical package that included a Wetlabs AC-9 meter
\citep{Doxaran_etal_2012}, but no Wetlabs C-Star 660-nm. The nearest
wavelength band on the AC-9 was 676 nm. It thus provided
$c_\mathrm{p}(676)$. Note that much higher sediment loads were observed
in the region in the past. For example, \citet[][their Fig.
10]{Carmack_Macdonald_2002} reported on near bottom SPM values of 3000 g
m$^{-3}$ due to resuspension of bottom sediments during a storm in
September 1987.

Data from all 28 stations with coincident measurements were used in the
development of relationships between $c_\mathrm{p}(660)$ and SPM and
between $c_\mathrm{p}(660)$ and POC. The particulate beam attenuation
coefficient correlated well with both SPM and POC (Fig.

\ref{fig:FIT}a, b).
\begin{equation}
\mathrm{SPM} = 1.933 \: c_\mathrm{p}(660)^{0.9364}
\end{equation}

and
\begin{equation}
\mathrm{POC} = 0.2071 \: c_\mathrm{p}(660)^{0.6842}
\end{equation}

[revised manuscript text omitted]

~~which impacted the patterns of $c_\mathrm{p}(660)$ and chl~\textit{a}~~

 on the Mackenzie Shelf (Fig. \ref{fig:VECTOR}d).

Episodes of high along-shelf current speeds (dark green in Fig. \ref{fig:VECTOR}c), such as at the end of the MALINA expedition in late August 2009, but also in November 2008, February, May and July 2009, were generally associated with reductions in salinity and temperature at the CA05 mooring, and perhaps also linked to shelfbreak transport of SPM with downwelling flow.

\subsection{Effects of river runoff and sea ice melt on SPM distributions on the }

This section is focused on discussion of SPM fields (Fig. 10), derived from $c_\mathrm{p}(660)$ profiles using Eq. 2, along with supporting temperature and salinity fields (Fig. 11). We recall that both Eq. 2 and Eq. 3 are valid for $c_\mathrm{p}(660)$ values up to 3.1 m$^{-1}$ (Fig. 5). Thus, this excludes the most mineral dominated waters on the shelf with SPM over 5.6 g m$^{-3}$ and POC over about 0.5 g m$^{-3}$. Within the valid range the presented SPM [g m$^{-3}$] can be converted to POC [g m$^{-3}$] according to POC $= 0.1279$ SPM$^{0.7307}$, which was derived from the regression analysis of POC vs. SPM data.

\subsubsection{Clear waters}
SPM ranging between 0.04 and 0.06 g m$^{-3}$ was found in offshore waters in each of the three transect lines measured during June-August (Fig. 10). The low SPM values were especially widespread in August 2009 (MALINA) with wedges of very clear water extending far onto the shelf. The corresponding POC ranged from 0.01 to 0.02 g m$^{-3}$. The extension of clear waters onto the shelf as a wedge between the surface plume and the turbid near bottom layer has been described by \citet{Carmack_etal_1989}. It appears that neither particle settling from the surface plume nor the resuspension of bottom sediments were sufficient in August 2009 to increase these clear water values of $c_\mathrm{p}(660)$ above those found in deep basin surface waters. The landward extension of the clear water layer was particularly noticeable on line 600 (Fig. 10j) which corresponds with the Mackenzie Trough (a submarine canyon), the main river channel and the most distinct surface plume feature of the transects. The conditions during MALINA differed from the previous years particularly in terms of sea ice coverage (Fig. 7b). The break-up of the landfast shelf} \label{ice on the shelf occurred late and ice floes were not readily transported away from the shelf due to the northerly and later weak winds. Furthermore, multiyear ice extended further south compared to previous years considered in this study (Fig. 7). As surface salinity remained low for the length of the line 300 (Fig. 11h), the melt water from this ice appears to have influenced the low SPM levels in the shelf waters by increasing the stratification that reduced mixing and by hindering the spread of the particle-rich river plume.

\subsubsection{Effect of ice -melt on SPM distribution}
Comparatively high levels of SPM were found along line 300 (Kugmallit Valley) near the shelf bottom in August 2009 with particularly high values extending across the shelf (Fig. 10h). Such SPM patterns are indicative of downwelling return flow from the shelf after upwelling-inducing wind conditions relaxed. These near-bottom concentrations match those observed during the fall 2007 (Fig. 10f) under high winds (Fig. 8) and brine release from forming ice (Fig. 7) and generally higher salinity shelf waters (Figs. 11b, f). The presence of sea ice and its meltwater on the shelf during August 2009, as seen from the low surface temperatures and salinities at $\sim$70.9$^{\circ}$ N (Fig. 11h), can explain the containment of the spreading of the plume along line 300 (Fig. 10h). High particle settling rates from a slow moving or stagnant river plume may in turn explain the high near bottom SPM which then could be transported along the shelf bottom with the return flow of the upwelled waters.

A contrasting situation is provided by the conditions observed along line 300 during June-July 2008 (IPY-CFL study) (Figs. 10g and 11g). During the IPY-CFL, ice coverage on the shelf was low (Fig. 6b) and upwelling-inducing winds prevailed throughout June and early July (Fig. 8}

). Consequently, the two compared SPM sections along line 300 differed markedly (Fig. 10g, h). As seen in Fig. 10g, in 2008 the turbid surface river plume spread northward past the shelfbreak. This buoyancy-driven flow was likely enabled by the absence of ice melt water. The near-bottom turbidity was low likely owing to conditions resulting from the notable upwelling event evidenced by the high salinity of the shelf bottom water and the extent of the turbid surface plume (see Fig. 11g).

\subsubsection{River plume variability}
Wind-forcing largely controls the flow direction of the Mackenzie River plume. Due to the size and shape of the Mackenzie Shelf, the most likely direction for the Mackenzie River plume to spread significant distances past the shelfbreak is to the northwest \citep{Doxaran_etal_2012}. During the spring freshet in June 2009, sustained easterly along-shelf winds caused the flaw-lead polynya to widen along the Mackenzie Shelf and a turbid river plume extended northwestward from the landfast ice to the pack ice (Fig. S2). The MALINA sampling occurred during a time of transition from a northwestward plume (during easterly winds) towards a Coriolis-forced right turning plume flowing eastward along the coast. Plumes of both directions are visible in MODIS satellite images for the period of the MALINA expedition \citep{Doxaran_etal_2012, Forest_etal_2013}. By 26 July 2009, the plume was clearly seen extending out past the tip of Cape Bathurst. The sampling along lines 600 and 700 was conducted during the first half of August 2009, following a two-week period of easterly winds (Fig. 8a). By 26 July, the plume was clearly seen extending out past the tip of Cape Bathurst.\ref{fig:VECTOR}a). By mid-August only very weak features remained from the northwestward plume. Notably, both river discharge and ice concentrations on the shelf were reduced by half during the period of one month (Fig. 6).

\ref{fig:RD-SIC}).

Figures 10i, j,\ref{fig:SPM}g, h and 11i, j\ref{fig:TS}g, h show the river plume extending northwest along the Mackenzie Trough (line 600). The Mackenzie River plume occupied an about 15 m thick layer at the sea surface both in July 2004 and August 2009. A sharp decrease in SPM was found immediately below this layer. The surface plumes were accompanied byhad low salinitiessalinity, high meteoric water fractions (Table 1 and, at least for 2009,  Fig. \ref{fig:SSURF}a), and high CDOM fluorescence (Fig. 3f)\ref{fig:FLUO}f), at least in 2009, and a high < 1 $\mu$m particle volume fraction (Fig. \ref{fig:POC2SPM}c, d), indicating a riverine origin \citep{Matsuoka_etal_2012}.. Interestingly, particle concentrations differed markedly for the two years compared. In 2004, high levels of SPM extended the full length of the transect with values reaching 4 g m$^{-3}$ as far as 70$^{\circ}$ N. In contrast, in 2009 the SPM values observed in the plume were only about 10 \% of the 2004 values but still distinctly noticeable because the plume overlaid a layer of very clear water. Also, the waters beneath the river plume in 2004 were

significantly more turbid compared to 2009, probably due to settling of particles from the plume.

Although the timing of the transect measurements in 2004 and 2009 was a month apart, overall conditions on the shelf were not markedly different. Easterly winds were weak in both cases (Fig. 8),\ref{fig:VECTOR}), ice coverage on the shelf was 30--40 \%, and the river discharge was $\sim$13,000-14,000 m$^3$ s$^{-1}$ during both years (Fig. 6).\ref{fig:RD-SIC}). Moreover, the cross sections along lines 100 (Fig. \ref{fig:SPM}a, c) and 300 (Figs. 10 and 11 (Fig. \ref{fig:SPM}d, f) show very similar features and particle concentrations during the two years. The differences between the two situations can be attributed to the seasonal timing. The 2004 transects were measured in early July soon after the break-up of the landfast sea ice cover and the surge of backed-up river waters across the delta and estuary. In contrast, the 2009 measurements were conducted much later in the season after landfast ice break-up. Consequently, in 2004 the surface plume was likely conditioned by a greater initial SPM discharge at the river mouth and by a higher momentum compared to 2009 so that it was capable of keeping more particles in suspension for a longer distance, including larger-sized particles if present. MODIS imagery of sea-surface temperature for 2 July 2004 (Fig. S3 in Supplementary Material) highlights this river plume inertia.

\subsubsection{Surface versus near-bottom cross-shelf SPM distributions} Comparatively high levels of SPM were found along line 300 (Kugmallit Valley) near the shelf bottom in August 2009 with particularly high values extending across the shelf (Fig. \ref{fig:SPM}f). On line 600 (Fig. \ref{fig:SPM}h), a nepheloid layer with SPM > 0.001 g m$^{-3}$ formed near the 33.1 PSU isohaline at $\sim$100 m depth. It was accompanied by a strong chl-\textit{a} fluorescence signal (Fig. \ref{fig:FLUO}e). Elevated near-bottom and shelfbreak SPM values were also observed during CASES and IPY-CFL (Fig. \ref{fig:SPM}d, g). Such SPM patterns are indicative of downwelling return flow from the shelf after upwelling-inducing wind conditions relaxed. The presence of sea ice and its meltwater on the shelf during August 2009, as seen from the low surface temperatures and salinities at $\sim$70.9$^{\circ}$ N (Fig. \ref{fig:TS}f) and high meltwater fractions (Fig. \ref{fig:SSURF}a and Table 1), can explain the containment of the spreading of the plume along line 300 (Fig. \ref{fig:SPM}f). High particle settling rates from a slow moving or stagnant river plume may in turn explain the elevated near bottom SPM which then could be transported along the shelf bottom with the return flow of the upwelled waters.

A contrasting situation is provided by the conditions observed along line 300 during June-July 2008 (IPY-CFL study) (Figs. \ref{fig:SPM}e and \ref{fig:TS}e). During the IPY-CFL, ice coverage on the shelf was low (Fig. \ref{fig:RD-SIC}b) and upwelling-inducing winds prevailed throughout June and early July (Fig. \ref{fig:VECTOR}a). Consequently, the two compared SPM sections along line 300 differed markedly (Fig. \ref{fig:SPM}e, f). As seen in Fig. \ref{fig:SPM}e, in 2008 the turbid surface river plume spread northward past the shelfbreak. At the same time, the near-bottom turbidity was low likely owing to conditions resulting from upwelling, evidenced by the high salinity of the shelf

bottom water and the extent of the surface plume (see \ref{fig:TS}e). This offshore surface flow was made possible by the absence of sea ice and ice meltwater (buoyancy forcing) and wind-driven Ekman transport.

The low SPM values were especially widespread in August 2009 (MALINA) with wedges of very clear water extending far onto the shelf. The conditions encountered during MALINA differed from expeditions in previous years  particularly in terms of sea ice coverage (Fig. \ref{fig:RD-SIC}b). The break-up  ~~Measurements in October 2007 showed a well-mixed upper layer of $\sim$30 m with the highest observed salinities and lowest temperatures on the shelf (Figs. 11b, f). These high salinities were caused by upwelling that was forced by strong easterly winds (Fig. 8; see also \citeauthor{Tremblay_etal_2011}, \citeyear{Tremblay_etal_2011}), but were likely also related to new ice formation that was taking place in shelf waters (Figs. 6 and 7). To illustrate this point, a 30-PSU salinity of a 20-m deep water column would increase by only 0.3 PSU from salt rejected by the formation of 0.3 m thick sea ice (World Meteorological Organization classification for the maximum thickness of the young ice type). Therefore, it is unlikely that sea ice formation alone could account for the observed high salinity.~~

~~Similarly to the physical properties, the SPM estimates from $c_\mathrm{p}(660)$ were well-mixed in shelf waters with estimated values reaching 4 g m$^{-3}$ (Fig. 10b, f). Despite the overall higher salinity of the water column in October compared to summer, a halocline was present at 30 m depth (Fig. 11f) beneath which high SPM levels extended towards the shelfbreak. This is an indication of cross-shelf transport of sediment near the bottom. The importance of the release of high-density brine from sea ice formation to particle transport across the Canadian Beaufort shelfbreak was discussed by Forest et al. (2007).~~

~~In October 2007 a 40--50 m thick layer of the upper water column with SPM in the range 0.40--0.55 g m$^{-3}$ extended the full length of the still ice-free line 100 (Fig. 10b). These are the highest values seen for surface waters within the Amundsen Gulf in our study. The source of these particles in the Amundsen Gulf is difficult to trace. Winds were sufficiently strong to cause resuspension of SPM on the shelf and other nearby coastal areas. However, the wind direction was easterly (Fig. 8) such that surface waters on the Mackenzie shelf and in Amundsen Gulf flowed mainly northwest, i.e., away from Amundsen Gulf (measured with acoustic current profiler on mooring CA05; data not shown). In freezing waters the $c_\mathrm{p}(660)$ signal could have been affected by the formation of frazil ice (small ice crystal particles). However, in this case this was not likely to happen because the surface layer along line 100 remained 0.5--1.5  $^{\circ}$C above the freezing point even though ice was forming on the shelf and along the coast (Fig. 7).~~ of the landfast ice on the shelf occurred relatively late and ice floes were not readily transported away from the shelf due to the northerly and, then later, weak winds (Fig. \ref{fig:VECTOR}a). Furthermore, multiyear ice extended further south compared to the two other years considered in this study (Fig. \ref{fig:CIS}).  At around 70.5$^{\circ}$ N on line 300, which coincides with northward extent of the river plume and rapid decrease in water column SPM levels (Fig. \ref{fig:SPM}f), the surface

salinity decreased below 27 PSU and temperature was <5 $^{\circ}$C (Fig. \ref{fig:TS}f) with over >70 \% $f_{SIM}$ fraction of the freshwater (Fig. \ref{fig:SSURF}a). As sea surface salinity remained low for the length of line 300 (Fig. \ref{fig:SPM}f), we argue that the meltwater from this ice influenced the low SPM levels in the shelf waters by increasing the stratification, reducing vertical mixing, and hindering the northward spread of the particle-rich river plume.

Another contrasting situation is seen in the Amundsen Gulf along line 100 (Fig. \ref{fig:SPM}a--c) where differences in conditions between the years can be explained by the presence or absence of sea ice, and the history of wind forcing as it relates to SPM transport from the shelf. Whereas ice free and comparatively clear surface waters were present in 2008 (Fig. \ref{fig:SPM}b), turbid (i.e., high $c_\mathrm{p}(660)$ and SPM) surface waters extended across Amundsen Gulf in 2004 and 2009 (Figs. \ref{fig:SPM}a, c), and the surface was furthermore partially ice covered in June 2004 (Fig. \ref{fig:CIS}a). The temperature and salinity fields, however, showed only modest differences between conditions in 2004, 2008, and 2009 (Fig. \ref{fig:TS}a--c). This suggests that the turbid surface waters in 2004 and 2009 were caused by the presence of shelf waters with particles originating from the Mackenzie River and/or via resuspension of shelf sediments. This is corroborated by the observed high meteoric water fractions in 2004 and 2009 (Table 1), and the high fraction of < 1 $\mu$m particles in the surface waters in 2009 (Fig. \ref{fig:POC2SPM}d). The equally fresh but clear surface layer in July 2008, after a long period of easterly winds (Fig. \ref{fig:VECTOR}a) and consequent westward circulation on the shelf \citep{Mol_etal_2018}, was however associated with sea ice meltwater with relatively low concentration of particles. The observations that $f_{MW}$ at stations 110 and 140 in July 2008 (IPY-CFL) were of similar magnitude to those observed during CASES and MALINA may be an indication of the importance of resuspension in the supply of SPM to surface water.

\citet{Tremblay_etal_2011} ~~reported on the upwelling of nutrients to reach the highest concentration of nitrate (16.8 ${\mu}$M) ever observed in the region on the shelf northwest of Cape Bathurst, which caused an increase in primary production. These high nitrate values did not extend far past the shelfbreak and remained low across Amundsen Gulf. Although, chl-\textit{a} fluorescence data for the surface waters indicated the elevated concentrations of phytoplankton \citep{Tremblay_etal_2011}, it is not possible to conclude that phytoplankton concentrations were sufficient to explain the high $c_\mathrm{p}(660)$.~~

 discussed the conditions in 2008, as well as nutrient dynamics, leading up to the high primary productivity observed in the Amundsen Gulf during the summer of 2008. The productivity of the SCM is generally proportional to the concentration of chl-\textit{a} and limited by light and nutrient availability \citep{Martin_etal_2010}.

\citet{Tremblay_etal_2011} proposed that the unusually early clearing of sea ice in 2008 was ~~enhanced by the upwelled nutrient-rich waters and, at the time of our measurements, biogenic material was being transported seaward in an intermediate nepheloid layer across the shelfbreak at 50--70 m depth \citep[Figs. 3 and 10;][]{Tremblay_etal_2011}. The shelf circulation at play makes it conceivable that the transport of biogenic material produced on the shelf, including resuspension of settled particles originating from an earlier bloom (e.g. ice algae), could potentially influence the formation and maintenance of the SCM in the off-shelf region. Thus, the study of shelf-basin exchange processes leading to the subsurface transport of nutrients and biogenic material may be of biological importance to improve the understanding of pelagic-benthic coupling in the region.~~

the key factor in increasing the subsurface light availability and primary productivity. However, the influence of the optical water clarity of the surface water layer was not considered. For example, Figs. \ref{fig:SPM}a--c reveal that in July 2008 , beneath the low turbidity surface layer, a higher SPM in the SCM centered at the 31.5 PSU isohaline (~50 m depth) was observed compared to June 2004 and August 2009

when surface water layers ~~extended across Amundsen Gulf in 2004, 2007 and 2009 (Fig. 10a, b, d), and the surface was partially ice covered in June 2004 (Fig. 7a). Time of year and consequent differences in river discharge and ice conditions are naturally expected to influence the size of the Mackenzie River plume. The temperature and salinity fields, however, show only modest differences between conditions in 2004, 2008 and 2009 (Fig. 11a, c, d). This suggests that the fresh and turbid surface layers present in 2009 were caused by the spreading of freshwaters affected by the Mackenzie River, while the equally fresh but clear layer in 2008 was associated with sea ice melt water. The difference in conditions between the years may thus be explained by the presence or absence of sea ice, and generally by the history of wind forcing, and how these two factors affect the spreading of the river plume.~~

layers are produced primarily by resuspension of bottom sediments settled onto the shelf or slope, and provide evidence for the transport of suspended particles and water away from the shelf. It is important to differentiate these layers from the mainly locally formed subsurface chl-\textit{a} maximum (SCM) layer that is commonly present in the Canadian Beaufort Sea \citep{Martin_etal_2010, Tremblay_etal_2011}. As the SCM seems to intersect with the shelf bottom (Fig. 3), the presence of relatively high chl-\textit{a} concentrations within subsurface nepheloid layers may however conceal the presence of minerogenic particles at the same depth.

On line 600, two subsurface nepheloid layers (in addition to the surface river plume) extending from the shelf at depths of 100--130 m and 200--250 m, were observed in 2004 and 2009 (Fig. 10i, j). These two layers appeared to form near where the 33.1 PSU isohalines intersected the shelf seafloor and immediately above and below a slightly less sloping section of the Mackenzie Trough bottom (Fig. 11i, j). However, only the upper layer was accompanied by relatively high chl \textit{a} fluorescence (Fig. 3e). The depths of 100 m and greater are beneath the euphotic layer rendering primary production negligible. Thus, these chl \textit{a} containing particles likely represent transported particles that originated in shallower shelf waters.

Numerous intermediate nepheloid layers (INLs) are seen in the upper 500 m of the water column throughout the Amundsen Gulf (Figs. 10a-d) and extending into Canada Basin (Figs. 10e-j). The cast-to-cast variability in the depth location of these INLs is large (Fig. 12) making it difficult to trace the shelf/slope origin of the INLs in this dynamic system. Generally, the SPM of INLs in offshore waters was an order of magnitude smaller than in the benthic nepheloid layer (BNL) on shelf and particle concentrations decreased with distance from the shelf.

Beneath 500 m depth, the vertical profiles of SPM still showed numerous inversions (Fig. 12). Generally, however, the particle concentration at specific depths decreased as bottom depth increased as it also relates to the distance from the shelfbreak. This decrease is approximately exponential with distance from the shelfbreak. In waters less than 3000 m deep located on the continental slope and rise, the SPM began to increase from about the mid depth of the water column which had the clearest waters. The thickness of these BNLs ranged from $\sim$200 m (station 340) to over 1000 m (Fig. 12). Past the 3000 m bottom depth, BNLs were essentially absent with the clearest waters found close to the bottom as may also be the case for the Canada Basin abyssal plain \citep{Hunkins_etal_1969}. Near-bottom SPM values based on $c_\mathrm{p}(660)$ were $\sim$2$\times$10$^{-3}$ g m-3 at the station CB 27, and decreased to $\sim$1$\times$10$^{-3}$ g m$^{-3}$ at 3500 m at CB 21 (74.0042$^{\circ}$ N, 139.8699$^{\circ}$ W, i.e., 113 km north of CB 27) on 9 October. After detaching from the BNL on the slope, INLs were advected along isopycnal surfaces. With distance, INLs became thinner with lower SPM owing to lateral spreading to cover larger areas, mixing at layer boundaries, and settling of particles.

It is thus of interest to investigate in more detail two obvious INLs seen in Fig. 12a. Station 530 showed an INL at ~1200 m depth with SPM of

$\sim$0.030 g m$^{-3}$. SPM in the overlying water was 0.021 g m$^{-3}$. It is clear that the INL must have been detached and advected from the bottom layers on the slope (perhaps near to adjacent stations 550 or 440 with a near-bottom SPM of ~0.050 g m$^{-3}$ at 1050--1100 m). Interestingly, station CB-27 located at 73$^{\circ}$ N showed a similar INL with the SPM maximum of 6.9$\times$10$^{-3}$ g m$^{-3}$ at 1170 m where the potential density anomaly, $\sigma_\theta$, was 28.032 kg m$^{-3}$. However, neither CB-23 nor CB-31 located to the east had INLs at that isopycnal. Over the distance of 240 km separating the stations 530 and CB-27, the SPM decreased by about 0.023 g m$^{-3}$, which corresponds to about 1$\times$10$^{-4}$ g m$^{-3}$ km$^{-1}$.

A notable INL at stations CB-23, CB-27, and CB-21 was spreading in the layer immediately below the isopycnal surface where the potential density anomaly $\sigma_\theta$ reached 28.096 kg m$^{-3}$ or the salinity reached 34.956 (Fig. 12). The depth where this occurred varied between the stations. Beneath this interface the potential temperature was uniform with depth, thereby marking a transition to the adiabatic Canada Basin bottom water layer \citep[e.g.,][]{Timmermans_etal_2003}. Assuming that the particles in the INL were from the bottom layer of CB-31 (~1920 m depth with $\sigma_\theta$ = 28.093 kg m$^{-3}$), then the transport of particles from the bottom of station CB-31 to the INL at station CB-23 requires a 560 m increase in depth over a 100 km distance, which equals a sinking rate of 5.6 m km$^{-1}$. Such transport of particles crosses isopycnal surfaces, suggesting the predominant role of particle settling in addition to advective transport. Furthermore, following the BNL to station 330 from CB-31, the BNL depth increased by 900 m over the distance of 88 km such that the descent rate was 10.2 m km$^{-1}$. This decrease in the rate of sinking with distance from the slope may illustrate the process by which the larger/heavier particles sink faster, or are broken down by biological processes, and are gradually removed from the nepheloid layer until only smaller particles remain in suspension. Some smaller sized particles transport of SPM in surface plumes may additionally have detached from the BNL forming the numerous INLs as observed, for example, at station CB-31 at depths between 1100 and 1900 m.

The maximum SPM within the INL at station CB-23 was 0.0126 g m$^{-3}$ at 2470 m depth. At CB-27 the maximum was 8.2$\times$10$^{-3}$ g m$^{-3}$ at 2600 m (Fig. 12). The SPM levels above the INLs (with $\sigma_\theta$ = 28.095) were 0.010 and 0.027 g m$^{-3}$, respectively. Given that the INL depth increased by 130 m over the 128 km distance that separated the two stations, the INL descent rate was about 1 m km$^{-3}$. A thinner (50 m thick) and weaker INL with a maximum SPM of 3.2$\times$10$^{-3}$ g m$^{-3}$ at 2656 m was observed at CB-21 (Fig. 12d). The INL descent rate over 113 km distance between CB-27 and CB-21 was about 0.5 m km$^{-1}$. Because the maximum SPM at these INLs occurred at the same $\sigma_\theta$, this descent of the INLs was determined by the water mass structure, however, the decrease of SPM within the maxima reflect processes such as spreading, settling, aggregation, scavenging, and mixing of particulate matter. The SPM in the INL was found to decrease linearly with the square root of distance from the shelf while background SPM decreased exponentially with distance. A correlation analysis based on data from station 330 and the four stations from CB-31 through CB-21,

influence primary productivity in Amundsen Gulf by reducing light
penetration.

\conclusions

[revised manuscript text omitted]

~~The presence of the SCM layer \citep{Martin_etal_2010} at the base of the Pacific Summer Water mass was a consistent feature in the southern Beaufort Sea but was also observed to intersect with the BNL on the shelf (Fig. 3). Apart from the SCM, the depth locations and particle concentrations of INLs were found to be highly variable and numerous in top 500 m of the water column, highlighting the complex conditions responsible for their formation. The high SPM seen on the shelf did not extend far past the shelfbreak except in a westward flowing river plume (Fig. 10; see also Fig. S2 in Supplementary Material). Thus, subsurface sediment transport beyond the shelfbreak must occur in a near bottom BNL which detaches into INLs at specific depths as determined by physical processes and particle characteristics. Further research is required to explore this observation in detail. Past the shelfbreak, the SPM at specific depths generally decreased with distance from the shelfbreak and with increasing depth. The $c_\mathrm{p}(660)$ profiles in Canada Basin waters agree with the two types of profiles described in \citet{Hunkins_etal_1969}, first, in waters with bottom depths less than about 3000 m the SPM had minimum values roughly at mid-depths of the water column and then increased towards the bottom forming a c-shaped profile, and second, in waters exceeding the 3000 m depth the SPM reached minimum values near the bottom. The deepest INL (below which no INLs were seen) extending to the Canada Basin abyssal plain was observed at the 2500--2600 m depth at the top of the adiabatic Canada Basin bottom water layer \citep{Timmermans_etal_2003}.~~

As the Arctic continues to warm, the open water season is expected to become increasingly longer and the extent of multiyear ice further decline \citep{Stroeve_etal_2014}. The reduction in ice coverage in the Beaufort Sea implies an increase in SPM dynamics on the continental margin due to the associated changes in wind forcing and river discharge \citep{Carmack_etal_2006}. Greater wind and wave forcing on open waters is expected to increase particle concentrations on the shelf. However, the presence of both clear intermediate waters and highly turbid bottom waters observed on the shelf in this study highlighted interesting linkages to the effect of sea ice on river water and particle transport on the shelf, which need further study. The processes that operate within subsurface layers and ice-covered waters cannot be deciphered through satellite remote sensing, so their quantification requires in-situ monitoring. Optical beam transmission is a simple yet efficient tool for mapping SPM distributions. The relationship between SPM and $c_\mathrm{p}(660)$ developed in this study can be applied to past and future transmissometer observations to monitor changes in SPM. Vertical measurements reaching all the way to the seafloor would be very beneficial when attempting to determine lateral SPM transport. This is typically not done due to the risk to the instruments. Furthermore,

ongoing research that considers current speeds together with particle size distributions are needed in order to shed more light on particle transport and settling processes across the Beaufort Sea continental shelf, and slope and rise, which are experiencing considerable change in response to river discharge, sea ice coverage, and wind forcing. The results from this study can help evaluate numerical models which may be used to investigate sensitivities of SPM dynamics associated with oceanographic and forcing conditions on the Mackenzie Shelf.

%% The following commands are for the statements about the availability of data sets and/or software code corresponding to the manuscript.
%% It is strongly recommended to make use of these sections in case data sets and/or software code have been part of your research the article is based on.

% \codeavailability{TEXT} %% use this section when having only software code available

\dataavailability{All data used in this study are available in the following online data bases: Polar Data Catalogue, Water Survey of Canada (Environment Canada), Canadian Ice Service (Environment Canada), National Centers for Environmental Prediction (NCEP), and the French national IMBER/SOLAS data base.} %% use this section when having only data sets available

% \codedataavailability{TEXT} %% use this section when having data sets and software code available

% \sampleavailability{TEXT} %% use this section when having geoscientific samples available

%\appendix
%\section{}     %% Appendix A
%
%\subsection{}     %% Appendix A1, A2, etc.
%
%
%\noappendix     %% use this to mark the end of the appendix section
%
%%% Regarding figures and tables in appendices, the following two options are possible depending on your general handling of figures and tables in the manuscript environment:
%
%%% Option 1: If you sorted all figures and tables into the sections of the text, please also sort the appendix figures and appendix tables into the respective appendix sections.
%%% They will be correctly named automatically.
%

```
%%% Option 2: If you put all figures after the reference list, please
insert appendix tables and figures after the normal tables and figures.
%%% To rename them correctly to A1, A2, etc., please add the following
commands in front of them:
%
%\appendixfigures  %% needs to be added in front of appendix figures
%
%\appendixtables   %% needs to be added in front of appendix tables
%
%%% Please add \clearpage between each table and/or figure. Further
guidelines on figures and tables can be found below.

\authorcontribution{JKE drafted the manuscript, analyzedanalysed the data
and prepared the figures. JKE and DD collected and analysed the SPM and
POC data, while RR conducted the particle size distribution sampling. BL
conducted all $\delta^{18}O$ sampling. All coauthors contributed to
writing the manuscript.} %% optional section

\competinginterests{The authors declare that they have no conflict of
interest.} %% this section is mandatory even if you declare that no
competing interests are present

% \disclaimer{TEXT} %% optional section

[revised manuscript text omitted]

2009.

```latex
\end{thebibliography}

%% Since the Copernicus LaTeX package includes the BibTeX style file
copernicus.bst,
%% authors experienced with BibTeX only have to include the following two
lines:
%%
%% \bibliographystyle{copernicus}
%% \bibliography{example.bib}
%%
%% URLs and DOIs can be entered in your BibTeX file as:
%%
%% URL = {http://www.xyz.org/~jones/idx_g.htm}
%% DOI = {10.5194/xyz}

%% LITERATURE CITATIONS
%%
%% command                          & example result
%% \citet{jones90}|                 & Jones et al. (1990)
%% \citep{jones90}|                 & (Jones et al., 1990)
%% \citep{jones90,jones93}|         & (Jones et al., 1990, 1993)
%% \citep[p.~32]{jones90}|          & (Jones et al., 1990, p.~32)
%% \citep[e.g.,][]{jones90}|        & (e.g., Jones et al., 1990)
%% \citep[e.g.,][p.~32]{jones90}|   & (e.g., Jones et al., 1990, p.~32)
%% \citeauthor{jones90}|            & Jones et al.
%% \citeyear{jones90}|              & 1990

%% FIGURES

%% When figures and tables are placed at the end of the MS (article in
one-column style), please add \clearpage
%% between bibliography and first table and/or figure as well as between
each table and/or figure.

\clearpage

\begin{figure*}[t]
\includegraphics[width=12cm]{Fig117.7cm]{Fig01_revision.pdf}
\caption{Map of study area with stations sampled along transect lines 100
to 700 during the MalinaMALINA expedition in 2009. CTD/Rosette water
sampling was conducted on the 28 stations marked by stars with black
borders. Black circles are the three locations selected for NCEP 10 m
winds. The green square near station 140 indicates the location of the
long-term mooring CA05 with a current meter at 178 m or 202 m depth.. The
cyan circles mark the locations for three of the profiles shown in Fig.
12.\ref{fig:deepSPM}. The fourth station, CB-21, was located 1$^{\circ}$
north of CB-27.}
\label{fig:MAP}
\end{figure*}
```

```
\begin{figure*}[t]
\includegraphics[width=17.7cm]{Fig02_revision.png}
\caption{Fields of water salinity (left panels) and
particulate beam attenuation coefficient at 660 nm, $c_\mathrm{p}(660)$,
(right panels) for (a--b)  surface, (c--d) 30 m depth, and (e--f) 80 m
during the MALINA 2009 expedition
. Dashed contour lines in (a) are the fraction of meteoric water (\%) of
the freshwater.}
\label{fig:SSURF}
\end{figure*}

\begin{figure*}[t]
\includegraphics[width=17.7cm]{Fig03_revision.pdf}
\caption{
Sections of geostrophic current velocity (colours and white contours)
perpendicular to transect lines 100 (a), 200 (b), 300 (c), and 600 (d).
Note the changes in scale. The grey contour lines are for potential
temperature. Geopotential heights were referenced to 500 m. Positive
current values are generally for the direction perpendicular to the
transect lines (see Figure 1) either towards northwest (a) or west (b--c)
or southwest (d). The location of the current meter on the CA05 mooring
is shown in (a).}
\label{fig:GEOSTROPHIC}
\end{figure*}

\begin{figure*}[t]
\includegraphics[width=17.7cm]{Fig04_revision.pdf}
\caption{
Voltage readings from the chlorophyll fluorometer (left panels) and CDOM
fluorometer (right panels) for transects (a--b) 100, (c--d) 300 and (e--
f) 600.}

\label{fig:FLUO}
\end{figure*}

\begin{figure}[t]
\includegraphics[width=
17.7cm]{Fig05_revision.png}
\caption{
(a) POC to SPM ratio for surface samples within the study area, (b)
relationship between POC to SPM ratio and meteoric water fraction of
freshwater in surface waters (see Fig. 2a), and relationship between
volume fraction of particles less than 1 $\mu$m in diameter
and (c) salinity

, and (d) meteoric water fraction of freshwater. Values in (d) are
limited to surface waters.}
\label{fig:POC2SPM}
\end{figure}

\begin{figure*}[t]
\includegraphics[width=17.7cm]{Fig06_revision.pdf}
```

\caption{
SPM and POC as a function of particulate beam attenuation
coefficient at 660 nm based on measurements from \textit{CCGS Amundsen}
during the MALINA expedition in 2009 (a--b), and as a function of
the particulate beam attenuation coefficient at 676 nm measured with the
AC-9 from the barge (c--d) (the latter data contain only surface
samples). The dotted squares in (c) and (d) indicate axes limits in (a)
and (b), respectively. The colours of the data points indicate
POC/SPM categories: mineral-dominated (red), mixed (blue), and organic-
dominated (green).}

\label{fig:FIT}
\end{figure*}

\begin{figure*}[t]
\includegraphics[width=17cm]{Fig07_revision}.pdf}

~~\caption{Sections of geostrophic current velocity (colors and white contours) perpendicular to transect lines 100 (a), 200 (b), 300 (c), and 600 (d). Note the changes in scale. The grey contour lines are for potential temperature. Geopotential heights were referenced to 500 m. Positive current values are generally for the direction perpendicular to the transect lines (see Figure 1) either towards northwest (a) or west (b, c) or southwest (d).}~~

\caption{

Concentration of suspended particulate matter, SPM, calculated from measurements of particulate beam attenuation coefficient at 660 nm, $c_\mathrm{p}(660)$,  Eq. 2 for lines 100, 300 and 600 during different field campaigns in 2004,  2008, and 2009, as indicated.}
\label{fig:SPM}
\end{figure*}

\begin{figure*}[t]

\includegraphics[width=17cm]{Fig08_revision}.pdf}
\caption{
As Fig. \ref{fig:SPM} but for measurements of water temperature (colours) and salinity (contour lines).}
\label{fig:TS}
\end{figure*}

\begin{figure*}[t]

\includegraphics[width=17.7cm]{Fig09_revision}.pdf}
\caption{
Vertical profiles of (a) suspended particulate matter, SPM, calculated from particulate beam attenuation coefficient at 660 nm, $c_\mathrm{p}(660)$, and (b) temperature, $T$, and salinity, $S$, at selected ‚Äúdeep‚Äù stations. Inserts (c) and (d) show transmissometer data (converted to SPM using Eqs. 1--2) that were collected in Canada Basin during 21--23 September 2009 and made available by the Beaufort Gyre Exploration Program based at the Woods Hole Oceanographic Institution (http://www.whoi.edu/beaufortgyre) in collaboration with researchers from Fisheries and Oceans Canada at the Institute of Ocean Sciences. Insert (e) shows a close up of the potential temperature, $\theta$, and $S$ for CB-23, CB-27 and CB-21 at the interface to the

Canada Basin Bottom Water layer. Grey horizontal lines indicate bottom
depths and are underlain by station numbers (see Fig. 1 for locations).}
\label{fig:deepSPM}
\end{figure*}

\begin{figure}[t]
\includegraphics[width=12cm]{Fig10_revision.pdf}
\caption{
(a) Daily discharge for the Mackenzie River at the Arctic Red River
location (10LC014). Data obtained from Environment Canada. (b) Weekly ice
coverage for the Mackenzie Shelf area calculated using IceGraph 2.0
provided online by the Canadian Ice Service. Time periods for the %four
three expeditions considered in this study are also indicated in colour
shades.}
\label{fig:RD-SIC}
\end{figure}

\begin{figure*}[t]
\includegraphics[width=7cm]{Fig11_revision.png}
\caption{
Ice coverage data from the Canadian Ice Service. The blue labels denote
areas of first-year ice (‚Äòf‚Äô) and multi-year ice (‚Äòm‚Äô), while
numbers that follow indicate ice concentration in tenths (9+ indicates >
90 \%). The areas of the two ice types are also associated with colours;
green for first-year ice, and red for multi-year ice. The colour shade
relates to concentration.}
\label{fig:CIS}
\end{figure*}

\begin{figure*}[t]
\includegraphics[width=17.7cm]{Fig12_revision.pdf}
\caption{
Progressive vector plots of (a) daily average wind (NCEP, 10 m) and (b)
currents from mooring CA05, with inset for 2003--2004 CASES data.
Colours in (a) indicate daily average wind speeds shown in colour bar.
The blue circles in (a) and (b) show the approximate times of the ship-
based transect sampling across the Mackenzie shelf break used in this
study. The black line shows the direction along the shelfbreak referenced
to True North. In (b), the same vector plot for currents is shown three
times, but the colours of each plot indicate either current speed,
salinity or temperature as denoted next to the lines and shown in more
detail in (c) and (d). The start of each month is indicated.
For the inset in (b) showing 2003--2004, the colours indicate salinity as
denoted.
Time series for 2008--2009 for (c) current speed (colours are current
directions), (d) salinity (red) and temperature (blue) at 178 m depth.}
\label{fig:VECTOR}
\end{figure*}

\begin{table}[t]
\caption{Salinity, saline end-member, meteoric and sea ice melt fractions
(\%), and meteoric water percentage of freshwater content (MW\% =
$f_{MW}/(f_{MW}+f_{SIM})\times 100$) for surface seawater samples
obtained at matching station locations during CASES 2004 and MALINA 2009.

```latex
A few matching station were also sampled during IPY-CFL 2008. The samples
were collected with Niskin bottles on a CTD-Rosette. The fraction of the
saline end-member ($f_{PSW}$) represents Pacific Summer Water with a
salinity of 31.5 PSU.}
\begin{tabular}{ c | c | c c c c c }
\tophline
Cruise  & Station & Salinity & $f_{PSW}$ & $f_{MW}$ & $f_{SIM}$ & MW\%
\\
\middlehline
CASES  & 110 & 25.4 & 80.1 & 6.6 & 13.3 & 34.8 \\
2004   & 140 & 27.8 & 84.6 & 5.1 & 10.4 & 33.2 \\
          & 150 & 29.3 & 89.1 & 7.5 &   3.4 & 68.6  \\
          & 170 & 29.8 & 91.0 & 7.0 &   2.1 & 77.5  \\
          & 320 & 29.4 & 89.2 & 1.3 &   9.4  & 12.2  \\
          & 340 & 27.1 & 81.4 & 3.4 & 15.2 & 18.3  \\
          & 360 & 25.1 & 79.9 & 19.6 & 0.5 & 97.5  \\
          & 380 & 24.8 & 76.2 & 19.5 & 4.3 & 82.0  \\
          & 390 & 25.4 & 78.8 & 20.4 & 0.8 & 96.3 \\
          & 660 & 15.9 & 48.5 & 40.9 & 7.7 & 79.3  \\
          & 670 & 16.9 & 52.6 & 40.2 &   7.2 & 84.7  \\
          & 690 &    8.8 & 26.0 & 60.5 & 13.5 & 81.8  \\
\middlehline
IPY-CFL & 110 & 28.2 & 85.1 & 4.9 & 10.1 &   32.5 \\
2008    & 140 & 28.2 & 85.1 & 4.3 & 10.6 &   28.8  \\
          & 160 & 30.3 & 91.6 & 3.7 &   4.7 &   44.4  \\
    & 320$^*$ & 26.3 & 79.5 & 12.4 & 8.1  & 60.5  \\
    & 340$^*$ & 25.0 & 75.5 & 17.5 & 7.1  & 71.1  \\
          & 390 & 29.4 & 88.9 & 11.0 & 0.2 & 98.6  \\
\middlehline
MALINA & 110 & 28.9 & 86.6 & 4.8 & 8.6 & 35.6  \\
 2009   & 150 & 29.4 & 89.4 & 4.9 & 5.7 & 46.1  \\
          & 170 & 29.3 & 89.9 & 6.3 & 3.9 & 62.0  \\
          & 320 & 26.5  & 79.2 & 6.3 & 14.5 & 30.2  \\
          & 340 & 26.9 & 79.7 & 4.5 & 15.6 & 22.2  \\
          & 360 & 26.5 & 78.4 & 4.6 & 17.0 & 21.2 \\
          & 380 & 27.7 & 83.1 & 6.1 & 10.8 & 36.0 \\
          & 390 & 27.2 & 83.5 & 7.8 & 8.7 & 47.2  \\
          & 660 & 21.9 & 63.9 & 17.0 & 19.1 & 47.1  \\
          & 670 & 23.4 & 68.0 & 15.3 & 16.7 & 47.8  \\
          & 690 & 27.2 & 67.2 & 18.6 & 14.2 & 56.7  \\
\bottomhline
\end{tabular}
\belowtable{$^*$Approximately 5 nm south of station, which is half way to
next station} % Table Footnotes
\end{table}

\end{document}
```

---

## Referee Report (RR1)

Second review of '*Patterns of suspended particulate matter across the continental margin in the Canadian Beaufort Sea during summer*' by Ehn, Reynolds, Stramski, Doxaran, Lansard and Babin

Many improvements have been made to this manuscript. It is now much easier to read, has a more logical structure, and the important results are clearer. There were still a few places where I found the manuscript confusing. In addition, I questioned why 2007 were data removed from the revised manuscript? I also have some minor suggestions, listed below:

- Page 1, Line 18 – Put (Fig. 1) after Mackenzie Shelf
- Figure 1:
  - Mackenzie Trough and Kugmallit Valley should be labeled
  - I still can't tell the different between the purple and red stars
  - Please make it clear in the caption that the small black dots are stations sampled by the barge
- Page 2, line 23 – Where is the proof that the material reaching the Canada Basin deep ocean is thousands of years old?
- Page 3, lines 3 to 11 – I find these sentences difficult to read and confusing. I suggest that the authors tighten up the writing so that the main points are clearer.
- Page 6, lines 23 to 24 – Why are the end members used here for Pacific Summer Water different than those used by Yamamoto-Kawai et al. (2008)?
- Section 2.6 – Please state in this section the depth of the various instruments on the moorings.
- Figure 2 – I think that this figure would be much easier to read if the stations were added to it. I find the varying bathymetry disorienting and I think it would be easier to read if the stations were on the figures as stable objects.
- Figure 3:
  - I can't read the colorbars in a) and d)
  - I suggest that the range for all colorbars is the same (-9 to 9 cm/s)
- Page 8, lines 11 to 15 – Please rewrite to make the key points clearer. Has this eastward current been observed before in this area?
- Page 8, lines 25 to 26 – This sentence doesn't make sense to me. Please clarify.
- Page 9, lines 2 to 3 – Water at 13 m is not surface water. Please clarify.
- Page 9, lines 15 to 16 – Here POC/SPM ratios are reported in fractions but they are reported in percentage in Figure 5. I suggest that these units are consistent.
- Page 9, lines 16 to 18 – Please add (Fig. 5a) at the end of this sentence.
- Page 9, line 34 – Based on the caption, I don't think that Fig. 5c shows subsurface water, just surface samples
- Section 3.4:
  - Why weren't sections of POC shown?
  - Why was 2007 data excluded from the revised manuscript?
- Page 12, line17 – I suggest adding 'At the surface" before 'SPM decreased..."

- Page 13, lines 10 to 13 – I find this sentence confusing. I don't see any data describing shelf circulation in this manuscript.
- Section 3.4.2
    - I suggest that this section is rewritten so that it has more clarity. I found it difficult to follow and to understand the key points of this section
- Page 13, line 19 – I don't think 'inversions' is the right word here. Perhaps 'features' is a better word?
- Page 13, line 23 – Which station has a 1000 m thick BNL?
- Page 13, line 26 – At what depth was the BNL at station CB-27?
- Page 13, line 31 – At what depth was the INL at stations CB23, CB27, and CB21?
- Page 15, lines 21 to 22 – Please add a reference to this sentence.
- Figure 12
    - I find figures 12a and 12b difficult to read
        - The labels are challenging (e.g. does D08 stand for December 08?)
        - I can't see the inset line of axes values on Figure 12b
        - I suggest that for CASES, MALINA, and CFL that data from 1-2 months before the cruises are shown. I don't think that a full year of data are needed here
    - Figure 12 c – there is no colorbar to indicate current direction – please add
- Page 15, line 31 – I think that there are westerly winds from October and December through March?
- Pages 15, lines 30 to 34 – I find these sentences confusing and contradictory. Please clarify.
- Page 16, lines 16 to 17 – I can't see cross-shelf currents in figure 12a. They're undoubtedly there but there is not enough information in the caption or figure for me to tell when currents are cross-shelf

---

## Referee Report (RR2)

Second REVIEW of the manuscript:

**"Patterns of suspended particulate matter across the continental margin in the Canadian Beaufort Sea", Jens K. Ehn, Rick A. Reynolds, Dariusz Stramski, David Doxaran, and Marcel Babin**

General comments

The reviewed manuscript presents major revisions to the original version and has taken into account many comments proposed by the reviewers. In particular, the authors focused on the main findings and eliminated as they state, "unnecessary descriptions".
The introduction of new data on the oxygen isotopic composition is a valuable addition to their dataset. It clarifies some flaws from the original manuscript, especially on the origin and composition of the particulate matter by distinguishing between characteristics of particles from riverine input and open water particles, especially those contained in sea ice meltwater.
I still regret that the authors did not attempt a multivariate analysis of their data (e.g. PCA). However, the reviewed manuscript focuses now clearly on some major findings, which does not necessarily call for a more extensive data treatment. But in this case, the authors should limit their discussion to these major findings and eliminate sections that would only contribute to the scientific content if the data were treated as I had proposed. Otherwise, these data and discussions are too flaw and superfluous, since the core data do illustrate the main findings of this manuscript.
I will explain what I would consider to be these main findings in the following specific comments.

Specific comments

Introduction: The revised introduction gives a clear overview of the different aspects of particle dynamics in this very complex environment with a major riverine input and a continental shelf exposed to a dynamic pattern of currents, which strongly depend on wind forcing but also on the variable annual cycle of ice coverage. The revised text presents the scientific context in a concise and well-focused way, which facilitates the lecture and understanding of the rest of the manuscript.

Paragraph 3: This paragraph is clearly structured. It starts with the MALINA data on the hydrology (3.1.) and then on the particle distribution (3.2.), and it presents first evidence on some main findings, i.e., the driving forces (wind, ice coverage, meltwater) of a dynamic environment and the broad range of particle size, composition and concentration. This logically leads to the relationship between particles and beam attenuation (3.3.) and finally to a comprehensive description of suspended particulate matter (SPM) distribution in space and time (3.4.). The distinction of particle characteristics between river water and ice melt water, although not much surprising, is one of the main findings of this paper.

Paragraph 3.3.: The new text focuses much better on the relationship between SPM, POC and beam attenuation, by avoiding confuse and detailed descriptions of the

regression analysis, which is now in the supplementary material, where it appears in a clear form.

I have one major concern about the mathematics of the relationship. On page 10, line 28, the authors present the "counterparts" of equations 2) and 3). Unless I misinterpreted the expression "counterpart", I did not come up with the same equations. I had not seen this in my first review, but since that relationship is another major outcome of this paper, the "counterparts" need to be clarified, and I hope it does not concern equations 2) and 3), on which all subsequent calculations are based.

Paragraph 3.4.: This paragraph together with 3.6. contains the main findings of this article, the temporal and spatial variation of the particle distribution and dynamics, which depend on 1) river discharge, 2) ice coverage and meltwater and 3) wind forcing.

Page 12, lines 22-25: The argument that resuspension was insufficient to increase clear water beam attenuation values of wedges that reach far onto the shelf (lines 100 and 600 in 2009) contradicts their argument in paragraph 3.6.2., line 34, where resuspension of shelf sediments could explain the turbid surface waters. Errors of this type could be avoided in a multivariate analysis of the data.

I suggest integrating paragraph 3.4.1. into 3.4. and to remove paragraph 3.4.2. As I had already mentioned in my first review, this section was the least convincing. Although the authors removed the most critical part, the paragraph as a whole does not really contribute to the main findings, let alone the title of the manuscript, which focuses on the continental margin. I understand well their argument to keep the figure (former fig. 11, now fig. 9), but suggest that they only present a couple of contrasting SPM profiles to illustrate the shelf to basin differences and discuss this within the general context of nepheloid layers at the end of section 3.4. No need to go into details about thickness of these layers and particle concentrations and transport.

Paragraphs 3.5. and further: I don't think that it is necessary to separately discuss the data on environmental forcing and oceanographic conditions. The data presented in these sections do not illustrate a specific finding *per se*, but help to interpret and explain the preceding data, which is done in section 3.6. That said, Fig. 10 can still be maintained and used in section 3.6., Fig. 11 at best be presented as supplementary material, and Fig. 12 remains a very complicated one despite the simplifications done by the authors. Not surprising that there were confusions in interpreting the data. On page 16, line 6, they talk about southwesterly winds at the end of July, which become southeasterly ones on the same page, line 18.The importance of this figure is to show the periods of upwelling and downwelling favourable wind conditions. Why not put the Figs. 12a and b to the supplementary material and make a graph (histogram type), which shows on two time axes the periods of easterly and the periods of westerly winds and on the y-axis the average wind speed? This would be sufficient to illustrate the discussion in section 3.6. together with references from the literature (Carmack, Dmitrenko, Macdonald, Forest, Mol).

Paragraphs 3.6.: As I said for 3.4., the main findings are presented in 3.6. and 3.4.

Page 18, lines 28 and further: I do not completely agree with the

interpretations in this paragraph. As I said before, the resuspension hypothesis contradicts the text on page 12 and the temperature and salinity fields (page 18, lines 32-33) are modestly different in 2008 and 2009. 2004 is quite different with salinity values >30 and temperatures not exceeding 2 degrees Celcius, while they were >5 degrees and salinity <28.5 in 2008 and 2009. Again, a multivariate analysis may have shed a clear light on these interpretations. I would therefore suggest to only discuss the influence of light and SPM on primary productivity for the Amundsen Gulf and line 100.

Conclusion: By removing/modifying Fig. 12, the paragraph about the mooring data (lines 28 and further) could be more general and highlight the second part related to Fig. 8 (page 20, line 1 and further) and including Fig. 2 where the upwelling onto the shelf is also illustrated by the east-west salinity gradients related to easterly wind conditions.

Technical corrections

- Page 8, line 3: The percentage here is rather confusing since meteoric water fraction given in percent is discussed. I would give a salinity value (e.g.: >29 PSU).
- Page 9, lines 15,16: In Fig. 5a values are given as percentage. It is better to match the text with the figure.
- Page 18, line 16: Add "Fig." to "8e".
- Page 18, line 25: Fig. 8f not 7f.
- Page 19, line 31: cross-shelf (see also page 12, 14, 17 line 1, 30, 25: cross-section)

References

References Forest et al. 2010 and Spall et al. 2014 are not cited in the text.

Figures

The dates in Fig. 10 are rather confusing.

---

## Author Response (AR2)

**Associate Editor Decision: Publish subject to minor revisions (review by editor)** (17 Jan 2019) by Tina Treude
Comments to the Author:
Dear Dr. Ehn and Co-workers,

The referees had a close look at your revision and both agree that the manuscript has significantly improved. However, both still suggest some minor adaptations, which I believe you should be able to implement. Referee #1 advises to remove any interpretations that should only be made after applying a multivariance analysis. Referee #2 has mainly technical comments.

Please provide a detailed point-by-point response to the reviewers and how you implemented their comments. Please submit your revised manuscript with all new edits highlighted.

Let me know if you have any questions
With kind regards
Tina Treude

Responses to the second review of the manuscript:

"Patterns of suspended particulate matter across the continental margin in the Canadian Beaufort Sea during summer", Jens K. Ehn, Rick A. Reynolds, Dariusz Stramski, David Doxaran, Bruno Lansard, and Marcel Babin

We greatly appreciate the comments from the reviewers. Here, we provide our detailed point-by-point responses and any description of action taken in regards to the comments by Referee #1 and Referee #2. The Referees' comments are shown in regular font; our responses follow each comment in blue font.

**Response to Referee #1**

**"Patterns of suspended particulate matter across the continental margin in the Canadian Beaufort Sea", Jens K. Ehn, Rick A. Reynolds, Dariusz Stramski, David Doxaran, and Marcel Babin**

General comments

The reviewed manuscript presents major revisions to the original version and has taken into account many comments proposed by the reviewers. In particular, the authors focused on the main findings and eliminated as they state, "unnecessary descriptions".

The introduction of new data on the oxygen isotopic composition is a valuable addition to their dataset. It clarifies some flaws from the original manuscript, especially on the origin and composition of the particulate matter by distinguishing between characteristics of particles from riverine input and open water particles, especially those contained in sea ice meltwater. I still regret that the authors did not attempt a multivariate analysis of their data (e.g. PCA). However, the reviewed manuscript focuses now clearly on some major findings, which does not necessarily call for a more extensive data treatment. But in this case, the authors should limit their discussion to these major findings and eliminate sections that would only contribute to the scientific content if the data were treated as I had proposed. Otherwise, these data and discussions are too flaw and superfluous, since the core data do illustrate the main findings of this manuscript.

REPLY: We really appreciate the time taken by the Reviewer to help with the structure of the manuscript. We believe that the structure and associated content of the manuscript improved significantly as a result of our revisions in response to Reviewer's comments. However, as explained in greater detail in our previous responses to Reviewer's comments, we have not been able to completely comply with all suggestions. In particular, we did not conduct the multivariate analysis because the potential additional meaningful insights from such analysis, as applied to the available datasets which are unavoidably limited in size and scope, are unclear to us.

I will explain what I would consider to be these main findings in the following specific comments.

Specific comments

Introduction: The revised introduction gives a clear overview of the different aspects of particle dynamics in this very complex environment with a major riverine input and a continental shelf

exposed to a dynamic pattern of currents, which strongly depend on wind forcing but also on the variable annual cycle of ice coverage. The revised text presents the scientific context in a concise and well-focused way, which facilitates the lecture and understanding of the rest of the manuscript.

REPLY: Thank you.

Paragraph 3: This paragraph is clearly structured. It starts with the MALINA data on the hydrology (3.1.) and then on the particle distribution (3.2.), and it presents first evidence on some main findings, i.e., the driving forces (wind, ice coverage, meltwater) of a dynamic environment and the broad range of particle size, composition and concentration. This logically leads to the relationship between particles and beam attenuation (3.3.) and finally to a comprehensive description of suspended particulate matter (SPM) distribution in space and time (3.4.). The distinction of particle characteristics between river water and ice melt water, although not much surprising, is one of the main findings of this paper.

REPLY: Thank you.

Paragraph 3.3.: The new text focuses much better on the relationship between SPM, POC and beam attenuation, by avoiding confuse and detailed descriptions of the regression analysis, which is now in the supplementary material, where it appears in a clear form.
I have one major concern about the mathematics of the relationship. On
page 10, line 28, the authors present the "counterparts" of equations 2) and 3). Unless I misinterpreted the expression "counterpart", I did not come up with the same equations. I had not seen this in my first review, but since that relationship is another major outcome of this paper, the "counterparts" need to be clarified, and I hope it does not concern equations 2) and 3), on which all subsequent calculations are based.

REPLY: We did not simply invert Eqs. 2 and 3 algebraically, but the regression functions for the "counterparts" (i.e. cp(660) vs. SPM and cp(660) vs POC) were recalculated with cp(660) as the outcome ("independent") variable. We corrected the relevant sentence to make it clearer, by replacing the original "The counterparts of Eqs. 2 and 3 are…" with "For this case the best-fit regression functions are…".

Paragraph 3.4.: This paragraph together with 3.6. contains the main findings of this article, the temporal and spatial variation of the particle distribution and dynamics, which depend on 1) river discharge, 2) ice coverage and meltwater and 3) wind forcing.

REPLY: No reply is necessary here.

Page 12, lines 22-25: The argument that resuspension was insufficient to increase clear water beam attenuation values of wedges that reach far onto the shelf (lines 100 and 600 in 2009) contradicts their argument in paragraph 3.6.2., line 34, where resuspension of shelf sediments could explain the turbid surface waters. Errors of this type could be avoided in a multivariate analysis of the data.

REPLY: Thank you for your comment. We do not see a significant contradiction between the two arguments. We feel that the explanation of the clear water wedge formation is accurate. However, we did not perhaps express sufficiently clearly that we did not mean that this sediment

in the surface layer (plume) was vertically mixed from the bottom at the station, but rather advected from shallow waters closer to the shore. We cannot say with full certainty that all particles came from the Mackenzie River as some particles were likely added to the surface plume in shallow shelf waters where the plume still touched the bottom. Hence, we have added to section 3.6.2 line 34 the following:
"… and/or from bottom resuspension in shallow shelf waters closer to the shore."

I suggest integrating paragraph 3.4.1. into 3.4. and to remove paragraph 3.4.2. As I had already mentioned in my first review, this section was the least convincing. Although the authors removed the most critical part, the paragraph as a whole does not really contribute to the main findings, let alone the title of the manuscript, which focuses on the continental margin. I understand well their argument to keep the figure (former fig. 11, now fig. 9), but suggest that they only present a couple of contrasting SPM profiles to illustrate the shelf to basin differences and discuss this within the general context of nepheloid layers at the end of section 3.4. No need to go into details about thickness of these layers and particle concentrations and transport.

REPLY: Thank you for your suggestion. We have largely addressed your suggestion, but not fully. We did not remove paragraph 3.4.2 (Deep waters), but combined the discussion on subsurface nepheloid layers (including the subsurface chlorophyll maximum) into one section titled "subsurface nepheloid layers". This section includes the last paragraph on page 12 at the end of the section. Regarding section 3.4.2, which has not been removed, we feel that it is of relevance to show that what occurs below 500m is still linked to the shelf. Furthermore, the continental rise is technically still considered part of the continental margin, hence we think including it fits within what the title suggests.

Paragraphs 3.5. and further: I don't think that it is necessary to separately discuss the data on environmental forcing and oceanographic conditions. The data presented in these sections do not illustrate a specific finding *per se*, but help to interpret and explain the preceding data, which is done in section 3.6. That said, Fig. 10 can still be maintained and used in section 3.6., Fig. 11 at best be presented as supplementary material, and Fig. 12 remains a very complicated one despite the simplifications done by the authors. Not surprising that there were confusions in interpreting the data. On page 16, line 6, they talk about southwesterly winds at the end of July, which become southeasterly ones on the same page, line 18. The importance of this figure is to show the periods of upwelling and downwelling favourable wind conditions. Why not put the Figs. 12a and b to the supplementary material and make a graph (histogram type), which shows on two time axes the periods of easterly and the periods of westerly winds and on the y-axis the average wind speed? This would be sufficient to illustrate the discussion in section 3.6. together with references from the literature (Carmack, Dmitrenko, Macdonald, Forest, Mol).

REPLY: Earlier we did try to structure the manuscript as suggested. Without the support of section 3.5, however, we found that section 3.6 became very unclear. We decided that it was better to first present the physical forcing as its own section (3.5) and then refer to this background information in section 3.6.

We agree that with the Reviewer's suggestion that Fig. 11 could be moved to the supplementary materials, and have done so. The text has been changed throughout the manuscript to accommodate this. Thank you for this suggestion.

Regarding page 16 line 11 vs. line 18: We had a mistake on line 16. However, now we have changed the text at both locations to read "easterly" rather than "southeasterly" as there was not much of a southerly component.

With regard to modification of Fig. 12a (now Fig. 11a) the graphs below show the cross-shelf (roughly N-S) and along-shelf (roughly E-W) components of the wind vectors plotted in Fig. 12a where they are shown as progressive vector plots (cumulative sum). The U and V components of the wind have been rotated by 52 degrees to calculate the cross and along shelf components. Three-day averages of the data are shown to reduce variability. The negative values in the along-shore plot (lower subplot) are linked to when we see the cross-shelf currents in Fig. 12b. These graphs are perhaps simpler to understand than the progressive vector plot. However, in our opinion, the progressive vector plot in Fig. 12a shows the timing of these upwelling inducing wind events just as well as the graphs below. Secondly, Fig. 12a (and Fig. 12b) also include CASES 2004 data, which is not easily shown in the below graphs and would likely necessitate additional graphs. Therefore, after considering the suggestion by the reviewer, we prefer to keep the original graph.

[Figure]

Paragraphs 3.6.: As I said for 3.4., the main findings are presented in 3.6. and 3.4. Page 18, lines 28 and further: I do not completely agree with the interpretations in this paragraph. As I said before, the resuspension hypothesis contradicts the text on page 12 and the temperature and salinity fields (page 18, lines 32-33) are modestly different in 2008 and 2009. 2004 is quite different with salinity values >30 and temperatures not exceeding 2 degrees Celcius, while they were >5 degrees and salinity <28.5 in 2008 and 2009. Again, a multivariate analysis may have shed a clear light on these interpretations. I would therefore suggest to only discuss the influence of light and SPM on primary productivity for the Amundsen Gulf and line 100.

REPLY: In each case the water column remains stratified eliminating the possibility of clear waters being caused by deeper waters coming to the surface. Regarding the resuspension we

have further clarified the text  as the SPM in the surface layer was most likely advected to the location from the inner shelf and river. The relevant sentence now reads as follows:

"This suggests that the turbid surface waters in 2004 and 2009 were caused by the advection of shelf waters containing particles that originated from the Mackenzie River  **and/or from bottom resuspension in shallow shelf waters closer to the shore.**"

Conclusion: By removing/modifying Fig. 12, the paragraph about the mooring data (lines 28 and further) could be more general and highlight the second part related to Fig. 8 (page 20, line 1 and further) and including Fig. 2 where the upwelling onto the shelf is also illustrated by the east-west salinity gradients related to easterly wind conditions.

REPLY: Because we chose not to remove or modify Fig. 12 (see our earlier reply), we have not changed this paragraph.

Technical corrections

Page 8, line 3: The percentage here is rather confusing since meteoric water fraction given in percent is discussed. I would give a salinity value (e.g. >29 PSU).

REPLY: In fact, we report the relative contribution of meteoric water and sea ice meltwater. Thus, we wanted to point out that equal contribution between the two was not really important as there was relatively little freshwater overall. The use of the % value, instead of salinity, make this more clear, in our opinion.

Page 9, lines 15,16: In Fig. 5a values are given as percentage. It is better to match the text with the figure.

REPLY: We have changed the POC/SPM ratio reported within the text and figures to be expressed as % in order to be consistent with the meteoric water fraction (also expressed in %). The regression equation in Fig. 5b has also be adjusted apply for percentage values.

Page 18, line 16: Add "Fig." to "8e".

REPLY: Thank you. Done.

Page 18, line 25: Fig. 8f not 7f.

REPLY: Thank you. Done.

Page 19, line 31: cross-shelf (see also page 12, 14, 17 line 1, 30, 25: cross-section)

REPLY: Changed at all locations as suggested.

Figures

The dates in Fig. 10 are rather confusing.

REPLY: We have specified in the figure caption that the date labels on the x-axes are expressed as month/day: "The x-axis labels are dates expressed in the format of month/day, and are spaced 4 weeks apart." We hope this makes it clearer.

Responses to the second review of the manuscript:

"Patterns of suspended particulate matter across the continental margin in the Canadian Beaufort Sea during summer", Jens K. Ehn, Rick A. Reynolds, Dariusz Stramski, David Doxaran, Bruno Lansard, and Marcel Babin

We greatly appreciate the comments from the reviewers. Here, we provide our detailed point-by-point responses and any description of action taken in regards to the comments by Referee #1 and Referee #2. The Referees' comments are shown in regular font; our responses follow each comment in blue font.

**Response to Referee #2**

Second review of '*Patterns of suspended particulate matter across the continental margin in the Canadian Beaufort Sea during summer*' by Ehn, Reynolds, Stramski, Doxaran, Lansard and Babin

Many improvements have been made to this manuscript. It is now much easier to read, has a more logical structure, and the important results are clearer. There were still a few places where I found the manuscript confusing. In addition, I questioned why 2007 were data removed from the revised manuscript? I also have some minor suggestions, listed below:

REPLY: We removed the 2007 ArcticNet data to better focus the manuscript on river and meltwater during the summer season. Ice formation is an additional important process during the late fall. For instance, frazil ice formation may have been an important factor in determining the beam attenuation, which would deserve a separate analysis and discussion beyond the use of the main relationship between the particles and optical beam attenuation, which underlies our study.

Page 1, Line 18 – Put (Fig. 1)
after Mackenzie Shelf

REPLY: We do not think it is needed to refer to Fig. 1 in the first line of the introduction.

Figure 1: Mackenzie Trough and Kugmallit Valley should be labeled. I still can't tell the different between the purple and red stars

REPLY: Just as we do not refer to Fig. 1 at the beginning of the introduction, we do not think it is necessary to refer to the figure when we mention Mackenzie Trough and Kugmallit Valley in the 1st paragraph of the introduction. The focus of the figure is to show the sampling stations and we feel that adding the labels would add unnecessary detail.
The blue stars represent 2-day stations; however, there is actually no need to mention them because, for the purpose of this manuscript, they are no different from regular CTD/Rosette stations with water sampling. Hence, we have changed their color to red. Thank you for pointing this out.

Figure 1: Please make it clear in the caption that the small black dots are stations sampled by the barge.

REPLY: We added to the figure caption: "Stars indicate stations visited by *CCGS Amundsen* and small circles indicate the estuarine stations sampled by the barge."

Page 2, line 23 – Where is the proof that the material reaching the Canada Basin deep ocean is thousands of years old?

REPLY: The paper by Honjo et al. (2010) we cite states "In contrast, the POC exported to 3067 m had an apparent $^{14}$C age of 1900 years, indicating it was predominantly derived from aged, allochthonous carbon."

Page 3, lines 3 to 11 – I find these sentences difficult to read and confusing. I suggest that the authors tighten up the writing so that the main points are clearer.

REPLY: We changed the first sentence on line 3 from "The proportion of organic to inorganic material is important because mineral particles typically have higher refractive index compared to organic particles, and thus generally produce higher scattering per unit mass concentration" to "The relationships are affected by the proportions of organic and inorganic material, because mineral…". We think this wording better fits in the paragraph. In our opinion, additional changes are not needed.

Page 6, lines 23 to 24 – Why are the end members used here for Pacific Summer Water different than those used by Yamamoto-Kawai et al. (2008)?

REPLY: In our study, the water mass analysis is based on temperature, salinity, $\delta^{18}$O, total alkalinity and dissolved $O_2$ concentration data (see Lansard, B., Mucci, A., Miller, L. A., Macdonald, R. W., and Gratton, Y.: Seasonal variability of water mass distribution in the southeastern Beaufort Sea determined by total alkalinity and $\delta^{18}$O, Journal of Geophysical Research, 117, C03003, C3, C03003, doi: 10.1029/2011jc007299, 2012). In contrast, Yamamoto-Kawai et al. (2008) used nitrate, nitrite, ammonium, and phosphate data, together with $\delta^{18}$O and S data, to quantitatively estimate the distributions of the four main sources of waters to the Canada Basin: Atlantic water, Pacific water, sea ice meltwater, and meteoric water.

The definition of the seawater end-member in the Arctic Ocean differs between the studies for a practical reason that a given number of tracers, n, permits solutions for only n+1 water masses. This implies that with two tracers only one saline end-member may be used. Depending on the study objectives and location, the saline end-member has been assigned to the UHW or Pacific Water [Macdonald et al., 1989, 1995, 2002; Yamamoto-Kawai et al., 2008, 2009], Atlantic Water [Östlund and Hut, 1984; Schlosser et al., 2002; Anderson et al., 2004b; Yamamoto-Kawai et al., 2005; Newton et al., 2008], and the Polar Mixed Layer [Alkire and Trefry, 2006]. Because the computation of freshwater and sea ice melt fractions (and inventory) depends on the seawater end-member definition, absolute results cannot be compared directly among studies using different saline end-member baselines, although patterns in distributions may be compared.

Section 2.6 – Please state in this section the depth of the various instruments on the moorings.

REPLY: We state the depth of the RCM11 current meter for the years used in this study, and it is the only instrument on the mooring from which we utilize data. We do not think it necessary to include here the depths of the other instruments on the mooring. Each year the mooring contained various instruments at varying depths, and sometimes instruments malfunctioned.

Therefore, the description of these sensor depths would be long and largely redundant, especially as we only use the RCM11.

Figure 2 – I think that this figure would be much easier to read if the stations were added to it. I find the varying bathymetry disorienting and I think it would be easier to read if the stations were on the figures as stable objects.

REPLY: We have added dots to mark locations for the CTD/Rosette stations sampled onboard the Amundsen. Note that the salinity measurements taken onboard the barge are not included (the barge CTD did not have a transmissometer).

Figure 3:
I can't read the colorbars in a) and d)
I suggest that the range for all colorbars is the same (-9 to 9 cm/s)

REPLY: We think that the colorbars are clearly displayed in a) and d). Perhaps the Reviewer received a different version of the figure. We agree with the second point and have made a new figure with each subplot having a consistent scale between -6 and 9 cm/s. Higher current speeds are shown by contour lines.

Page 8, lines 11 to 15 – Please rewrite to make the key points clearer. Has this eastward current been observed before in this area?

REPLY: We cite the studies that have shown the eastward shelfbreak current which, in our opinion, is clear. To our knowledge, there have been no other publications showing such result.

Page 8, lines 25 to 26 – This sentence doesn't make sense to me. Please clarify.

REPLY: To clarify we added the underlined text to this sentence: "However, the measured particulate absorption at 660 nm was found to be smaller by 1–4 orders of magnitude than cp(660) and can thus be ignored (data not shown)."

Page 9, lines 2 to 3 – Water at 13 m is not surface water. Please clarify.

REPLY: The sentence clearly states that we sampled surface water in waters that were 13 m deep.

Page 9, lines 15 to 16 – Here POC/SPM ratios are reported in fractions but they are reported in percentage in Figure 5. I suggest that these units are consistent.

REPLY: We have changed the POC/SPM ratio reported within the text and figures to be expressed as % in order to be consistent with the meteoric water fraction (also expressed in %). Consequently we have also adjusted the coefficients in the regression in Fig. 5b.

Page 9, lines 16 to 18 – Please add (Fig. 5a) at the end of this sentence.

REPLY: Done.

Page 9, line 34 – Based on the caption, I don't think that Fig. 5c shows subsurface water, just surface samples

REPLY: Thank you for pointing this out. Data points in (c) with S > 30 are for subsurface samples. This was not correctly described in the caption. We have added the following sentence to the caption to rectify this: "Values in (a), (b) and (d) are limited to surface water samples, while data points in (c) with salinities over 30 PSU represent subsurface samples."

Section 3.4:

- Why weren't sections of POC shown?

REPLY: Sections for POC look very similar to SPM (but have different range in colourbars) because they are both derived from cp(660). Therefore, we have chosen to just show SPM as there is no added value in duplicating the contour plots. However, we have provided the equation POC = 0.1279*SPM^0.7307 on page 12, line 14, which provides the relationship between POC and SPM.

Why was 2007 data excluded from the revised manuscript?

REPLY: We removed the 2007 ArcticNet data to better focus the manuscript on river and meltwater during the summer season. Ice formation is an additional important process during the late fall. For instance, frazil ice formation may have been an important factor in determining the beam attenuation, which would deserve a separate analysis and discussion beyond the use of the main relationship between the particles and optical beam attenuation, which underlies our study.

Page 12, line17 – I suggest adding 'At the surface" before 'SPM decreased..."

REPLY: Following this suggestion we changed to "SPM in the surface layer decreased…".

Page 13, lines 10 to 13 – I find this sentence confusing. I don't see any data describing shelf circulation in this manuscript.

REPLY: We show geostrophic currents in Fig. 3, we discuss and infer upwelling and downwelling circulation on the shelf from cp, S and CDOM data, and we cite papers describing circulation (e.g. Mol et al, 2018).

Section 3.4.2
I suggest that this section is rewritten so that it has more clarity. I found it difficult to follow and to understand the key points of this section

REPLY: As the "deep water" section is essentially about nepheloid layers, we have attempted to improve this section by creating a subsection entitled "Subsurface nepheloid layers" that combines all discussion on subsurface nepheloid layers (from original page 12 line 28, to page 14 line 12).

Page 13, line 19 – I don't think 'inversions' is the right word here. Perhaps 'features' is a better word?

REPLY: We have changed to "…numerous INLs and the decrease in SPM beneath or above INLs, which rules out turbulent mixing and suggests lateral advection in the formation of this SPM structure.".

Page 13, line 23 – Which station has a 1000 m thick BNL?

REPLY: Stations 620 and CB-23. See Fig. 9.

Page 13, line 26 – At what depth was the BNL at station CB-27?

REPLY: We argue that there is no BNL at CB-27 as there is no increase in SPM near the bottom.

Page 13, line 31 – At what depth was the INL at stations CB23, CB27, and CB21?

REPLY: This was described in the next paragraph (page 14 lines 3-12). However, we have now combined these two paragraphs.

Page 15, lines 21 to 22 – Please add a reference to this sentence.

REPLY: We added a reference to Carmack and Macdonald (2002), who describe this flow along the coast in the absence of wind forcing.

Figure 12
I find figures 12a and 12b difficult to read
- The labels are challenging (e.g. does D08 stand for December 08?)
- I can't see the inset line of axes values on Figure 12b
- I suggest that for CASES, MALINA, and CFL that data from 1-2 months before the cruises are shown. I don't think that a full year of data are needed here

REPLY: To address your first point, we have added the following sentence to the caption (now Fig. 11): "The black squares in (a) and (b) indicate the start of each month (the first letter of month followed by year, e.g., D03 stands for December 2003), while the blue circles show the approximate times of the ship-based transect sampling across the Mackenzie shelf break used in this study." The scale for the inset is the same as for the main figure. With regard to the third point this is probably true but we feel that including the longer time series better shows the overall wind and circulation in the study area, and allow us to compare the situation during other expeditions that did not occur during the same time of the year (just as we also compare the discharge in Fig 10).

Figure 12 c – there is no colorbar to indicate current direction – please add

REPLY: The colors in 12c were simply to highlight the direction of the current, which is seen in 12b. Figures 12b and 12c are complementary (Note that Fig. 12 is now Fig. 11). We have also wrote in the text what the main current directions are.

Page 15, line 31 – I think that there are westerly winds from October and December through March?

REPLY: Westerly winds continue until end of January 2008, then become weak and switch to easterly. We agree with October 2008. Thus, we have kept our text as it was.

Pages 15, lines 30 to 34 – I find these sentences confusing and contradictory. Please clarify.

REPLY: To clarify this paragraph, we rewrote the sentence in line 30 and removed the last sentence in lines 33-34. The text now reads as follows:

"Two notable periods dominated by westerly winds occurred during the month of October 2008, and during December 2008 to the end of January 2009. Typically, the westerly wind episodes were characterized by relatively low wind speeds."

Then, we added to section 3.5.3:
"The long periods (i.e., from October to November 2008, from December 2008 to April 2009, and from May to June 2009) of along-shelf southwestward currents at the 178 m depth at mooring CA05 (Fig. 11b) were related to periods with either weak winds or westerly winds Fig. 11a)." [Note Fig. 12 is now 11 after removal of the ice charts in Fig. 11].

Page 16, lines 16 to 17 – I can't see cross-shelf currents in figure 12a. They're undoubtedly there but there is not enough information in the caption or figure for me to tell when currents are cross-shelf

REPLY: The cross-shelf currents would be seen in Fig. 12b (now Fig. 11b) as the currents moving towards northwest, and accompanied by salinity and temperature increases. The text inserted above (i.e., "The long periods…") replaces the text mentioned in lines 16-17 to make it clearer.

**Created with Latexdiff**

[revised manuscript text omitted]

---

## Author Response (AR3)

**Associate Editor Decision: Publish subject to technical corrections** (15 Mar 2019) by Tina Treude

Comments to the Author:

Dear Dr. Ehn and Co-Authors,

Thank you for the submission of the revised manuscript. I am satisfied by your responses to the referees and your revision improved the clarity of the manuscript. I consider it acceptable subject to one technical correction:

Please remove the weblink in the Figure 9 caption (and any other weblink should I have missed it). Weblinks are not permanent and subject to change, while your publication will last much longer. Hence, a weblink could become invalid in the future. Please replace the weblink with a permanent citation.

With kind regards
Tina Treude

REPLY: We have removed the weblinks as requested. There are no other weblinks in the document. We have also added a mention of the Beaufort Gyre Exploration Project in the Acknowledgement section. Thank you.

The figures in this submitted version has not changed from the previous submission. However, we have high resolution pdf Fig. 2 that we would like to submit at a little later stage. We would also like to redraw the salinity contour lines in Fig. 2a to include near shore coastal observations. Presently the lead authors is travelling, and is thus not in a position to make the changes before the submission deadline.